# Closed-form $\ell_r$ norm scaling with data for overparameterized linear regression and diagonal linear networks under $\ell_p$ bias

**Shuofeng Zhang**

Rudolf Peierls Centre for Theoretical Physics

University of Oxford

shuofeng.zhang@physics.ox.ac.uk

**Ard Louis**

Rudolf Peierls Centre for Theoretical Physics

University of Oxford

ard.louis@physics.ox.ac.uk

## Abstract

For overparameterized linear regression with isotropic Gaussian design and minimum-$\ell_p$ interpolator with $p \in (1, 2]$, we give a unified, high-probability characterization for the scaling of the family of parameter norms $\{\|\widehat{w_p}\|_r\}_{r \in [1,p]}$ with sample size. We solve this basic, but unresolved question through a simple dual-ray analysis, which reveals a competition between a signal *spike* and a *bulk* of null coordinates in $X^\top Y$, yielding closed-form predictions for (i) a data-dependent transition $n_\star$ (the "elbow"), and (ii) a universal threshold $r_\star = 2(p - 1)$ that separates $\|\widehat{w_p}\|_r$'s which plateau from those that continue to grow with an explicit exponent. This unified solution resolves the scaling of *all* $\ell_r$ norms within the family $r \in [1, p]$ under $\ell_p$-biased interpolation, and explains in one picture which norms saturate and which increase as $n$ grows. We then study diagonal linear networks (DLNs) trained by gradient descent. By calibrating the initialization scale $\alpha$ to an effective $p_{\text{eff}}(\alpha)$ via the DLN separable potential, we show empirically that DLNs inherit the same elbow/threshold laws, providing a predictive bridge between explicit and implicit bias. Given that many generalization proxies depend on $\|\widehat{w_p}\|_r$, our results suggest that their predictive power will be highly sensitive to which $\ell_r$ norm is used.

## 1 Introduction

Many modern generalization measures for machine learning tasks are anchored on the parameter norm instead of parameter count (Neyshabur et al., 2015c;a; Yoshida & Miyato, 2017; Miyato et al., 2018; Cisse et al., 2017). Yet, most analyses of overparameterized regression still treat "the norm" monolithically—typically defaulting to $\ell_2$. If one is going to use a parameter norm, *which $\ell_r$* should be used, and how does that choice interact with the inductive bias that selects the interpolator (e.g., minimum-$\ell_p$)? This question has been comparatively less studied. We address this question first in a simpler but core setting—linear regression—and then connect the picture to diagonal linear networks (DLNs). Our experiments reveal that sweeping $(r, p)$ produces non-trivial behavior: even for the *same* interpolating predictor, some $\ell_r$ norms plateau while others keep growing with distinct slopes; in mixed cases, the elbow's location shifts with $p$, and *which $r$'s* plateau depends on the setting.

In linear regression it is well understood that the value of $p$ *shapes* the inductive bias (sparser as $p \downarrow 1$, denser as $p \uparrow 2$), making the $r$–$p$ interaction concrete. Beyond explicit $\ell_p$ penalties, first-order optimization can *implicitly* select a geometry: in overparameterized linear regression, gradient methods recover the minimum-$\ell_2$ interpolant; in separable classification, gradient descent converges to a max-margin solution; and in diagonal/deep linear parameterizations, the separable potentials governing the dynamics interpolate between sparse- and dense-leaning behaviors depending on initialization and parameterization (Tibshirani, 1996; Frank & Friedman, 1993; Hoerl & Kennard, 1970; Chen et al., 2001; Zou & Hastie, 2005; Hastie et al., 2015; 2022; Soudry et al., 2018; Gunasekar et al., 2018; Ji & Telgarsky, 2019b; Chizat et al., 2019; Woodworth et al., 2020). This variety of

explicit/implicit pathways for $p$-like biases motivates our unified treatment of the *family* $\{\|\widehat{w}\|_r\}$ and explains why different $\ell_r$ proxies can exhibit qualitatively different $n$-dependence under a fixed training pipeline.

Concretely, we study the minimum-$\ell_p$ interpolator in high-dimensional linear regression with isotropic Gaussian design ($d > n$, $p \in (1,2]$), and we characterize—*in closed form and with high probability*—how the entire family $\{\|\widehat{w}_p\|_r\}_{r \in [1,p]}$ scales with $n$. A one-dimensional *dual–ray* analysis exposes a competition between a signal *spike* and a high-dimensional *bulk* in $X^\top Y$, yielding: (i) a data-dependent transition size $n_\star$ (an elbow in $n$), and (ii) a universal threshold $r_\star = 2(p-1)$ that separates norms that ultimately plateau ($r > r_\star$) from those that continue to grow with explicit exponents ($r \leq r_\star$). We also extend the picture to DLNs trained by gradient descent: calibrating the initialization scale $\alpha$ via the network's separable potential that gives an *effective* exponent $p_{\text{eff}}(\alpha)$, and with this calibration the observed $\ell_r$–vs–$n$ curves inherit the same elbow/threshold structure as explicit minimum-$\ell_p$ interpolation. *When the inductive bias is unknown a priori*—e.g., the operative $p$ of the training pipeline is unclear—our results imply that choosing the "right" $r$ for norm-based generalization measures can be delicate, since different $(r,p)$ pairs can produce opposite scaling behaviors (plateau vs. growth) as $n$ increases.

*Our contributions:*

1. **Strong sensitivity of the parameter norm as a function of the pair** $(r,p)$  We find a strong *qualitative* effect for the scaling of the parameter norm with data: for fixed $p$, certain $\ell_r$ norms plateau while others grow with different slopes; varying $p$ moves the elbow and reassigns which $r$'s plateau.

2. **Closed-form scaling laws for parameter norms.** We derive the first unified closed-form scaling laws for this problem. For $p \in (1,2]$ and all $r \in [1,p]$, we identify the universal threshold $r_\star = 2(p-1)$, give an explicit expression for the transition size $n_\star$, and provide plateau levels and growth exponents in both spike- and bulk-dominated regimes via a compact dual–ray argument.

3. **Extension of our theory to DLNs.** We map the DLN initialization scale to geometry: $\alpha \mapsto p_{\text{eff}}(\alpha)$. Using this map, we transfer the theory to DLNs and verify the predicted elbow/threshold behavior of the parameter norm empirically.

**Implications for practice.**   Because many norm-based generalization measures and diagnostics depend on $\|\widehat{w}\|_r$, our results imply that practitioners using norm-based bounds or proxies—especially in more complex models such as DNNs—should be cautious: conclusions can be *highly sensitive* to the choice of $r$, and the sensitivity depends on the underlying $\ell_p$ bias that selects the interpolator.

## 2   RELATED WORK

The focus of this paper is a basic question: for overparameterized linear regression and related diagonal linear networks (DLNs), how do the *parameter norms* $\{\|\widehat{w}\|_r\}_{r \in [1,p]}$ scale with sample size when the interpolator is selected by an $\ell_p$ bias? The links to generalization are therefore indirect: norm quantities often appear as proxies in modern generalization measures (Neyshabur et al., 2015b; Bartlett et al., 2017; Dziugaite & Roy, 2017), so understanding their $n$–scaling is informative, but we do not develop new generalization bounds here. Relatedly, recent analyses derive explicit norm upper bounds as intermediate steps toward generalization—often via Gaussian min–max techniques—for interpolators and max-margin procedures (Koehler et al., 2021; Donhauser et al., 2022).

**The $\ell_r$ family of linear–regression interpolators.**   A large body of work characterizes how explicit $\ell_p$ penalties shape linear estimators: ridge/Tikhonov ($\ell_2$) (Hoerl & Kennard, 1970), lasso ($\ell_1$) (Tibshirani, 1996; Efron et al., 2004; Knight & Fu, 2000; Zou, 2006), elastic net (mixtures of $\ell_1$ and $\ell_2$) (Zou & Hastie, 2005), and the bridge family ($\ell_p$ for $0 < p \leq 2$) (Frank & Friedman, 1993); basis pursuit gives the sparse interpolating extreme under equality constraints (Chen et al., 2001; Candès & Tao, 2007; Donoho, 2006; Bickel et al., 2009). High–dimensional convex-geometric analyses explain when these programs select structured solutions and how their solutions move with the data geometry (Chandrasekaran et al., 2012; Amelunxen et al., 2014; Bühlmann & van de Geer, 2011; Wainwright, 2019), and recent developments give precise characterizations for ridgeless (minimum-$\ell_2$) interpolation and its risk (Hastie et al., 2022;?). Our contribution complements this

literature by treating the *entire* norm family $\{\|\widehat{w}_p\|_r\}_{r \in [1,p]}$ for minimum–$\ell_p$ *interpolators* (with $p \in (1, 2]$) and deriving closed-form, high-probability scaling laws in $n$ across $r$. In this sense we move from "which $p$ shapes which estimator?" to "given $p$, how do all $\ell_r$ diagnostics behave as data grow?"

**Overparameterization in regression and deep networks.** The deep-learning era stimulated a re-examination of overparameterized regression, revealing phenomena such as double descent (Belkin et al., 2019; Nakkiran et al., 2021; Zhang et al., 2017; Nakkiran et al., 2020a; Adlam & Pennington, 2020) and benign overfitting for minimum-norm interpolators (Bartlett et al., 2020; Hastie et al., 2022; Muthukumar et al., 2020). These results show that linear regression can capture qualitative behaviors seen in deep learning models and that the *selected* interpolator's norm matters for risk. Our work leverages this bridge as motivation only: by explaining, in closed form, which $\ell_r$ norms plateau and which grow (and at what rates) under an $\ell_p$ bias, we clarify what one should expect from norm-based proxies commonly used in deep-net analyses. Because practical pipelines for deep models rarely specify the effective $p$, our finding that $\|\widehat{w}_p\|_r$ depends sensitively on the *pair* $(r, p)$ suggests caution when interpreting norm-based generalization diagnostics.

**Explicit/implicit regularization and DLNs.** Beyond explicit penalties, optimization can select solutions with an *implicit* geometry (Soudry et al., 2018; Lyu & Li, 2020; Gunasekar et al., 2018; 2017). In overparameterized linear regression, gradient methods recover the minimum-$\ell_2$ interpolant; in factorized or deep-linear parameterizations, the training dynamics induce separable potentials that interpolate between sparse- and dense-leaning behaviors depending on initialization and parameterization (Saxe et al., 2014b; Gunasekar et al., 2018; Ji & Telgarsky, 2019b; Chizat et al., 2019; Woodworth et al., 2020). We build on this perspective for DLNs: by calibrating the initialization scale to an effective $p_{\text{eff}}$, we show empirically that DLNs inherit the same elbow/threshold laws for $\{\|\widehat{w}\|_r\}$ as explicit minimum–$\ell_p$ interpolation.

**Proof techniques.** Our analysis borrows standard high-dimensional tools used throughout the modern regression literature—Gaussian concentration, blockwise (signal-vs-bulk) decompositions, and dual certificates in convex programs (Vershynin, 2018; Tropp, 2015)—and combines them with a one-dimensional "dual–ray" reduction tailored to the $\ell_p$ penalty. Two closely related works derive norm *upper* bounds as an intermediate step toward generalization, using the Gaussian Min–Max Theorem (GMT) and its convex analogue (CGMT): Koehler et al. (2021); Donhauser et al. (2022). Their GMT/CGMT-based proofs are conceptually similar in spirit; by contrast, our argument proceeds from first principles via a simple dual–ray balance and yields closed-form $n$-scaling laws without invoking GMT/CGMT (see also Gordon (1985); Thrampoulidis et al. (2014) for the GMT and CGMT statements).

## 3 FAMILY OF NORM MEASURES OF MINIMUM $\ell_p$-NORM INTERPOLATOR IN LINEAR MODELS

We now formalize the object introduced in the overview: for $p \in (1, 2]$ in overparameterized linear regression, we study the family $\{\|\widehat{w}_p\|_r\}_{r \in [1,p]}$ where $\widehat{w}_p$ is the minimum-$\ell_p$ interpolator. Our goal is to characterize how these norms scale with sample size $n$. Our results identify (i) a data-dependent elbow $n_\star$ and (ii) a universal threshold $r_\star = 2(p - 1)$ that separates plateauing from growing $\ell_r$'s.

**Data and settings.** We consider overparameterized linear models with $X \in \mathbb{R}^{n \times d}$, $d > n$, rows i.i.d. $\mathcal{N}(0, I_d)$, and

$$Y = Xw^\star + \xi, \qquad \xi \sim \mathcal{N}(0, \sigma^2 I_n).$$

The minimum-$\ell_p$ interpolator is

$$\widehat{w}_p \in \arg\min_{w \in \mathbb{R}^d} \|w\|_p \quad \text{s.t.} \quad Xw = Y, \qquad p \in (1, 2].$$

Let $s = \|w^\star\|_0$ denote the support size and $\tau_s^2 := \|w^\star\|_2^2 + \sigma^2$. In contrast to interesting recent work by Donhauser et al. (2022), our theory is *not* restricted to the $w^* = e_1$ limit of extreme sparse regression.

## 3.1 MAIN THEOREM

**Theorem 3.1** ($\ell_r$ scaling of minimum-$\ell_p$ interpolators). *Fix* $p \in (1, 2]$, *set* $q = \frac{p}{p-1}$, *and take* $r \in [1, p]$. *Assume*

$$\frac{d}{n} \to \kappa \in (1, \infty) \quad and \quad \liminf_{n \to \infty} \frac{d-s}{n} = \kappa_{\text{bulk}} > 0.$$

*Let* $w^\star$ *have support* $S$ *with* $|S| = s$, *and let*

$$\widehat{w}_p \in \arg\min_{w \in \mathbb{R}^d} \|w\|_p \quad s.t. \quad Xw = Y.$$

*Write* $W_q := \|w^\star\|_q^q$ *and* $m_t := \mathbb{E}|Z|^t$ *for* $Z \sim \mathcal{N}(0, 1)$. *Define the* ray scale $t_\star$ *via*

$$t_\star^{q-1} = \frac{\|Y\|_2^2}{\|X^\top Y\|_q^q} \asymp \frac{\tau_s^2 n}{\underbrace{n^q W_q}_{\text{spike}} + \underbrace{(d-s)\, m_q\, \tau_s^q\, n^{q/2}}_{\text{bulk}} + \underbrace{O\big(\tau_s^q\,(s\, n^{q/2} + s^{1+q/2})\big)}_{\text{remainder}}} \quad \text{w.h.p..} \quad (1)$$

*Then, w.h.p. (see Remark A.1),*

$$\|\widehat{w}_p\|_r \asymp \max\Big\{\, t_\star^{q-1}\, n^{q-1}\, \|w^\star\|_{(q-1)r}^{q-1}, \ (d-s)^{1/r} \big(t_\star\, \tau_s \sqrt{n}\big)^{q-1},$$

$$s^{\max\{1/r,\, (q-1)/2\}} \big(t_\star\, \tau_s \sqrt{n}\big)^{q-1}\Big\}. \quad (2)$$

*Introduce the* transition scale

$$n_\star \asymp \Big(\kappa_{\text{bulk}} \frac{\tau_s^q}{W_q}\Big)^{\frac{2}{q-2}}. \quad (3)$$

*In the two extremes, we obtain:*

**Spike-dominated** ($n \gg n_\star$):

$$\|\widehat{w}_p\|_r \asymp \begin{cases} \dfrac{\tau_s^{q+1}}{W_q}\, n^{\frac{1}{r} - \frac{1}{2(p-1)}}, & r \le 2(p-1), \\[2mm] \dfrac{\tau_s^2}{W_q}\, \|w^\star\|_{(q-1)r}^{q-1}, & r > 2(p-1). \end{cases} \quad (4)$$

**Bulk-dominated** ($n \ll n_\star$):

$$\|\widehat{w}_p\|_r \asymp \max\Big\{\, \kappa_{\text{bulk}}^{\frac{1}{r}-1}\, \tau_s\, n^{\frac{1}{r}-\frac{1}{2}}, \ \kappa_{\text{bulk}}^{-1}\, \tau_s^{2-q}\, \|w^\star\|_{(q-1)r}^{q-1}\, n^{\frac{q}{2}-1},$$

$$\kappa_{\text{bulk}}^{-1}\, \tau_s\, s^{\max\{1/r,\, (q-1)/2\}}\, n^{-1/2}\Big\}. \quad (5)$$

*Since* $d - s \asymp \kappa_{\text{bulk}} n$, *the last term equals* $\frac{\tau_s}{d-s}\, s^{\max\{1/r,\, (q-1)/2\}} \sqrt{n}$. *All* $\asymp$ *hide absolute constants depending only on* $(p, \kappa_{\text{bulk}}, r)$.

*Remark* 3.2 (Dual viewpoint). The constrained problem $\min_w \frac{1}{p}\|w\|_p^p$ s.t. $Xw = Y$ has unconstrained dual $\max_\lambda \lambda^\top Y - \frac{1}{q}\|X^\top \lambda\|_q^q$, with KKT conditions $Xw = Y$ and $X^\top \lambda = \nabla f(w)$. Restricting to the ray $\lambda = tY$ yields $t_\star^{q-1} = \|Y\|_2^2 / \|X^\top Y\|_q^q$. The "spike" vs. "bulk" terminology refers to which part of $\|X^\top Y\|_q$ controls $t_\star$.

*Proof sketch.* The behavior of the minimum-$\ell_p$ interpolator can be read through a simple dual lens: rather than track the optimizer directly, we examine a dual certificate that both fits the labels and respects a norm budget after passing through the design; pushing the dual along the label direction (a one-dimensional "ray") reveals a single diagnostic scale where the budget tightens, and this scale is controlled by two competing sources in the correlations $X^\top Y$: a "spike" part (true signal coordinates) that coherently accumulates with $n$, and a "bulk" part (many null coordinates) that aggregates small, mostly noisy effects. Balancing these two contributions defines a data-dependent transition sample size $n_\star$: for $n \ll n_\star$ the bulk dominates, the solution's mass is effectively spread over many coordinates, and the family $\{\|\widehat{w}_p\|_r\}$ grows with $n$ in the way our bulk formulas predict

(including the usual cross-$r$ ordering and an $n^{1/2}$-type trend visible in the plots); for $n \gg n_\star$ the spike dominates, mass concentrates on the support, and a clean threshold—determined by $p$ at $r = 2(p-1)$—splits the outcomes: $\ell_r$ plateaus for $r$ above the threshold and grows with a gentler, explicit slope for $r$ below it. Standard concentration for Gaussian designs justifies the spike/bulk decomposition and the stability of the ray scale, and the KKT linkage between the dual certificate and the primal coordinates turns these ingredients into the unified bound, the expression for $n_\star$, and the two regime descriptions stated in the theorem. Full details are deferred to Appendix A. □

## 4 COROLLARIES FOR CANONICAL TARGETS

To make the unified laws in Theorem 3.1 concrete, we specialize them to two canonical targets: (i) a single spike $w^\star = e_1$, and (ii) a flat $s$-sparse vector with equal magnitude $a$ on its support. Substituting the problem-specific scales $W_q = \|w^\star\|_q^q$ and $\tau_s^2 = \|w^\star\|_2^2 + \sigma^2$ into the elbow formula equation 3 and the spike-/bulk-dominated expressions equation 4–equation 5 yields closed-form, high-probability predictions for $\|\widehat{w}_p\|_r$ and the transition size $n_\star$. We record these specializations below as Corollaries 4.1 and 4.2, and use them as reference overlays in our experiments.

### 4.1 SINGLE SPIKE

**Corollary 4.1** (Single spike). *Under Theorem 3.1 with $w^\star = e_1$ and $\tau^2 = 1 + \sigma^2$, for any $r \in [1, p]$:*

$$\text{Bulk-dominated } (n \ll n_\star): \quad \|\widehat{w}_p\|_r \; \asymp \; \tau \, (d-1)^{\frac{1}{r}-1} \, n^{1/2},$$

$$\text{Spike-dominated } (n \gg n_\star): \quad \|\widehat{w}_p\|_r \; \asymp \; \begin{cases} \tau^{\,q+1} \, n^{\frac{1}{r} - \frac{1}{2(p-1)}} & \text{if } r \leq 2(p-1), \\ \tau^{\,2} & \text{if } r > 2(p-1). \end{cases}$$

**Interpretation.** Here $W_q = 1$ and $n_\star \asymp (\kappa_{\text{bulk}} \tau^{\,q})^{2/(q-2)}$ from equation 3. For $r > 2(p-1)$ the $\ell_r$ curves *plateau* at level $\asymp \tau^2$ once $n \gg n_\star$; for $r \leq 2(p-1)$ they continue to grow with slope $\frac{1}{r} - \frac{1}{2(p-1)}$.

### 4.2 FLAT SUPPORT

**Corollary 4.2** (Flat support). *Under Theorem 3.1 and a flat $w^\star$ on $S$ with $|S| = s$ and $w_j^\star = a\, s_j$ for $j \in S$ ($|s_j| = 1$), for any $r \in [1, p]$, w.h.p.:*

$$\text{Spike-dominated } (n \geq C n_\star): \quad \|\widehat{w}_p\|_r \; \asymp \; \begin{cases} \dfrac{(sa^2 + \sigma^2)^{\frac{q+1}{2}}}{s|a|^q} \, n^{\frac{1}{r} - \frac{1}{2(p-1)}} & r \leq 2(p-1), \\[2mm] s^{\frac{1}{r} - 1} \, \dfrac{sa^2 + \sigma^2}{|a|} & 2(p-1) < r \leq p, \end{cases}$$

$$\text{Bulk-dominated } (n \leq c n_\star): \quad \|\widehat{w}_p\|_r \; \asymp \; \max\Big\{ \kappa_{\text{bulk}}^{\frac{1}{r}-1} \, \tau_s \, n^{\frac{1}{r} - \frac{1}{2}}, \; \kappa_{\text{bulk}}^{-1} \tau_s^{2-q} \, s^{1/r} |a|^{q-1} \, n^{\frac{q}{2}-1}, $$
$$\kappa_{\text{bulk}}^{-1} \, \tau_s \, s^{\max\{1/r, \, (q-1)/2\}} \, n^{-1/2} \Big\}.$$

**Interpretation.** Here $W_q = s|a|^q$ and $\tau_s^2 = sa^2 + \sigma^2$, so equation 3 gives $n_\star \asymp (\kappa_{\text{bulk}} \tau_s^{\,q}/(s|a|^q))^{2/(q-2)}$, which grows with $s$ (the elbow shifts to larger $n$). In the spike-dominated plateau branch ($r > 2(p-1)$) the level scales as $s^{\frac{1}{r}-1} (sa^2 + \sigma^2)/|a|$, which is typically of the same order as the single-spike plateau for moderate $s$.

**Comparison across targets.** The threshold $r = 2(p-1)$ and the $n$-exponents in both regimes are *unchanged* between Corollaries 4.1 and 4.2. The differences lie in the *scales*: (i) the transition size moves from $n_\star \asymp (\kappa_{\text{bulk}} \tau^{\,q}/W_q)^{2/(q-2)}$ with $W_q = 1$, $\tau^2 = 1 + \sigma^2$ (single spike) to $n_\star \asymp (\kappa_{\text{bulk}} \tau_s^{\,q}/W_q)^{2/(q-2)}$ with $W_q = s|a|^q$, $\tau_s^2 = sa^2 + \sigma^2$ (flat), which scales roughly linearly in $s$ (cf. equation 3). Hence the elbow for regime change shifts to *larger* $n$ when we move from $e_1$ to a flat $w^\star$ with $s = 50$. (ii) In the spike-dominated plateau branch ($r > 2(p-1)$), the level changes from

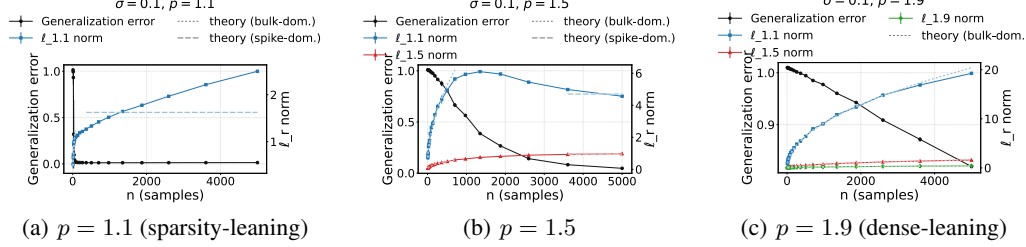

(a) $p = 1.1$ (sparsity-leaning)  (b) $p = 1.5$  (c) $p = 1.9$ (dense-leaning)

Figure 1: **Single spike $w^\star = e_1$; explicit minimum-$\ell_p$ interpolation.** Ordering across $r$ and the presence/absence of elbows follow Corollary 4.1; the bulk panels rise like $n^{1/2}$ and the spike-side panels plateau for $r > 2(p-1)$, consistent with equation 4-equation 5.

$\asymp \tau^2$ (single spike) to $\asymp s^{\frac{1}{r}-1} (sa^2+\sigma^2)/|a|$ (flat) [cf. equation 4 and Corollary 4.2]; for moderate $s$ this produces *comparable* numerical magnitudes, which is why the vertical ranges in our figures are similar. The regime labels (bulk vs. spike) and their slopes/plateaus therefore provide the informative contrast.

### 4.3 LINEAR REGRESSION WITH EXPLICIT MINIMUM-$\ell_p$ BIAS

Here the inductive bias is explicit: for a chosen $p$, the interpolator is the minimum-$\ell_p$ element among all $w$ with $Xw = Y$. Sweeping $p$ slides the solution from a more sparse-leaning geometry as $p \downarrow 1$ toward a more dense-leaning geometry as $p \uparrow 2$, revealing how the objective itself shapes the family $\{\|\widehat{w}_p\|_r\}_r$.

**Experimental protocol.**  We set $d = 50,000$, $\sigma = 0.1$, sweep $p \in \{1.1, 1.5, 1.9\}$, and vary $n$. Each plot overlays test MSE (left axis) and representative $\ell_r$ curves (right axis). For flat $w^*$ experiments, we kept $\|w^*\|_2 = 1$, i.e. $a = \frac{1}{\sqrt{s}}$. Additional noise sweeps are reported in Appendix C.

*What the figures show and why.*  In Fig. 1 (single spike), the left/middle/right panels follow the corollary's regime rules. In the left panel, for $r > 2(p-1)$ the curves flatten after the transition, while for smaller $r$ they retain the predicted growth; thin reference overlays (where present) trace these slopes. The middle panel exhibits a clear elbow near the predicted $n_\star$, which is close to $1.4 \times 10^3$ under our experimental parameters; beyond it, the $r > 2(p-1)$ curves plateau in line with equation 4, while the others keep their slope. The right panel stays bulk-dominated across the range, with the $\ell_r$ traces growing approximately as $n^{1/2}$ and ordered across $r$ as the bulk formula prescribes.

In Fig. 2 (flat $w^\star$ with $s$=50), the *same* slope/plateau rules apply, but the transition scale is larger: the elbow for $p$=1.5 appears at a later $n$ (or just off-range), consistent with $n_\star$ increasing roughly linearly with $s$ in equation 3. Across panels, the absolute $\ell_r$ values are numerically similar to Fig. 1; this matches the flat-support plateau level in Corollary 4.2, which for moderate $s$ is close to the single-spike level. The informative distinction is thus *where* the curves switch from bulk growth to spike plateaus and the persistence of the $n^{1/2}$ slope in regimes that remain bulk-dominated.

**Experiments with larger sparsity.**  We repeat the explicit minimum-$\ell_p$ runs at larger supports, $s \in \{500, 5000\}$, with the same $\|w^\star\|_2$=1 and noise level ($\sigma = 0.1$); see Appendix E, Figs. S13-S14. The qualitative picture from $s$=50 reappears but shifts to larger $n$, consistent with the transition size $n_\star$ in equation 3 growing with $s$. For small $p$ ($p$=1.1), the prolonged bulk-dominated window makes the *double-descent* pattern visible—generalization error first *increases* and then drops (most clearly at $s$=5000)—while the blue $\ell_{1.1}$ curve keeps rising along the bulk guide across the plotted range (Belkin et al., 2019; Nakkiran et al., 2020b; Hastie et al., 2022). For larger $p$ ($p$=1.5, 1.9), the curves remain monotonically decreasing; the minimized $\ell_p$ traces drift only mildly upward (no flattening within the range), reflecting the rounder geometry that avoids early over-reliance on noisy bulk directions. In all panels, the dashed overlays track the bulk/spike trends and the expected $r$-ordering of the $\ell_r$ diagnostics, matching the regime structure highlighted in the theory.

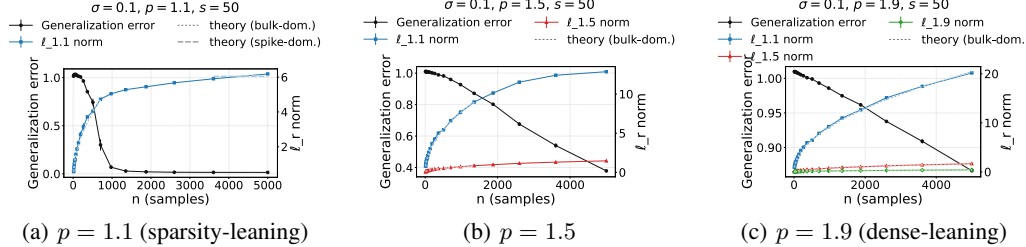

(a) $p = 1.1$ (sparsity-leaning)    (b) $p = 1.5$    (c) $p = 1.9$ (dense-leaning)

Figure 2: **Flat $w^\star$ ($s = 50$); explicit minimum-$\ell_p$ interpolation.** The scaling rules mirror the flat-support corollary: bulk growth persists until a larger transition scale, while spike-side $r$ values plateau; absolute levels are comparable to the single-spike case, as predicted by the plateau formulas.

## 4.4 DIAGONAL LINEAR NETWORK WITH IMPLICIT BIAS

Diagonal linear networks (DLNs) - deep linear models whose weight matrices are diagonal so that the effective predictor is the coordinatewise product of layer parameters—provide a tractable testbed for understanding optimization-induced geometry and implicit bias in overparameterized systems. They connect classical analyses of linear nets and factorized parameterizations (Saxe et al., 2014a; Ji & Telgarsky, 2019a; Arora et al., 2019; Gunasekar et al., 2017) with recent perspectives on how initialization and parameterization interpolate between "rich" and "kernel" behaviors (Chizat et al., 2019; Woodworth et al., 2020). A particularly useful feature—formalized for DLNs via a separable gradient-flow potential—is that the *scale of the initialization*, denoted $\alpha$, *continuously tunes* the implicit bias: small $\alpha$ yields a sparse-leaning geometry (an $\ell_1$-like penalty up to logarithmic factors), while large $\alpha$ approaches an $\ell_2^2$-type geometry; see the potential $Q_\alpha$ and its limits (Theorem 1 in Woodworth et al. (2020)) and related characterizations in Gunasekar et al. (2017); Arora et al. (2019).

**Calibrating $\alpha$ via an effective $p$.** To compare DLN runs with our explicit minimum-$\ell_p$ experiments, we convert $\alpha$ into an *effective $p$* by a data-free calibration. Following the separable potential view, we evaluate $Q_\alpha$ on $k$-sparse, unit-$\ell_2$ probes and fit the log-log slope of its $k$-dependence; matching that slope to the exact $k^{1-p/2}$ law of $\| \cdot \|_p^p$ yields a monotone map $\alpha \mapsto p_{\text{eff}}(\alpha)$ with limits $p_{\text{eff}}(\alpha) \to 1$ as $\alpha \to 0$ and $p_{\text{eff}}(\alpha) \to 2$ as $\alpha \to \infty$. This calibration is independent of $(n, \sigma)$ and lets us select $\alpha$ values that span a sparse-to-dense range comparable to $p \in \{1.1, 1.5, 1.9\}$. A full derivation and a visualization of $\alpha \mapsto p_{\text{eff}}(\alpha)$ are provided in Appendix B.

**Finite learning rate.** With a single-spike target ($w^\star = e_1$, sparsity $s{=}1$) and small initialization ($\alpha = 0.00102$, so $p_{\text{eff}} \approx 1.10$), we find that the learning rate lr can materially change the $\ell_r$-vs-$n$ scaling once label noise is present. When $\sigma{=}0$, the $\ell_{1.1}$ curve rapidly plateaus and is essentially insensitive to lr (see Appendix D for more details). In contrast, for $\sigma \in \{0.1, 0.5\}$ increasing lr produces a steadily rising $\ell_{1.1}$ and shifts the elbow to larger $n$; at the highest noise the effect is strongest-lr$=10^{-1}$ yields monotone growth across our range, whereas lr$=10^{-3}$ exhibits a transient rise followed by relaxation toward a plateau, indicating a rightward-moving elbow. We observe qualitatively similar trends for larger sparsity ($s{=}50$). A natural explanation is that finite step size together with noisy gradients turns (stochastic) gradient descent into a noisy dynamical system with an *effective temperature* that scales with lr and the noise level. The resulting diffusion broadens the stationary distribution and biases the predictor toward rounder (less sparse) geometries-effectively increasing $p_{\text{eff}}$-so mass leaks into bulk coordinates, delaying spike dominance and inflating $\ell_r$ before the eventual plateau (Mandt et al., 2017; Smith et al., 2018; Yaida, 2018; Jastrzebski et al., 2017).

**Experimental protocol.** We set $d = 50,000$, $\sigma = 0.1$, sweep $\alpha \in \{0.00102, 0.0664, 0.229\}$ (which according to our $\alpha$ to $p$ calibration $\approx p \in \{1.1, 1.5, 1.9\}$), and vary $n$. Each plot overlays test MSE (left axis) and representative $\ell_r$ curves (right axis). For flat $w^*$ again $a = \frac{1}{\sqrt{s}}$. Additional noise sweeps are reported in Appendix C.

Because $\alpha$ has been empirically calibrated to $p_{\text{eff}}(\alpha)$, the DLN panels mirror the *scaling* behavior seen with explicit minimum-$\ell_p$: for $w^\star = e_1$ (Fig. 3), smaller $\alpha$ (smaller $p_{\text{eff}}$) enters the spike-dominated regime earlier so that, for $r > 2(p{-}1)$, the $\ell_r$ curves flatten after the transition; larger $\alpha$ remains

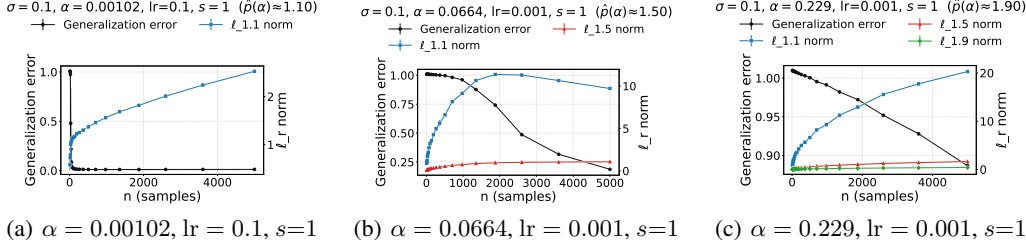

(a) $\alpha = 0.00102$, lr $= 0.1$, s=1  (b) $\alpha = 0.0664$, lr $= 0.001$, s=1  (c) $\alpha = 0.229$, lr $= 0.001$, s=1

Figure 3: **Single spike** $w^\star = e_1$**; diagonal linear network (DLN).** After calibrating $\alpha$ to $p_{\text{eff}}(\alpha)$, the regime structure matches the explicit $p$ case: smaller $\alpha$ exhibits earlier spike dominance and plateaus for $r > 2(p-1)$; larger $\alpha$ remains bulk-dominated with $n^{1/2}$-like growth.

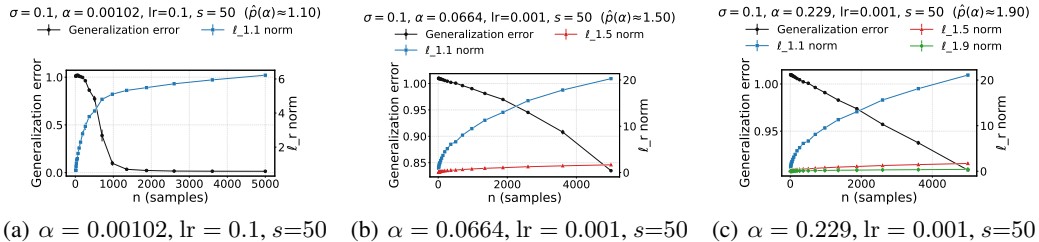

(a) $\alpha = 0.00102$, lr $= 0.1$, s=50  (b) $\alpha = 0.0664$, lr $= 0.001$, s=50  (c) $\alpha = 0.229$, lr $= 0.001$, s=50

Figure 4: **Flat** $w^\star$ **(s $= 50$); diagonal linear network (DLN).** The same scaling rules hold, but the elbow appears at larger $n$—in line with the flat-support transition scale—while absolute $\ell_r$ magnitudes remain comparable to the single-spike case.

bulk-dominated longer and the traces grow with the characteristic $n^{1/2}$ trend. For the flat target with $s=50$ (Fig. 4), the same rules apply but the elbow shifts to larger $n$, consistent with the $s$-dependent transition scale in the flat-support corollary. The absolute magnitudes of $\|\widehat{w}\|_r$ are similar across the two targets, as predicted by the plateau formulas, so the informative contrast again lies in the *location* of the elbow and the presence/absence of plateaus vs. bulk growth. We do not overlay theory on the DLN plots: our guarantees are stated in terms of the explicit parameter $p$, and deriving a closed-form $\alpha$-indexed analogue (especially under finite learning rates) is outside the scope of this work; the $\alpha \mapsto p_{\text{eff}}$ calibration serves precisely to make the scaling correspondence visible. In Appendix F we discuss how can we extend our main theorem to DLNs with explicit $\alpha$.

## 5 CONCLUSION AND DISCUSSION

We provided the first unified, closed-form characterization of how the entire family of norms $\{\|\widehat{w}_p\|_r\}_{r \in [1,p]}$ scales with sample size in overparameterized linear regression under minimum-$\ell_p$ interpolation ($p \in (1, 2]$). A one-dimensional dual-ray argument exposes a competition between a signal *spike* and a *bulk* of null coordinates in $X^\top Y$ and yields, with high probability: (i) a data-dependent elbow $n_\star$ at which bulk and spike balance [Eq. 8], and (ii) a universal threshold

$$r_\star = 2(p-1),$$

which separates $\ell_r$'s that ultimately plateau ($r > r_\star$) from those that continue to grow with an explicit exponent ($r \le r_\star$) in the spike-dominated regime (Theorem 3.1). The formulas give plateau levels and slopes in both bulk- and spike-dominated regimes, and specialize cleanly for canonical targets (single spike and flat support). Empirically, diagonal linear networks (DLNs) trained by gradient descent inherit the same elbow/threshold laws once the initialization scale $\alpha$ is calibrated to an effective $p_{\text{eff}}(\alpha)$ via the separable potential. Together, these results show that which $\ell_r$ one tracks matters: for a fixed $\ell_p$ bias, different $\ell_r$'s can exhibit qualitatively different $n$-laws.

**Intuition behind the regime transition.** The dual-ray lens reduces the interpolation geometry to a single scale $t_\star$ controlled by $\|X^\top Y\|_q$ ($q = p/(p-1)$). The *bulk* contributes $\asymp (d-s) m_q \tau_s^q n^{q/2}$

while the *spike* contributes $\asymp n^q W_q$, and their balance sets the elbow $n_\star$. Above the elbow, the KKT map raises correlations to the $(q-1)$ power; the sign of $\frac{1}{r} - \frac{1}{2(p-1)}$ dictates whether the bulk-type term decays (plateau) or grows (slope). This is the origin of the sharp threshold $r_\star = 2(p-1)$. Geometrically, smaller $p$ (sparser inductive bias) lowers $r_\star$, so more $\ell_r$'s plateau once the spike dominates; as $p \uparrow 2$, $r_\star$ approaches 2 and spike-side plateaus recede, consistent with the special role of $p = 2$ where there is no $n$-driven transition in the proportional limit.

**Implications for generalization proxies.** Many diagnostics and bounds in modern learning scale with a parameter norm (or a reparameterization-aware surrogate). Our results indicate that the predictive power of such proxies is *norm-choice sensitive*. For a given $\ell_p$ bias, $\ell_r$'s above $r_\star$ stabilize (after $n_\star$) and can serve as geometry-aligned capacity proxies, while $\ell_r$'s below $r_\star$ continue to reflect data growth through explicit exponents. In practice, the pair $(n_\star, r_\star)$ acts as a *norm-scaling signature*. Reporting only one norm—often $\ell_2$—risks conflating bulk vs. spike effects and can obscure regime changes that are visible in the $\ell_r$ family.

**From explicit to implicit bias.** By calibrating DLN initialization via a simple slope-matching map $\alpha \mapsto p_{\text{eff}}(\alpha)$, the empirical DLN curves line up with the explicit minimum-$\ell_p$ predictions under $p \leftarrow p_{\text{eff}}(\alpha)$. This provides a quantitative bridge between explicit and implicit bias: initialization steers the effective geometry, and the $(n_\star, r_\star)$ structure is inherited. Finite learning rates in the presence of label noise act like an effective temperature, increasing $p_{\text{eff}}$ and shifting elbows rightward—consistent with recent views of SGD as a noisy dynamical system.

**Relation to double descent and benign overfitting.** The bulk-side growth ($\propto n^{1/2}$ in prominent terms) and its eventual handoff to spike control rationalize when increasing $n$ first harms and then helps: early fits draw from many noisy bulk directions (large norms and higher variance), while beyond $n_\star$ the spike dominates and the relevant $\ell_r$'s plateau. Our explicit exponents and thresholds sharpen this picture and make precise which $\ell_r$ will display which trend at a given $(p, n)$.

**Scope and limitations.** Our guarantees assume isotropic Gaussian design, $p \in (1, 2]$, squared loss, and exact interpolation. At $p = 2$ the proportional regime admits no $n$-driven elbow. The DLN extension uses a data-free calibration to $p_{\text{eff}}(\alpha)$ rather than a fully rigorous, learning-rate-aware theory. Finally, classification losses and non-linear features (beyond DLNs) are outside our formal scope.

**Actionable guidance.** (i) When using norm-based capacity control, *choose the norm with the geometry*: if training is $\ell_p$-biased (explicitly or implicitly), track $\ell_r$ with $r > 2(p-1)$ to obtain a stable, post-elbow proxy; use $r \leq 2(p-1)$ when one *wants* a readout that continues to reflect data growth. (ii) Empirically estimate $(n_\star, r_\star)$ by fitting the predicted slopes to a small $\ell_r$ grid; this gives a compact fingerprint of model-data geometry and a practical meter for bulk vs. spike dominance.

*Future directions:*

- **Beyond isotropy and Gaussianity.** Extend the dual-ray analysis to anisotropic/sub-Gaussian designs (via whitening) and to heavy-tailed covariates; characterize how $n_\star$ and possibly $r_\star$ deform with the spectrum and tails of $X$.

- **From DLNs to nonlinear nets.** Replace the power link by depth-dependent implicit links in deep (nonlinear) architectures (e.g., path-norm or neural tangent/feature-learning regimes) and test whether an $r_\star$-type threshold persists.

- **Algorithmic knobs as geometry.** Develop a theory of $p_{\text{eff}}$ that accounts for step size, batch size, momentum, and label noise (Langevin/SGD limits), turning these knobs into quantitative geometric parameters with predictions for $(n_\star, r_\star)$.

- **Classification and margins.** Generalize the scaling laws to separable classification with cross-entropy/hinge losses, relating $r_\star$ to margin exponents and the growth/saturation of norm families along max-margin flows.

- **Tighter, $r$-aware bounds.** Build generalization bounds that track the *family* $\{\|\widehat{w}\|_r\}$ and explicitly incorporate the elbow/threshold structure, connecting to PAC-Bayes and margin-based analyses.

- **Practical diagnostics.** On modern deep models, measure several $\ell_r$-style surrogates (e.g., path norms) across data scale to estimate $(n_\star, r_\star)$ and evaluate which norms are reliable predictors of test error across regimes.

Overall, our results advocate replacing the monolithic notion of "the norm" by a *family* view. The elbow $n_\star$ and the threshold $r_\star$ provide simple, interpretable invariants that tie together explicit/implicit bias, data growth, and norm-based generalization measures, and they offer a compact vocabulary for describing—and ultimately controlling—interpolation in high dimensions.

**Reproducibility Statement**   All experiments in this work are fully reproducible with minimum hardware requirements. The anonymized code can be found at `https://github.com/shuofengzhang/minlp_codebase`.

**LLM usage**   General-purpose LLMs are used in this paper to aid or polish writing, e.g., checking typo/grammar slips.

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

## A  MINIMUM-$\ell_p$ INTERPOLATOR WITH $s$-SPARSE GROUND TRUTH

For completeness, we first introduce again the mathematical settings and restate our main theorem. We study $p \in (1, 2]$, set $q = \frac{p}{p-1} \in [2, \infty)$, and let $r \in [1, p]$. Dimensions $n, d \in \mathbb{N}$ with $d \geq n$. All $X \in \mathbb{R}^{n \times d}$ have i.i.d. $\mathcal{N}(0, 1)$ entries; columns are $X_{:,j}$. Noise $\xi \sim \mathcal{N}(0, \sigma^2 I_n)$, independent of $X$. The signal $w^\star \in \mathbb{R}^d$ is $s$-sparse with support $S \subset [d]$, $|S| = s$; we write $w_S^\star$ for its nonzeros. The response is $Y := Xw^\star + \xi$. The min-$\ell_p$ interpolator

$$\widehat{w}_p \in \arg\min\{\|w\|_p : Xw = Y\} \qquad (p > 1 \text{ ensures uniqueness})$$

is our object of interest. Shorthands:

$$\tau_s^2 := \|w^\star\|_2^2 + \sigma^2, \qquad W_q := \|w^\star\|_q^q = \sum_{j \in S} |w_j^\star|^q.$$

*Remark* A.1 (Standing assumptions and probability shorthand). We work in the proportional regime

$$\frac{d}{n} \to \kappa \in (1, \infty), \qquad \kappa_{\text{bulk}} := \liminf_{n \to \infty} \frac{d-s}{n} \in (0, \infty),$$

so $d - s = \Theta(n)$ and $s = O(n)$ (we do not require $s \leq n$). Unless stated otherwise, all hidden constants depend only on $(p, \kappa_{\text{bulk}})$ (and on $r$ when relevant), and "w.h.p." means probability at least $1 - Ce^{-cn} - 2d^{-\gamma}$. When we simplify remainders using $s \leq n$ (e.g., $\sqrt{sn} + s \rightsquigarrow \sqrt{sn}$), the corresponding $s > n$ form is always available and does not affect any $\asymp$ conclusions in Theorem A.2.

*On proportionality.* The assumption $d/n \to \kappa$ is only for cleanliness of exposition and to keep constants tidy; it is not essential to the argument. All places where it enters (e.g., the bulk $\ell_t$ embedding and the uniform column–norm control) can be run under the weaker—and often more realistic—conditions

$$\liminf_{n\to\infty} \frac{d-s}{n} = \kappa_{\text{bulk}} > 0, \qquad \log d = o(n), \qquad s = O(n).$$

In particular, our proofs and conclusions (same exponents in $n$, the threshold $r_\star = 2(p-1)$, and the high-probability events) remain valid even in "larger" aspect-ratio regimes (including $d/n \to \infty$) as long as $\log d = o(n)$ and the bulk density is bounded below. Under these weaker assumptions the hidden constants are uniform in $(n, d, s)$ and depend only on $(p, r, \kappa_{\text{bulk}})$ (and on a fixed upper bound for $s/n$ if desired), so no changes to the proofs are needed.

## A.1 Main theorem

**Theorem A.2** (Theorem 3.1 restated). *Fix* $p \in (1, 2]$, $q = \frac{p}{p-1}$, $r \in [1, p]$, *and suppose* $\liminf(d - s)/n = \kappa_{\text{bulk}} > 0$ *while* $d/n \to \kappa \in (1, \infty)$. *Then, w.h.p.,*

$$\|\widehat{w}_p\|_r \asymp \max\left\{ \underbrace{t_\star^{q-1}\, n^{q-1}\, \|w^\star\|_{(q-1)r}^{q-1}}_{\text{spike main }(S)},\; \underbrace{(d-s)^{1/r}\,\big(t_\star\,\tau_s\sqrt{n}\big)^{q-1}}_{\text{bulk }(S^c)},\; \underbrace{s^{\max\{1/r,\,(q-1)/2\}}\,\big(t_\star\,\tau_s\sqrt{n}\big)^{q-1}}_{\text{spike remainder}} \right\}.$$

$$\tag{6}$$

*where the* ray scale $t_\star$ *satisfies*

$$t_\star^{q-1} \asymp \frac{\|Y\|_2^2}{\|X^\top Y\|_q^q} \asymp \frac{\tau_s^2 n}{n^q W_q + (d-s)\, m_q\, \tau_s^q\, n^{q/2} + O\big(\tau_s^q(s\, n^{q/2} + s^{1+q/2})\big)} \qquad \text{w.h.p.} \tag{7}$$

*with* $m_t := \mathbb{E}|Z|^t$ *and* $Z \sim \mathcal{N}(0, 1)$. *Define the* dual-transition scale

$$n_\star \asymp \left(\kappa_{\text{bulk}} \frac{\tau_s^q}{W_q}\right)^{\frac{2}{q-2}}. \tag{8}$$

*Then, w.h.p., the following asymptotic simplifications hold:*

**Dual spike-dominated** $n \gg n_\star$.

$$\|\widehat{w}_p\|_r \asymp \begin{cases} \dfrac{\tau_s^{q+1}}{W_q}\, n^{\frac{1}{r} - \frac{1}{2(p-1)}}, & r \le 2(p-1), \\[2mm] \dfrac{\tau_s^2}{W_q}\, \|w^\star\|_{(q-1)r}^{q-1}, & r > 2(p-1). \end{cases} \tag{9}$$

**Dual bulk-dominated** $n \ll n_\star$.

$$\|\widehat{w}_p\|_r \asymp \max\left\{ \kappa_{\text{bulk}}^{\frac{1}{r}-1}\, \tau_s\, n^{\frac{1}{r}-\frac{1}{2}},\; \kappa_{\text{bulk}}^{-1}\, \tau_s^{2-q}\, \|w^\star\|_{(q-1)r}^{q-1}\, n^{\frac{q}{2}-1},\; \kappa_{\text{bulk}}^{-1}\, \tau_s\, s^{\max\{1/r,\,(q-1)/2\}}\, n^{-1/2} \right\}. \tag{10}$$

(Equivalently, using $d - s \asymp \kappa_{\text{bulk}} n$, the third term can be written as $\frac{\tau_s}{d-s}\, s^{\max\{1/r,(q-1)/2\}}\sqrt{n}$.)

*Remark* A.3 (When the third term is absorbed). If $r \le 2(p-1)$ and $s \le C\,(d-s)$ for an absolute constant $C$, then the third term in equation 5 is dominated by the first term (their ratio is $\lesssim (s/(d-s))^{1/r}$). In that case, equation 5 reduces to the two-term maximum

$$\|\widehat{w}_p\|_r \asymp \max\left\{ \kappa_{\text{bulk}}^{\frac{1}{r}-1}\, \tau_s\, n^{\frac{1}{r}-\frac{1}{2}},\; \kappa_{\text{bulk}}^{-1}\, \tau_s^{2-q}\, \|w^\star\|_{(q-1)r}^{q-1}\, n^{\frac{q}{2}-1} \right\}.$$

For $r > 2(p-1)$, no uniform absorption holds in general; the third term can dominate when $\|w^\star\|_{(q-1)r}$ is small relative to $\tau_s$.

*Remark* A.4 (Boundary $p = 2$). At $p = 2$ (so $q = 2$) the exponent in equation 8 diverges. In the proportional-$d$ regime ($d/n \to \kappa$) there is no $n$-driven transition; the relative sizes of the spike and bulk are constant-level. In the finite-$d$ regime (below) a concrete $n$-threshold does exist because $(d - s)$ does not scale with $n$.

## A.2 KEY LEMMAS AND PROOF OUTLINE

**Roadmap.** We prove Theorem 3.1 by (i) reducing the min-$\ell_p$ interpolator to a dual maximization and restricting the dual to the one-dimensional ray $\lambda = tY$, (ii) decomposing $\|X^\top Y\|_q^q$ into a spike term ($j \in S$) and a bulk term ($j \notin S$), and (iii) converting back to the primal via the KKT map, which raises correlations to the power $(q-1)$ and produces the three-term maximum in equation 6. The elbow at $r = 2(p-1)$ comes from the sign of $1/r - 1/(2(p-1))$, i.e., exactly whether the bulk-type contribution grows or plateaus in the spike-dominated regime. We work on a single high-probability event $\mathcal{E}$ (defined below) on which all concentration facts hold simultaneously.

**Global event.** Let $\mathcal{E}$ be the intersection of the column-norm, spectral, and bulk $\ell_t$ events from Lemmas A.5, A.8, and A.11. Then $\mathbb{P}(\mathcal{E}) \geq 1 - Ce^{-cn} - 2d^{-\gamma}$. All bounds below hold on $\mathcal{E}$.

### A.2.1 DUAL PROBLEM AND KKT

We briefly review Lagrangian duality for convex programs with equality constraints and then apply it to the minimum-$\ell_p$ interpolator.

**Primal problem and feasibility.** We consider

$$\min_{w \in \mathbb{R}^d} \ f(w) \quad \text{subject to} \quad Xw = Y, \qquad \text{with} \quad f(w) := \frac{1}{p}\|w\|_p^p,$$

where $p \in (1, 2]$. Since $X \in \mathbb{R}^{n \times d}$ has full row rank $n$ a.s. (for $d \geq n$ with i.i.d. $\mathcal{N}(0,1)$ entries), the affine constraint set $\{w : Xw = Y\}$ is nonempty for every $Y \in \mathbb{R}^n$. The objective $f$ is proper, closed, and *strictly convex* for $p > 1$ (indeed uniformly convex). Therefore, the primal minimizer $\widehat{w}_p$ exists and is unique. Introduce a Lagrange multiplier $\lambda \in \mathbb{R}^n$ for the equality constraint, and form the Lagrangian

$$\mathcal{L}(w, \lambda) := f(w) + \langle \lambda, Y - Xw \rangle.$$

The *dual function* is obtained by minimizing the Lagrangian over $w$:

$$g(\lambda) := \inf_{w \in \mathbb{R}^d} \left\{ f(w) - \langle X^\top \lambda, w \rangle \right\} + \langle Y, \lambda \rangle = -f^\star(X^\top \lambda) + \langle Y, \lambda \rangle,$$

where $f^\star$ is the convex conjugate of $f$:

$$f^\star(z) := \sup_{w \in \mathbb{R}^d} \left\{ \langle z, w \rangle - f(w) \right\}.$$

Since $f(w) = \sum_{i=1}^d |w_i|^p/p$ is separable, its conjugate is $f^\star(z) = \sum_{i=1}^d |z_i|^q/q = (1/q)\|z\|_q^q$, where $q = p/(p-1)$ is the Hölder conjugate of $p$. Indeed, for each coordinate

$$\sup_{t \in \mathbb{R}} \left\{ z\,t - |t|^p/p \right\}$$

is achieved at $t = \operatorname{sgn}(z)|z|^{q-1}$, with optimal value $|z|^q/q$. Therefore the dual function is

$$g(\lambda) = \langle Y, \lambda \rangle - \frac{1}{q}\|X^\top \lambda\|_q^q.$$

**Dual problem and strong duality.** The *dual problem* is $\max_{\lambda \in \mathbb{R}^n} g(\lambda)$, i.e.

$$\max_{\lambda \in \mathbb{R}^n} \ D(\lambda), \qquad D(\lambda) := \langle Y, \lambda \rangle - \frac{1}{q}\|X^\top \lambda\|_q^q.$$

This is a concave maximization problem (a smooth concave objective with no constraints). Strong duality holds in our setting by standard convex duality: the primal is convex, the constraint is affine, and feasibility holds (Slater's condition for equalities reduces to existence of a feasible point). Hence

$$\min_{w:\, Xw=Y} f(w) = \max_{\lambda \in \mathbb{R}^n} D(\lambda).$$

For convex programs with equality constraints, the Karush-Kuhn-Tucker (KKT) conditions are necessary and sufficient for optimality under strong duality. They read:

$$\text{(primal feasibility)} \quad Xw = Y, \qquad \text{(stationarity)} \quad 0 \in \partial f(w) - X^\top \lambda.$$

Because $p > 1$, $f$ is differentiable on $\mathbb{R}^d$ with gradient

$$\nabla f(w) = |w|^{p-2} \odot w = \text{sgn}(w) \odot |w|^{p-1},$$

so the subdifferential collapses to the singleton $\{\nabla f(w)\}$ and stationarity is

$$\nabla f(w) = X^\top \lambda.$$

At any primal-dual optimum $(\widehat{w}_p, \lambda^\star)$ we therefore have

$$X\widehat{w}_p = Y, \qquad X^\top \lambda^\star = \nabla f(\widehat{w}_p) = |\widehat{w}_p|^{p-2} \odot \widehat{w}_p. \tag{11}$$

The conjugate $f^\star$ is differentiable with $\nabla f^\star(z) = |z|^{q-2} \odot z = \text{sgn}(z) \odot |z|^{q-1}$, and the gradients are mutual inverses: $\nabla f^\star = (\nabla f)^{-1}$. Applying $\nabla f^\star$ to both sides of $X^\top \lambda^\star = \nabla f(\widehat{w}_p)$ gives the *coordinatewise KKT map*:

$$\widehat{w}_{p,i} = \left( \nabla f^\star(X^\top \lambda^\star) \right)_i = \text{sgn}\left( (X^\top \lambda^\star)_i \right) \left| (X^\top \lambda^\star)_i \right|^{q-1}. \tag{12}$$

Equivalently, $\widehat{w}_p = \nabla f^\star(X^\top \lambda^\star)$ and $X^\top \lambda^\star = \nabla f(\widehat{w}_p)$.

At optimality, Fenchel–Young gives $f(\widehat{w}_p) + f^\star(X^\top \lambda^\star) = \langle \widehat{w}_p, X^\top \lambda^\star \rangle$. Using $X\widehat{w}_p = Y$ and the expressions for $f$ and $f^\star$ yields the identities

$$\|X^\top \lambda^\star\|_q^q = \|\widehat{w}_p\|_p^p = \langle Y, \lambda^\star \rangle. \tag{13}$$

These will be used repeatedly to pass between the primal and dual scales.

The affine set $\{w : Xw = Y\}$ is a translate of $\ker(X)$, and minimizing $\|w\|_p$ over it finds the point where a scaled $\ell_p$ ball first touches this affine subspace. The *outer normal* to the $\ell_p$ ball at the touching point is $\nabla f(\widehat{w}_p) = |\widehat{w}_p|^{p-2} \odot \widehat{w}_p$, and the KKT condition $X^\top \lambda^\star = \nabla f(\widehat{w}_p)$ says that this normal lies in the row space of $X$. In coordinates, equation 12 shows that each coefficient of $\widehat{w}_p$ is a $(q-1)$-power of the correlation between the corresponding feature column $X_{:,i}$ and the dual multiplier $\lambda^\star$.

**Specialization at $p = 2$.** When $p = q = 2$, $\nabla f(w) = w$ and $\nabla f^\star(z) = z$. Then equation 11 reads $X^\top \lambda^\star = \widehat{w}_2$ and $X\widehat{w}_2 = Y$, which implies $XX^\top \lambda^\star = Y$ and hence $\lambda^\star = (XX^\top)^{-1}Y$. Therefore

$$\widehat{w}_2 = X^\top (XX^\top)^{-1}Y = X^+Y,$$

the minimum-$\ell_2$ (Moore–Penrose) interpolator. For $p \neq 2$ the same structure persists but the map $z \mapsto \nabla f^\star(z) = \text{sgn}(z)|z|^{q-1}$ is nonlinear, which is exactly what introduces the $(q-1)$-power in the subsequent spike/bulk analysis.

**Why duality helps here.** The dual objective

$$D(\lambda) = \langle Y, \lambda \rangle - \tfrac{1}{q}\|X^\top \lambda\|_q^q$$

separates the *data dependence* (linear in $Y$) from the *feature geometry* through $\|X^\top \lambda\|_q^q$. In our Gaussian design, the $d$ coordinates of $X^\top \lambda$ split naturally into the $s$ *spikes* (indices in $S$) and the $(d-s)$ *bulk*, for which we have precise $\ell_t$ concentration (Lemmas A.8 and A.11). Because $D$ is homogeneous in a simple way along the *ray* $\lambda = tY$,

$$D(t) = t\|Y\|_2^2 - \tfrac{t^q}{q}\|X^\top Y\|_q^q,$$

we will use the *ray scale* $t_\star$ (the maximizer of $D(tY)$) as a canonical scale for $\lambda^\star$; Lemma A.12 shows $\|\lambda^\star\|_2 \asymp t_\star\|Y\|_2$ and provides blockwise controls on $X^\top \lambda^\star$. The KKT map equation 12 then converts $\|X^\top \lambda^\star\|_{(q-1)r}^{q-1}$ into $\|\widehat{w}_p\|_r$, via $\||u|^{\odot(q-1)}\|_r = \|u\|_{(q-1)r}^{q-1}$, which is the backbone of the unified bound equation 6.

### A.2.2 Concentration for $Y$ and $X^\top Y$.

Let $m_t := \mathbb{E}|Z|^t$ for $Z \sim \mathcal{N}(0,1)$.

**Lemma A.5** (norm of $Y$). *With $Y := Xw^\star + \xi$ and $\tau_s^2 := \|w^\star\|_2^2 + \sigma^2$, we have*

$$\|Y\|_2^2 = \tau_s^2\, n\, (1 + o(1)) \qquad w.h.p.$$

*More quantitatively, for every $t > 0$,*

$$\Pr\Big(\big|\|Y\|_2^2 - \tau_s^2 n\big| \geq 2\tau_s^2\sqrt{nt} + 2\tau_s^2 t\Big) \;\leq\; e^{-t}.$$

*Proof.* For each row $i \in [n]$, $(Xw^\star)_i = \sum_{j=1}^d w_j^\star X_{i,j}$ is $\mathcal{N}(0, \|w^\star\|_2^2)$ since the $X_{i,j}$ are i.i.d. $\mathcal{N}(0,1)$ and independent in $j$; the rows are independent. The noise $\xi_i \sim \mathcal{N}(0, \sigma^2)$ is independent of $X$, hence

$$Y \;\sim\; \mathcal{N}(0, \tau_s^2 I_n), \qquad \frac{\|Y\|_2^2}{\tau_s^2} \;\sim\; \chi_n^2.$$

The standard Laurent–Massart inequality for $\chi_n^2$ variables (see e.g. *Ann. Statist.* 2000) yields, for all $t > 0$,

$$\Pr\Big(\|Y\|_2^2 - \tau_s^2 n \geq 2\tau_s^2\sqrt{nt} + 2\tau_s^2 t\Big) \leq e^{-t}, \qquad \Pr\Big(\tau_s^2 n - \|Y\|_2^2 \geq 2\tau_s^2\sqrt{nt}\Big) \leq e^{-t}.$$

Taking $t = cn$ gives $\|Y\|_2^2 = \tau_s^2 n(1 + o(1))$ with probability at least $1 - e^{-cn}$. $\qquad\square$

**Lemma A.6** (bulk coordinates of $X^\top Y$). *Conditional on $Y$, for each $j \notin S$,*

$$\langle X_{:,j}, Y \rangle \;\sim\; \mathcal{N}\big(0, \|Y\|_2^2\big),$$

*and the variables $\{\langle X_{:,j}, Y \rangle\}_{j \notin S}$ are i.i.d. given $Y$. Consequently, with $m_q := \mathbb{E}|Z|^q$ for $Z \sim \mathcal{N}(0,1)$,*

$$\sum_{j \notin S} \big|\langle X_{:,j}, Y \rangle\big|^q = (d - s)\, m_q\, \|Y\|_2^q\, \big(1 + o(1)\big) \;\asymp\; (d - s)\, \tau_s^q\, n^{q/2} \qquad w.h.p.$$

*Quantitatively, for any fixed $q \geq 2$ and any $u \in (0, 1)$,*

$$\Pr\left(\left|\frac{1}{d - s} \sum_{j \notin S} \frac{|\langle X_{:,j}, Y \rangle|^q}{\|Y\|_2^q} - m_q\right| > u \,\bigg|\, Y\right) \;\leq\; 2\exp\big(-c_q(d - s)\min\{u^2, u\}\big).$$

*Proof.* Fix $j \notin S$. The vector $X_{:,j} \sim \mathcal{N}(0, I_n)$ is independent of $(X_{:,k})_{k \in S}$ and $\xi$, hence independent of $Y = Xw^\star + \xi$, which depends only on the columns indexed by $S$ and on $\xi$. Conditional on $Y$, by rotational invariance,

$$\langle X_{:,j}, Y \rangle \;\overset{d}{=}\; \|Y\|_2\, Z_j, \qquad Z_j \sim \mathcal{N}(0, 1),$$

and independence across $j \notin S$ follows from the independence of the columns $\{X_{:,j}\}_{j \notin S}$.

Let $W_j := |Z_j|^q - m_q$. Then $W_j$ are i.i.d. mean-zero and sub-exponential with $\|W_j\|_{\psi_1} \leq C_q$ (a standard fact for polynomial functions of a standard Gaussian, see, e.g., Vershynin's *High-Dimensional Probability*). Bernstein's inequality for sub-exponential variables gives, for any $u > 0$,

$$\Pr\left(\left|\frac{1}{d - s} \sum_{j \notin S} W_j\right| > u \,\bigg|\, Y\right) \;\leq\; 2\exp\big(-c_q(d - s)\min\{u^2, u\}\big).$$

Multiplying back by $\|Y\|_2^q$ proves the conditional concentration display. Since $(d - s) \asymp n$ by assumption, taking $u \to 0$ slowly (e.g. $u = \sqrt{(\log n)/(d - s)}$) yields

$$\sum_{j \notin S} |\langle X_{:,j}, Y \rangle|^q = (d - s)m_q\|Y\|_2^q\,(1 + o(1))$$

with probability at least $1 - Ce^{-c(d-s)} \geq 1 - Ce^{-cn}$ (unconditionally). Finally, Lemma A.5 implies $\|Y\|_2^q \asymp \tau_s^q n^{q/2}$ w.h.p., completing the proof. $\qquad\square$

**Lemma A.7** (Signal block with integrated uniform column-norm control). *Let $X \in \mathbb{R}^{n \times d}$ have i.i.d. $\mathcal{N}(0,1)$ entries, $S \subset [d]$ with $|S| = s$, and $Y := Xw^{\star} + \xi$ where $\xi \sim \mathcal{N}(0, \sigma^2 I_n)$ is independent of $X$. Write $\tau_s^2 := \|w^{\star}\|_2^2 + \sigma^2$ and $W_q := \sum_{j \in S} |w_j^{\star}|^q$ for $q \geq 2$.*

*(i) Uniform column-norm concentration (over all $d$ columns). There exists a universal $c \in (0,1)$ such that, for every $u > 0$,*

$$\Pr\left(\max_{1 \leq j \leq d} \left| \frac{\|X_{:,j}\|_2^2}{n} - 1 \right| > u\right) \leq 2d \exp\left(-cn\min\{u^2, u\}\right). \tag{14}$$

*In particular, for any fixed $\gamma > 0$,*

$$u_n := \sqrt{\frac{(1+\gamma)\log d}{cn}} \in (0,1] \text{ for } n \text{ large, and } \Pr\left(\max_{j \leq d}\left|\frac{\|X_{:,j}\|_2^2}{n} - 1\right| > u_n\right) \leq 2d^{-\gamma}.$$

*(ii) Spike decomposition, explicit definition of $\zeta_j$, and $q$-moment bound. For each $j \in S$, define*

$$\zeta_j := \left\langle X_{:,j}, \sum_{k \in S \setminus \{j\}} w_k^{\star} X_{:,k} + \xi \right\rangle. \tag{15}$$

*Then*

$$\langle X_{:,j}, Y \rangle = w_j^{\star} \|X_{:,j}\|_2^2 + \zeta_j. \tag{16}$$

*Moreover, for each fixed $j \in S$,*

$$\mathbb{E}[\zeta_j \mid X_{:,j}] = 0, \qquad \mathrm{Var}(\zeta_j \mid X_{:,j}) = \left(\tau_s^2 - (w_j^{\star})^2\right)\|X_{:,j}\|_2^2, \tag{17}$$

*and, conditional on $X_{:,j}$,*

$$\zeta_j \sim \mathcal{N}\left(0, \left(\tau_s^2 - (w_j^{\star})^2\right)\|X_{:,j}\|_2^2\right). \tag{18}$$

(We do not assume or use independence between the collection $\{\zeta_j\}_{j \in S}$; the proof below controls their aggregate via operator-norm bounds.) *Consequently, with probability at least $1 - 2d^{-\gamma} - Ce^{-c\sqrt{ns}}$,*

$$\sum_{j \in S} \left|\langle X_{:,j}, Y \rangle\right|^q = n^q W_q (1 + o(1)) + O\left(\tau_s^q\left(sn^{q/2} + s^{1+q/2}\right)\right), \tag{19}$$

*where the $o(1)$ (as $n \to \infty$) and the hidden constants depend only on $q$ (hence on $p$). The mixed term $\Sigma_{j \in S}|a_j|^{q-1}|b_j|$ is absorbed by Young's inequality into the $n^q W_q$ leading term and the $\Sigma_{j \in S}|b_j|^q$ remainder, with a harmless change in constants.*

*Proof. Part (i):* For a fixed $j$, $Z_j := \|X_{:,j}\|_2^2 \stackrel{d}{=} \chi_n^2$. By Laurent–Massart, for all $x \geq 0$,

$$\Pr(Z_j - n \geq 2\sqrt{nx} + 2x) \leq e^{-x}, \qquad \Pr(n - Z_j \geq 2\sqrt{nx}) \leq e^{-x}.$$

A standard choice of $x$ (see derivation below) yields the Bernstein-type bound

$$\Pr\left(\left|\frac{Z_j}{n} - 1\right| > u\right) \leq 2\exp\left(-cn\min\{u^2, u\}\right) \qquad (\forall u > 0), \tag{20}$$

for some universal $c \in (0,1)$. Summing over $j = 1, \ldots, d$ gives equation 14. For the explicit choice $u_n = \sqrt{(1+\gamma)\log d/(cn)} \leq 1$ (for $n$ large),

$$2d\exp(-cnu_n^2) = 2d\exp(-(1+\gamma)\log d) = 2d^{-\gamma}.$$

*(Derivation of the Bernstein form):* If $u \in (0,1]$, choose $x = \frac{u^2 n}{8}$ to get $\Pr(Z_j - n \geq un) \leq e^{-\frac{u^2 n}{8}}$ and $x = \frac{u^2 n}{4}$ to get $\Pr(n - Z_j \geq un) \leq e^{-\frac{u^2 n}{4}}$. If $u \geq 1$, choose $x = c_0 un$ (e.g. $c_0 = 1/16$) so that $2\sqrt{nx} + 2x \leq un$, hence $\Pr(Z_j - n \geq un) \leq e^{-c_0 un}$. Combine and absorb constants into $c$.

*Part (ii):* The decomposition equation 16 is immediate from

$$Y = w_j^{\star} X_{:,j} + \sum_{k \in S \setminus \{j\}} w_k^{\star} X_{:,k} + \xi,$$

and independence/rotational invariance: conditional on $X_{:,j}$, $\langle X_{:,j}, X_{:,k}\rangle \sim \mathcal{N}(0, \|X_{:,j}\|_2^2)$ for $k \neq j$ and $\langle X_{:,j}, \xi\rangle \sim \mathcal{N}(0, \sigma^2 \|X_{:,j}\|_2^2)$, all independent. Let $a_j := w_j^\star \|X_{:,j}\|_2^2$ and $b_j := \zeta_j$ so that $\langle X_{:,j}, Y\rangle = a_j + b_j$. We show:

$$\sum_{j \in S} |a_j|^q = n^q W_q (1 + o(1)) \qquad \text{and} \qquad \sum_{j \in S} |b_j|^q \lesssim s\, \tau_s^q\, n^{q/2},$$

with the stated probability. Conditioned on the event from (i) with $u = u_n = o(1)$,

$$\max_{1 \leq j \leq d} \left| \frac{\|X_{:,j}\|_2^2}{n} - 1 \right| \leq u_n,$$

and by a mean-value bound, $\|X_{:,j}\|_2^{2q} = n^q(1 + O(u_n))$ uniformly in $j$. Hence

$$\sum_{j \in S} |a_j|^q = \sum_{j \in S} |w_j^\star|^q \|X_{:,j}\|_2^{2q} = n^q \sum_{j \in S} |w_j^\star|^q (1 + O(u_n)) = n^q W_q (1 + o(1)).$$

For any index set $T \subset [d]$, we write $X_{:,T} \in \mathbb{R}^{n \times |T|}$ for the submatrix formed by the columns $\{X_{:,j} : j \in T\}$. When convenient we abbreviate $X_{:,T}$ as $X_T$. For a vector $w \in \mathbb{R}^d$, $w_T$ denotes its restriction to $T$, and $T^c$ the complement of $T$ in $[d]$. Let $G := X_S^\top X_S$ and $D := \operatorname{diag}(\|X_{:,j}\|_2^2)_{j \in S}$. Then

$$b = (b_j)_{j \in S} = (G - D)\, w_S^\star + X_S^\top \xi.$$

We bound $\|b\|_2$ and then pass to $\ell_q$. Recall $b = (G - D)w_S^\star + X_S^\top \xi$, where $G := X_S^\top X_S \in \mathbb{R}^{s \times s}$ and $D := \operatorname{diag}(\|X_{:,j}\|_2^2)_{j \in S}$.

**Bound on** $\|(G - D)w_S^\star\|_2$**.** We have

$$\|(G - D)w_S^\star\|_2 \leq \|G - D\|_{\mathrm{op}} \|w^\star\|_2 \leq \left( \|G - nI_s\|_{\mathrm{op}} + \|D - nI_s\|_{\mathrm{op}} \right) \|w^\star\|_2. \tag{21}$$

*Singular-value bound for $G - nI_s$.* Let $s_{\max}(X_S)$ and $s_{\min}(X_S)$ denote the largest and smallest singular values of $X_S$. By the standard Gaussian singular-value concentration (see Vershynin, *High-Dimensional Probability*, Thm. 4.6.1), for any $t \geq 0$,

$$\mathbb{P}\Big( s_{\max}(X_S) \leq \sqrt{n} + \sqrt{s} + t,\ \ s_{\min}(X_S) \geq \sqrt{n} - \sqrt{s} - t \Big) \geq 1 - 2e^{-t^2/2}. \tag{22}$$

Conditioned on this event,

$$\begin{aligned} \|G - nI_s\|_{\mathrm{op}} &= \max \left\{ s_{\max}(X_S)^2 - n,\ n - s_{\min}(X_S)^2 \right\} \\ &\leq \left( \sqrt{n} + \sqrt{s} + t \right)^2 - n\ \vee\ n - \left( \sqrt{n} - \sqrt{s} - t \right)^2 \\ &\leq s + 2\sqrt{ns} + 2t(\sqrt{n} + \sqrt{s}) + t^2. \end{aligned} \tag{23}$$

Choosing $t = \sqrt{s}$ in equation 22–equation 23 yields, with probability at least $1 - 2e^{-s/2}$,

$$\|G - nI_s\|_{\mathrm{op}} \leq s + 2\sqrt{ns} + 2\sqrt{s}(\sqrt{n} + \sqrt{s}) + s \leq 4\sqrt{ns} + 4s. \tag{24}$$

*Diagonal bound for $D - nI_s$.* By the single-column deviation bound equation 20, for any $u > 0$ and any $j \in S$,

$$\Pr\left( \left| \frac{\|X_{:,j}\|_2^2}{n} - 1 \right| > u \right) \leq 2\exp\left( -c\, n\, \min\{u^2, u\} \right).$$

Union-bounding this over the $s$ indices $j \in S$ and taking

$$u_S := \sqrt{\frac{s}{n}}, \tag{25}$$

we obtain

$$\mathbb{P}\left( \max_{j \in S} \left| \frac{\|X_{:,j}\|_2^2}{n} - 1 \right| > u_S \right) \leq \begin{cases} C\, e^{-c\, s}, & s \leq n, \\ C\, e^{-c'\sqrt{ns}}, & s > n. \end{cases} \tag{26}$$

hence, on this event,

$$\|D - nI_s\|_{\mathrm{op}} = \max_{j \in S} \big|\|X_{:,j}\|_2^2 - n\big| \leq nu_S = \sqrt{ns}. \tag{27}$$

Combining equation 21, equation 24, and equation 27, we arrive at

$$\|(G - D)w_S^\star\|_2 \leq \big(4\sqrt{ns} + 4s + \sqrt{ns}\big)\|w^\star\|_2 \leq \big(5\sqrt{ns} + 4s\big)\|w^\star\|_2, \tag{28}$$

with probability at least $1 - 2e^{-s/2} - Ce^{-c'\sqrt{ns}}$.

Now we bound $\|X_S^\top \xi\|_2$. Conditionally on $X_S$, the vector $X_S^\top \xi$ is Gaussian with covariance

$$\Sigma := \mathrm{Var}\big(X_S^\top \xi \,\big|\, X_S\big) = \sigma^2 G.$$

Write the eigenvalues of $G$ as $\mu_1, \ldots, \mu_s \geq 0$. Then

$$\|X_S^\top \xi\|_2^2 \overset{d}{=} \sum_{i=1}^{s} \lambda_i Z_i^2, \qquad \lambda_i := \sigma^2 \mu_i, \ \ Z_i \overset{\mathrm{i.i.d.}}{\sim} \mathcal{N}(0, 1).$$

The weighted $\chi^2$ tail of Laurent–Massart (2000, Lemma 1) states that for all $x \geq 0$,

$$\mathbb{P}\left(\sum_{i=1}^{s} \lambda_i Z_i^2 \geq \sum_{i=1}^{s} \lambda_i + 2\sqrt{\left(\sum_{i=1}^{s} \lambda_i^2\right)x} + 2\big(\max_i \lambda_i\big)x \,\bigg|\, X_S\right) \leq e^{-x}. \tag{29}$$

Since $\sum_i \lambda_i = \sigma^2 \mathrm{tr}(G)$, $\sum_i \lambda_i^2 = \sigma^4 \mathrm{tr}(G^2) \leq \sigma^4 s\|G\|_{\mathrm{op}}^2$, and $\max_i \lambda_i = \sigma^2\|G\|_{\mathrm{op}}$, inserting these into equation 29 and choosing $x = s$ gives, with conditional probability $\geq 1 - e^{-s}$,

$$\|X_S^\top \xi\|_2^2 \leq \sigma^2\Big(\mathrm{tr}(G) + 4s\|G\|_{\mathrm{op}}\Big). \tag{30}$$

We now bound $\mathrm{tr}(G)$ and $\|G\|_{\mathrm{op}}$ on the events already used in Step A. First, by equation 25–equation 26,

$$\mathrm{tr}(G) = \sum_{j \in S} \|X_{:,j}\|_2^2 \leq sn(1 + u_S) = sn + s\sqrt{ns}. \tag{31}$$

Second, from equation 22 with $t = \sqrt{s}$,

$$\|G\|_{\mathrm{op}} = s_{\max}(X_S)^2 \leq \big(\sqrt{n} + \sqrt{s} + \sqrt{s}\big)^2 \leq n + 4\sqrt{ns} + 4s. \tag{32}$$

Plugging equation 31–equation 32 into equation 30 and taking square roots, we obtain

$$\begin{aligned}
\|X_S^\top \xi\|_2 &\leq \sigma\sqrt{sn + s\sqrt{ns} + 4s(n + 4\sqrt{ns} + 4s)} \\
&\leq \sigma\Big(\sqrt{sn} + \sqrt{s\sqrt{ns}} + 2\sqrt{sn} + 4s\Big) \\
&\leq C\sigma\big(\sqrt{sn} + s\big),
\end{aligned} \tag{33}$$

where in the last step we used $\sqrt{s\sqrt{ns}} = s^{3/4}n^{1/4} \leq \frac{1}{2}(\sqrt{sn} + s)$.

$\ell_2$ and $\ell_q$ bounds for $b$. Combining equation 28 and equation 33,

$$\|b\|_2 \leq \|(G - D)w_S^\star\|_2 + \|X_S^\top \xi\|_2 \leq C\Big(\sqrt{ns}\|w^\star\|_2 + s\|w^\star\|_2 + \sigma\sqrt{sn} + \sigma s\Big). \tag{34}$$

In particular, when $s \leq n$ the $s$ terms are dominated by $\sqrt{ns}$ and

$$\|b\|_2 \leq C\tau_s\sqrt{sn} \qquad (\text{since } \tau_s^2 = \|w^\star\|_2^2 + \sigma^2). \tag{35}$$

*(Refined q-moment bound via decoupling).* Introduce i.i.d. "ghost" columns $\{X'_{:,j}\}_{j \in S}$ independent of $(X, \xi)$ and set

$$\zeta'_j := \langle X'_{:,j}, u_j \rangle, \qquad u_j := X_{:,S\setminus\{j\}} w_{S\setminus\{j\}}^\star + \xi.$$

By a standard decoupling inequality for Gaussian chaos of order two (de la Peña and Giné, *Decoupling: From Dependence to Independence*, 1999, Thm. 3.5.3), there exists $C_q < \infty$ (depending only on $q$) such that for all $t > 0$,

$$\mathbb{P}\Big(\sum_{j \in S} |\zeta_j|^q > t\Big) \leq C_q \, \mathbb{P}\Big(\sum_{j \in S} |\zeta_j'|^q > t/C_q\Big).$$

Conditional on $\{u_j\}$, the variables $\{\zeta_j'\}_{j \in S}$ are independent centered Gaussians with variances $\|u_j\|_2^2$. On the singular-value and column-norm events used above (cf. equation 22 with $t = \sqrt{s}$ and equation 14), uniformly in $j$,

$$\|u_j\|_2^2 \leq \|X_{:,S}\|_{\mathrm{op}}^2 \|w^\star\|_2^2 + \|\xi\|_2^2 \leq C\big(n + 4\sqrt{ns} + 4s\big)\|w^\star\|_2^2 + C\sigma^2 n \leq C\,\tau_s^2\,(n+s).$$

Hence, conditionally on $\{u_j\}$, each $|\zeta_j'|^q$ is sub-exponential with $\psi_1$-norm $\leq C\,\tau_s^q\,(n+s)^{q/2}$. Bernstein's inequality then yields

$$\sum_{j \in S} |\zeta_j'|^q \leq C\,\tau_s^q\Big(s\,n^{q/2} + s^{1+q/2}\Big) \qquad \text{with conditional probability at least } 1 - Ce^{-cs}.$$

Unconditioning and applying decoupling gives, with probability at least $1 - 2d^{-\gamma} - Ce^{-cs}$,

$$\sum_{j \in S} |b_j|^q \;=\; \sum_{j \in S} |\zeta_j|^q \;\leq\; C\,\tau_s^q\Big(s\,n^{q/2} + s^{1+q/2}\Big). \tag{36}$$

In particular, if $s \leq n$ this simplifies to $\sum_{j \in S} |b_j|^q \leq C\,s\,\tau_s^q\,n^{q/2}$.

For the cross term, for $q \geq 2$ and any $a, b \in \mathbb{R}$ we have the elementary inequality

$$\big||a+b|^q - |a|^q\big| \;\leq\; C_q\big(|a|^{q-1}|b| + |b|^q\big), \tag{37}$$

for a constant $C_q$ depending only on $q$. Summing equation 37 over $j \in S$ with $a_j = w_j^\star \|X_{:,j}\|_2^2$ and $b_j = \zeta_j$, and applying Hölder,

$$\sum_{j \in S}\big||a_j+b_j|^q - |a_j|^q\big| \leq C_q \sum_{j \in S} |a_j|^{q-1}|b_j| + C_q \sum_{j \in S} |b_j|^q$$

$$\leq C_q\Big(\sum_{j \in S} |a_j|^q\Big)^{\frac{q-1}{q}}\Big(\sum_{j \in S} |b_j|^q\Big)^{\frac{1}{q}} + C_q \sum_{j \in S} |b_j|^q. \tag{38}$$

Set

$$A := \sum_{j \in S} |a_j|^q, \qquad B := \sum_{j \in S} |b_j|^q.$$

Apply Young's inequality with conjugate exponents $r = \frac{q}{q-1}$ and $s = q$: for any $\varepsilon > 0$,

$$A^{\frac{q-1}{q}} B^{\frac{1}{q}} \;\leq\; \frac{\varepsilon}{r}\,A + \frac{\varepsilon^{-(q-1)}}{s}\,B \;=\; \frac{q-1}{q}\,\varepsilon A + \frac{1}{q}\,\varepsilon^{-(q-1)}\,B. \tag{39}$$

With $A = n^q W_q(1+O(u_n))$ and the bound $B \leq C\,\tau_s^q(s\,n^{q/2} + s^{1+q/2})$ from equation 36, choosing a fixed $\varepsilon \in (0,1)$ (e.g. $\varepsilon = \frac{1}{2}$) absorbs the mixed term into the leading $A$ and the $B$-remainder (with a harmless change of constants). Consequently,

$$\sum_{j \in S}\big||a_j+b_j|^q - |a_j|^q\big| = O\Big(\tau_s^q\big(s\,n^{q/2} + s^{1+q/2}\big)\Big),$$

which yields equation 19. When $s \leq n$ the remainder simplifies to $O\big(s\,\tau_s^q\,n^{q/2}\big)$. $\qquad\square$

Combining Lemmas A.6–A.7 yields the decomposition

$$\|X^\top Y\|_q^q \;=\; n^q\,W_q\,(1+o(1)) + (d-s)\,m_q\,\tau_s^q\,n^{q/2}\,(1+o(1)) + O\Big(\tau_s^q\big(s\,n^{q/2}+s^{1+q/2}\big)\Big) \qquad \text{w.h.p.} \tag{40}$$

A.2.3 Bulk $\ell_q$-embedding and Gaussian $\ell_t$ relations.

**Lemma A.8** (uniform $\ell_q$ control on the bulk operator). *Let $q \in [2, \infty)$ and assume $\kappa_{\mathrm{bulk}} := \liminf_{n \to \infty} \frac{d-s}{n} > 0$. There exist constants $0 < c_q \le C_q < \infty$, depending only on $(q, \kappa_{\mathrm{bulk}})$, such that, with probability at least $1 - Ce^{-cn}$, simultaneously for all $\lambda \in \mathbb{R}^n$,*

$$c_q (d-s) \|\lambda\|_2^q \le \sum_{j \notin S} |\langle X_{:,j}, \lambda \rangle|^q \le C_q (d-s) \|\lambda\|_2^q. \tag{41}$$

(Here we absorb the Gaussian absolute moment $m_q = \mathbb{E}|Z|^q$ into the constants $c_q, C_q$; in equation 42 we keep $m_t$ explicit.) *Moreover, for every $t \in [1, q]$, there exist constants $0 < c_t \le C_t < \infty$, depending only on $(t, \kappa_{\mathrm{bulk}})$, such that, w.h.p., uniformly in $\lambda \in \mathbb{R}^n$,*

$$c_t^{1/t} (d-s)^{1/t} m_t^{1/t} \|\lambda\|_2 \le \left\| \left( |\langle X_{:,j}, \lambda \rangle| \right)_{j \notin S} \right\|_t \le C_t^{1/t} (d-s)^{1/t} m_t^{1/t} \|\lambda\|_2, \tag{42}$$

*where $m_t := \mathbb{E}|Z|^t$ for $Z \sim \mathcal{N}(0,1)$.*

*Proof.* Fix $\lambda \in \mathbb{R}^n$, and if $\lambda \ne 0$ write $u := \lambda/\|\lambda\|_2 \in \mathbb{S}^{n-1}$. By homogeneity,

$$\sum_{j \notin S} |\langle X_{:,j}, \lambda \rangle|^q = \|\lambda\|_2^q \sum_{j \notin S} |\langle X_{:,j}, u \rangle|^q, \tag{43}$$

and similarly for any $t \in [1, q]$,

$$\left\| (|\langle X_{:,j}, \lambda \rangle|)_{j \notin S} \right\|_t = \|\lambda\|_2 \left( \sum_{j \notin S} |\langle X_{:,j}, u \rangle|^t \right)^{1/t}. \tag{44}$$

Thus it suffices to prove the bounds for unit $u$.

Let $T := S^c$ and $m := |T| = d - s$. Fix $u \in \mathbb{S}^{n-1}$ and $t \in [1, q]$. Define

$$Y_j^{(t)}(u) := \left| \langle X_{:,j}, u \rangle \right|^t, \qquad j \in T.$$

Since the columns $\{X_{:,j}\}_{j \in T}$ are i.i.d. $\mathcal{N}(0, I_n)$ and independent of $u$, the random variables $\{Y_j^{(t)}(u)\}_{j \in T}$ are i.i.d.

**Definition A.9** (Orlicz $\psi_\nu$ norm and sub-Weibull class). For $\nu \in (0, 2]$ and a real random variable $Z$, the Orlicz norm

$$\|Z\|_{\psi_\nu} := \inf \left\{ K > 0 : \mathbb{E} \exp\left( \frac{|Z|^\nu}{K^\nu} \right) \le 2 \right\}.$$

If $\|Z\|_{\psi_\nu} < \infty$, we say $Z$ is *sub-Weibull of order $\nu$*. Special cases: $\nu = 2$ (sub-Gaussian) and $\nu = 1$ (sub-Exponential). Two basic properties we use are

$$\mathbb{P}(|Z| > x) \le 2 \exp\left( -c (x/\|Z\|_{\psi_\nu})^\nu \right) \quad (\forall x \ge 0), \tag{45}$$

$$\|Z - \mathbb{E}Z\|_{\psi_\nu} \le 2 \|Z\|_{\psi_\nu}. \tag{46}$$

**Definition A.10** (Gaussian absolute moment). For $t > 0$, let $Z \sim \mathcal{N}(0, 1)$ and define

$$m_t := \mathbb{E}|Z|^t = 2^{t/2} \frac{\Gamma\left( \frac{t+1}{2} \right)}{\sqrt{\pi}}.$$

*Classification of $Y_j^{(t)}(u)$ in $\psi_\nu$ (with explicit mgf computation).* Since $\langle X_{:,j}, u \rangle \sim \mathcal{N}(0,1)$, write $Z \sim \mathcal{N}(0,1)$ and set $W := |Z|^t$. For any $K > 0$,

$$\left( \frac{W}{K} \right)^{2/t} = \left( \frac{|Z|^t}{K} \right)^{2/t} = \frac{|Z|^2}{K^{2/t}}.$$

Let

$$\theta := \frac{1}{K^{2/t}}.$$

Then

$$\mathbb{E}\exp\left((W/K)^{2/t}\right) = \mathbb{E}\exp\left(\theta\,Z^2\right).$$

Compute this expectation explicitly: using the standard normal density $\varphi(z) = (2\pi)^{-1/2}e^{-z^2/2}$,

$$
\begin{aligned}
\mathbb{E}\left[e^{\theta Z^2}\right] &= \int_{\mathbb{R}} e^{\theta z^2}\,\varphi(z)\,dz = \frac{1}{\sqrt{2\pi}}\int_{\mathbb{R}} e^{\theta z^2}\,e^{-z^2/2}\,dz \\
&= \frac{1}{\sqrt{2\pi}}\int_{\mathbb{R}} e^{-(\frac{1}{2}-\theta)\,z^2}\,dz \;=\; \frac{1}{\sqrt{2\pi}}\cdot\sqrt{\frac{\pi}{\frac{1}{2}-\theta}} \;=\; \frac{1}{\sqrt{1-2\theta}}, \qquad \text{for } \theta < \tfrac{1}{2}. \quad (47)
\end{aligned}
$$

Equivalently, since $Z^2 \sim \chi_1^2$, the mgf of $\chi_1^2$ is $(1-2\theta)^{-1/2}$ for $\theta < 1/2$, which matches equation 47. We now choose $K$ so that $\theta < 1/2$ and the expectation is uniformly bounded by a constant $\leq 2$. Take

$$K_t := (4t)^{t/2} \quad \Longrightarrow \quad \theta = \frac{1}{K_t^{2/t}} = \frac{1}{4t} \leq \frac{1}{4} \qquad (t \geq 1). \quad (48)$$

Then, by equation 47,

$$\mathbb{E}\exp\left((W/K_t)^{2/t}\right) = \frac{1}{\sqrt{1-\frac{2}{K_t^{2/t}}}} = \frac{1}{\sqrt{1-\frac{1}{2t}}} \leq \frac{1}{\sqrt{1-\frac{1}{2}}} = \sqrt{2} \;<\; 2, \quad (49)$$

where we used $t \in [1, q]$ (hence $t \geq 1$). By the definition of the Orlicz norm,

$$\big\| \, |Z|^t \, \big\|_{\psi_{2/t}} \;\leq\; K_t \;=\; (4t)^{t/2}. \quad (50)$$

Centering preserves the class up to a factor 2 (by equation 46), hence

$$\big\| \, |Z|^t - m_t \, \big\|_{\psi_{2/t}} \;\leq\; 2K_t \;=\; 2(4t)^{t/2}. \quad (51)$$

Finally, define

$$\nu(t) \;:=\; \min\{1,\, 2/t\}. \quad (52)$$

Since $2/t \geq 1$ for $t \leq 2$ and $2/t < 1$ for $t > 2$, combining equation 51 with equation 52 yields the uniform (in $u$) classification

$$\big\| \, Y_j^{(t)}(u) - m_t \, \big\|_{\psi_{\nu(t)}} \;\leq\; K_t' \qquad \text{with } K_t' := 2(4t)^{t/2}. \quad (53)$$

This bound is uniform in $u$ because $\langle X_{:,j}, u\rangle \overset{d}{=} \mathcal{N}(0,1)$ for every fixed $u \in \mathbb{S}^{n-1}$.

*Empirical-mean concentration at fixed $u$.* From equation 53 and independence across $j \in T$, a Bernstein-type inequality for sums of i.i.d. sub-Weibull($\nu$) variables (e.g. Theorem 3.1 in Kuchibhotla–Chakrabortty, 2018) yields, for any $\varepsilon > 0$,

$$\mathbb{P}\left(\left|\frac{1}{m}\sum_{j\in T}\left(Y_j^{(t)}(u) - m_t\right)\right| > \varepsilon\right) \;\leq\; 2\exp\left\{-c_{\nu(t)}\,m\,\min\left(\frac{\varepsilon^2}{K_t'^2},\, \left(\frac{\varepsilon}{K_t'}\right)^{\nu(t)}\right)\right\}. \quad (54)$$

Taking $\varepsilon = \delta\,m_t$ with $\delta \in (0,1)$, and absorbing the fixed ratio $m_t/K_t'$ (which depends only on $t$) into the constant, we obtain

$$\mathbb{P}\left(\left|\frac{1}{m}\sum_{j\in T}Y_j^{(t)}(u) - m_t\right| > \delta\,m_t\right) \;\leq\; 2\exp\left(-c_t\,m\,\min\{\delta^2,\, \delta^{\nu(t)}\}\right), \quad (55)$$

where $c_t > 0$ depends only on $t$ (hence only on $p$). In the sub-Exponential range $t \in [1,2]$, $\nu(t) = 1$ and equation 55 simplifies to

$$\mathbb{P}\left(\left|\frac{1}{m}\sum_{j\in T}Y_j^{(t)}(u) - m_t\right| > \delta\,m_t\right) \;\leq\; 2\exp\left(-c_t\,m\,\min\{\delta^2,\delta\}\right). \quad (56)$$

Finally, note that

$$\mathbb{E}Y_j^{(t)}(u) \;=\; m_t, \quad (57)$$

by Definition A.10, completing Step 1.

Now we can construct a net on the sphere and a uniform bound on that net. Let $\varepsilon \in (0, 1/8]$ be a fixed absolute constant (to be chosen below). There exists an $\varepsilon$-net $\mathcal{N}_\varepsilon \subset \mathbb{S}^{n-1}$ with

$$|\mathcal{N}_\varepsilon| \ \leq \ \left(1 + \frac{2}{\varepsilon}\right)^n \ \leq \ C_\varepsilon^n. \tag{58}$$

Applying equation 55 with $\delta = \delta_t \in (0, 1/4]$ (a small absolute constant depending only on $t$) and union-bounding over $\mathcal{N}_\varepsilon$ yields

$$\mathbb{P}\left(\exists\, v \in \mathcal{N}_\varepsilon : \ \left|\frac{1}{m}\sum_{j \in T} Y_j^{(t)}(v) - m_t\right| > \delta_t\, m_t\right) \leq 2\,|\mathcal{N}_\varepsilon|\,\exp\big(-c_t\, m\,\min\{\delta_t^2, \delta_t\}\big)$$

$$\leq 2\exp\Big(n\log C_\varepsilon - c_t'm\Big). \tag{59}$$

Because $m \geq \kappa_{\mathrm{bulk}} n$ and $\kappa_{\mathrm{bulk}} > 0$, by taking $\delta_t$ fixed (e.g. $\delta_t = 1/4$) and $\varepsilon$ fixed (e.g. $\varepsilon = 1/8$), the right-hand side of equation 59 is $\leq Ce^{-cn}$. Therefore, with probability at least $1 - Ce^{-cn}$, simultaneously for all $v \in \mathcal{N}_\varepsilon$,

$$(1 - \delta_t)\, m_t \ \leq \ \frac{1}{m}\sum_{j \in T} |\langle X_{:,j}, v\rangle|^t \ \leq \ (1 + \delta_t)\, m_t. \tag{60}$$

We are ready to extend from the net to the whole sphere. Fix arbitrary $u \in \mathbb{S}^{n-1}$ and pick $v \in \mathcal{N}_\varepsilon$ with $\|u - v\|_2 \leq \varepsilon$. For any $a, b \in \mathbb{R}$ and any $t \geq 1$, the elementary inequalities

$$|a + b|^t \ \leq \ 2^{t-1}\big(|a|^t + |b|^t\big), \qquad |a|^t \ \leq \ 2^{t-1}\big(|a + b|^t + |b|^t\big) \tag{61}$$

hold. Applying equation 61 with $a = \langle X_{:,j}, v\rangle$ and $b = \langle X_{:,j}, u - v\rangle$, we get

$$|\langle X_{:,j}, u\rangle|^t \leq 2^{t-1}\Big(|\langle X_{:,j}, v\rangle|^t + |\langle X_{:,j}, u - v\rangle|^t\Big), \tag{62}$$

$$|\langle X_{:,j}, u\rangle|^t \geq 2^{1-t}\,|\langle X_{:,j}, v\rangle|^t - |\langle X_{:,j}, u - v\rangle|^t. \tag{63}$$

Average equation 62 and equation 63 over $j \in T$ and divide by $m$ to obtain

$$\frac{1}{m}\sum_{j \in T} |\langle X_{:,j}, u\rangle|^t \leq 2^{t-1}\left(\frac{1}{m}\sum_{j \in T} |\langle X_{:,j}, v\rangle|^t + \frac{1}{m}\sum_{j \in T} |\langle X_{:,j}, u - v\rangle|^t\right), \tag{64}$$

$$\frac{1}{m}\sum_{j \in T} |\langle X_{:,j}, u\rangle|^t \geq 2^{1-t}\frac{1}{m}\sum_{j \in T} |\langle X_{:,j}, v\rangle|^t - \frac{1}{m}\sum_{j \in T} |\langle X_{:,j}, u - v\rangle|^t. \tag{65}$$

For any $w \in \mathbb{R}^n$,

$$\frac{1}{m}\sum_{j \in T} |\langle X_{:,j}, w\rangle|^t = \|w\|_2^t \cdot \frac{1}{m}\sum_{j \in T} |\langle X_{:,j}, \widehat{w}\rangle|^t, \qquad \widehat{w} := \frac{w}{\|w\|_2} \ \text{(if } w \neq 0). \tag{66}$$

Define the functional and its extremal values

$$A(u) := \frac{1}{m}\sum_{j \in T} |\langle X_{:,j}, u\rangle|^t, \qquad S := \sup_{u \in \mathbb{S}^{n-1}} A(u), \qquad I := \inf_{u \in \mathbb{S}^{n-1}} A(u).$$

By equation 66 and $\|u - v\|_2 \leq \varepsilon$,

$$\frac{1}{m}\sum_{j \in T} |\langle X_{:,j}, u - v\rangle|^t = \|u - v\|_2^t \cdot \frac{1}{m}\sum_{j \in T} |\langle X_{:,j}, \widehat{u - v}\rangle|^t \leq \varepsilon^t\, S,$$

where we used the definition of $S$ in the last inequality. On the event equation 60 (from Step 2), $A(v) \in [(1 - \delta_t)m_t, (1 + \delta_t)m_t]$ for every $v \in \mathcal{N}_\varepsilon$. Plugging these into equation 64–equation 65 yields

$$A(u) \leq 2^{t-1}\Big(A(v) + \varepsilon^t S\Big),$$

$$A(u) \geq 2^{1-t} A(v) - \varepsilon^t S.$$

Taking the supremum over $u \in \mathbb{S}^{n-1}$ in the upper bound:

$$S \leq 2^{t-1}\big((1+\delta_t)m_t + \varepsilon^t S\big) \quad \Longrightarrow \quad S \leq \frac{2^{t-1}}{1 - 2^{t-1}\varepsilon^t}(1+\delta_t)m_t.$$

Taking the infimum over $u \in \mathbb{S}^{n-1}$ in the lower bound:

$$I \geq 2^{1-t}(1-\delta_t)m_t - \varepsilon^t S.$$

Choose fixed $\delta_t \leq \frac{1}{4}$ and $\varepsilon \leq \frac{1}{8}$; then

$$2^{t-1}\varepsilon^t = \frac{(2\varepsilon)^t}{2} \leq \frac{(1/4)^t}{2} \leq \frac{1}{8},$$

so $1 - 2^{t-1}\varepsilon^t \geq 7/8$ and thus

$$S \leq \frac{2^{t-1}}{7/8}(1+\delta_t)m_t \leq C_t\, m_t,$$

for a constant $C_t < \infty$ depending only on $t$. Substituting this bound for $S$ back into the inequality for $I$ gives

$$I \geq 2^{1-t}(1-\delta_t)m_t - \varepsilon^t C_t m_t \geq c_t\, m_t,$$

for some $c_t > 0$ (depending only on $t$). Therefore, with probability at least $1 - Ce^{-cn}$,

$$c_t\, m_t \leq \frac{1}{m}\sum_{j \in T} |\langle X_{:,j}, u\rangle|^t \leq C_t\, m_t \qquad \text{simultaneously for all } u \in \mathbb{S}^{n-1}. \tag{67}$$

Multiplying equation 67 by $m = d - s$ and using equation 43 with $t = q$ yields

$$c_q\,(d-s)\,\|\lambda\|_2^q \leq \sum_{j \notin S} |\langle X_{:,j}, \lambda\rangle|^q \leq C_q\,(d-s)\,\|\lambda\|_2^q,$$

which is equation 41. Likewise, combining equation 67 with equation 44 gives

$$c_t^{1/t}\,(d-s)^{1/t}\,m_t^{1/t}\,\|\lambda\|_2 \leq \big\|(|\langle X_{:,j}, \lambda\rangle|)_{j \notin S}\big\|_t \leq C_t^{1/t}\,(d-s)^{1/t}\,m_t^{1/t}\,\|\lambda\|_2,$$

which is equation 42. $\qquad\square$

### A.2.4 Spike $\ell_t$ control for $X^\top Y$

**Lemma A.11** (spike $\ell_t$ control for $X^\top Y$). *Fix any $t \in [1, q]$ and $\gamma > 0$. With probability at least $1 - 2d^{-\gamma} - Ce^{-cs}$,*

$$\left\|\big(|\langle X_{:,j}, Y\rangle|\big)_{j \in S}\right\|_t = n\,\|w^\star\|_t\,(1 + O(u_n)) \;\pm\; C\,\tau_s\Big(\sqrt{n}\,s^{\max\{1/t,\,1/2\}} + s^{1 + (1/t - 1/2)_+}\Big), \tag{68}$$

*where $u_n := \sqrt{(1+\gamma)\log d/(c\,n)} = o(1)$ and $(x)_+ := \max\{x, 0\}$. In particular, if $s \leq n$ then the error simplifies to*

$$\left\|\big(|\langle X_{:,j}, Y\rangle|\big)_{j \in S}\right\|_t = n\,\|w^\star\|_t\,(1 + O(u_n)) \;\pm\; C\,\tau_s\,\sqrt{n}\,s^{\max\{1/t,\,1/2\}}. \tag{69}$$

*All constants may depend on $t$ (hence on $p$) but not on $(n, d, s)$.*

*Proof.* For each $j \in S$,

$$\langle X_{:,j}, Y\rangle = w_j^\star\,\|X_{:,j}\|_2^2 + \zeta_j, \qquad \zeta_j := \Big\langle X_{:,j},\; \sum_{k \in S\backslash\{j\}} w_k^\star X_{:,k} + \xi\Big\rangle. \tag{70}$$

Conditional on $X_{:,j}$,

$$\mathbb{E}[\zeta_j \mid X_{:,j}] = 0, \qquad \mathrm{Var}(\zeta_j \mid X_{:,j}) = (\tau_s^2 - (w_j^\star)^2)\,\|X_{:,j}\|_2^2, \tag{71}$$

and $\zeta_j \mid X_{:,j} \sim \mathcal{N}(0, (\tau_s^2 - (w_j^\star)^2)\|X_{:,j}\|_2^2)$ by independence and rotational invariance. Define

$$a_j := w_j^\star\|X_{:,j}\|_2^2, \qquad b_j := \zeta_j, \qquad a := (a_j)_{j \in S},\; b := (b_j)_{j \in S}.$$

By the uniform column-norm bound equation 14 with $u = u_n = o(1)$, we have

$$\max_{1 \leq j \leq d} \left| \frac{\|X_{:,j}\|_2^2}{n} - 1 \right| \leq u_n \qquad \text{with probability at least } 1 - 2d^{-\gamma}. \tag{72}$$

On this event,

$$\left\| (|a_j|)_{j \in S} \right\|_{\ell_t} = \Big( \sum_{j \in S} |w_j^\star|^t \|X_{:,j}\|_2^{2t} \Big)^{1/t} = n \Big( \sum_{j \in S} |w_j^\star|^t (1 + O(u_n))^t \Big)^{1/t}$$

$$= n \|w^\star\|_t \left(1 + O(u_n)\right). \tag{73}$$

Let $X_S$ be the $n \times s$ submatrix with columns $\{X_{:,j}\}_{j \in S}$, and set

$$G := X_S^\top X_S, \qquad D := \mathrm{diag}\big(\|X_{:,j}\|_2^2\big)_{j \in S}.$$

From equation 70, in vector form

$$b = (G - D)\, w_S^\star \; + \; X_S^\top \xi. \tag{74}$$

We bound the two terms separately.

*(i) Control of $(G - D)w_S^\star$.* By the triangle inequality and operator norm submultiplicativity,

$$\|(G - D)w_S^\star\|_2 \; \leq \; \|G - D\|_{\mathrm{op}} \|w^\star\|_2 \; \leq \; \big(\|G - nI_s\|_{\mathrm{op}} + \|D - nI_s\|_{\mathrm{op}}\big) \|w^\star\|_2. \tag{75}$$

Gaussian singular-value concentration (Vershynin, HDP, Thm. 4.6.1) gives, for any $t \geq 0$,

$$\mathbb{P}\Big(s_{\max}(X_S) \leq \sqrt{n} + \sqrt{s} + t, \; s_{\min}(X_S) \geq \sqrt{n} - \sqrt{s} - t\Big) \; \geq \; 1 - 2e^{-t^2/2}. \tag{76}$$

On this event,

$$\|G - nI_s\|_{\mathrm{op}} = \max\big\{ s_{\max}(X_S)^2 - n, \; n - s_{\min}(X_S)^2 \big\}$$

$$\leq (\sqrt{n} + \sqrt{s} + t)^2 - n \;\; \vee \;\; n - (\sqrt{n} - \sqrt{s} - t)^2$$

$$\leq s + 2\sqrt{ns} + 2t(\sqrt{n} + \sqrt{s}) + t^2. \tag{77}$$

Taking $t = \sqrt{s}$ yields, with probability $\geq 1 - 2e^{-s/2}$,

$$\|G - nI_s\|_{\mathrm{op}} \; \leq \; 4\sqrt{ns} + 4s. \tag{78}$$

By the $S$-only column-norm event equation 26 (with $u_S = \sqrt{s/n}$),

$$\|D - nI_s\|_{\mathrm{op}} = \max_{j \in S} \big| \|X_{:,j}\|_2^2 - n \big| \; \leq \; nu_S \; = \; \sqrt{ns}.$$

Combining this with equation 75 and equation 78 yields

$$\|(G - D)w_S^\star\|_2 \; \leq \; C\,(\sqrt{ns} + s)\,\|w^\star\|_2 \qquad \text{with probability at least } 1 - 2e^{-s/2} - Ce^{-c\sqrt{ns}}. \tag{79}$$

*(ii) Control of $X_S^\top \xi$.* Conditionally on $X_S$, one has $X_S^\top \xi \sim \mathcal{N}(0, \sigma^2 G)$. Writing $\{\mu_i\}_{i=1}^s$ for the eigenvalues of $G$ and $\lambda_i := \sigma^2 \mu_i$, Laurent–Massart's weighted $\chi^2$ tail (2000, Lemma 1) yields, for all $x \geq 0$,

$$\mathbb{P}\left( \sum_{i=1}^s \lambda_i Z_i^2 \geq \sum_i \lambda_i + 2\sqrt{\Big(\sum_i \lambda_i^2\Big)x} + 2(\max_i \lambda_i)\, x \;\Big|\; X_S \right) \; \leq \; e^{-x}. \tag{80}$$

Using $\sum_i \lambda_i = \sigma^2 \mathrm{tr}(G)$, $\sum_i \lambda_i^2 \leq \sigma^4 s\, \|G\|_{\mathrm{op}}^2$, and $\max_i \lambda_i = \sigma^2 \|G\|_{\mathrm{op}}$, and taking $x = s$ gives, with conditional probability $\geq 1 - e^{-s}$,

$$\|X_S^\top \xi\|_2^2 \; \leq \; \sigma^2 \Big( \mathrm{tr}(G) + 4s\, \|G\|_{\mathrm{op}} \Big). \tag{81}$$

On the event equation 76 with $t = \sqrt{s}$ and equation 72,

$$\mathrm{tr}(G) = \sum_{j \in S} \|X_{:,j}\|_2^2 \; \leq \; sn(1 + u_n) = sn + o(sn), \quad \|G\|_{\mathrm{op}} = s_{\max}(X_S)^2 \; \leq \; n + 4\sqrt{ns} + 4s.$$

$$\tag{82}$$

Plugging equation 82 into equation 81 and taking square roots,

$$\|X_S^\top \xi\|_2 \;\leq\; C\,\sigma\big(\sqrt{sn}+s\big) \qquad \text{with prob.} \geq 1 - 2e^{-s/2} - e^{-s}. \tag{83}$$

Combining equation 79, equation 83, and equation 74,

$$\|b\|_2 \;\leq\; C\,\tau_s\big(\sqrt{sn}+s\big) \qquad \text{with prob.} \geq 1 - 2d^{-\gamma} - Ce^{-cs}. \tag{84}$$

For $t \in [1, 2]$, the norm monotonicity in $\mathbb{R}^s$ gives

$$\|b\|_{\ell_t} \;\leq\; s^{1/t-1/2}\,\|b\|_2. \tag{85}$$

For $t \geq 2$, $\|b\|_{\ell_t} \leq \|b\|_2$. Hence, for all $t \in [1, q]$,

$$\|b\|_{\ell_t} \;\leq\; s^{(1/t-1/2)_+}\,\|b\|_2 \;\leq\; C\,\tau_s\Big(\sqrt{n}\,s^{\max\{1/t,\,1/2\}} + s^{1+(1/t-1/2)_+}\Big), \tag{86}$$

where we used equation 84. In particular, if $s \leq n$ then $s^{1+(1/t-1/2)_+} \leq \sqrt{n}\,s^{\max\{1/t,\,1/2\}}$ and equation 86 reduces to

$$\|b\|_{\ell_t} \;\leq\; C\,\tau_s\,\sqrt{n}\,s^{\max\{1/t,\,1/2\}}. \tag{87}$$

Finally, by the triangle inequality,

$$\big\|(|a_j + b_j|)_{j \in S}\big\|_{\ell_t} \;\leq\; \big\|(|a_j|)_{j \in S}\big\|_{\ell_t} + \big\|(|b_j|)_{j \in S}\big\|_{\ell_t}, \tag{88}$$

$$\big\|(|a_j + b_j|)_{j \in S}\big\|_{\ell_t} \;\geq\; \big\|(|a_j|)_{j \in S}\big\|_{\ell_t} - \big\|(|b_j|)_{j \in S}\big\|_{\ell_t}, \tag{89}$$

and combining with equation 73 and equation 86 (or equation 87 when $s \leq n$) yields equation 68 (and equation 69). $\qquad\square$

### A.2.5 Ray controls: minimal comparison and blockwise bounds

For the ray $\lambda = tY$ we have the one-dimensional dual objective

$$D(t) := \langle Y, tY \rangle - \tfrac{1}{q}\|X^\top(tY)\|_q^q \;=\; t\,\|Y\|_2^2 \;-\; \frac{t^q}{q}\,\|X^\top Y\|_q^q. \tag{90}$$

Since $D''(t) = -(q-1)\,t^{q-2}\,\|X^\top Y\|_q^q < 0$ for all $t > 0$, $D$ is strictly concave on $[0, \infty)$ and admits a unique maximizer $t_\star$ given by the first-order condition $D'(t_\star) = 0$:

$$t_\star^{q-1} = \frac{\|Y\|_2^2}{\|X^\top Y\|_q^q}. \tag{91}$$

At this maximizer,

$$D(t_\star) = t_\star\|Y\|_2^2 - \frac{t_\star^q}{q}\|X^\top Y\|_q^q = \Big(1 - \frac{1}{q}\Big)\,t_\star^q\|X^\top Y\|_q^q = \Big(1 - \frac{1}{q}\Big)\,\|X^\top(t_\star Y)\|_q^q. \tag{92}$$

**Lemma A.12** (Ray controls). *Let $p \in (1, 2]$, $q = \frac{p}{p-1} \in [2, \infty)$, and define $t_\star$ by equation 91. With probability at least $1 - Ce^{-c(d-s)} - Ce^{-c\sqrt{ns}}$ (constants depend only on $(q, \kappa_{\mathrm{bulk}})$), the following hold simultaneously.*

(One-sided value comparison).

$$D(\lambda^\star) \;\geq\; D(t_\star) \qquad \text{and} \qquad \|X^\top\lambda^\star\|_q^q \;\geq\; \|X^\top(t_\star Y)\|_q^q. \tag{93}$$

(Dual-norm scale). *There exist $0 < c_1 \leq C_1 < \infty$ depending only on $(q, \kappa_{\mathrm{bulk}})$ such that*

$$c_1\,t_\star\,\|Y\|_2 \;\leq\; \|\lambda^\star\|_2 \;\leq\; C_1\,t_\star\,\|Y\|_2. \tag{94}$$

(Bulk block at level $t \in [1, q]$). *For each $t \in [1, q]$ there exist $0 < c_t \leq C_t < \infty$ (depending only on $(t, \kappa_{\mathrm{bulk}})$) such that*

$$c_t^{1/t}\,(d-s)^{1/t}m_t^{1/t}\,t_\star\|Y\|_2 \;\leq\; \Big\|\big(|\langle X_{:,j}, \lambda^\star\rangle|\big)_{j \notin S}\Big\|_t \;\leq\; C_t^{1/t}\,(d-s)^{1/t}m_t^{1/t}\,t_\star\|Y\|_2, \tag{95}$$

where $m_t = \mathbb{E}|Z|^t$ for $Z \sim \mathcal{N}(0, 1)$.

(Spike block: two-sided $t$-level perturbation). *For every $t \in [1, q]$,*

$$\left\|\left(|\langle X_{:,j}, \lambda^\star\rangle|\right)_{j\in S}\right\|_t = t_\star \left\|\left(|\langle X_{:,j}, Y\rangle|\right)_{j\in S}\right\|_t \pm C_2 t_\star \|Y\|_2 s^{(1/t-1/2)+} (\sqrt{n} + \sqrt{s}), \quad (96)$$

*for a constant $C_2 = C_2(q, \kappa_{\mathrm{bulk}})$. In particular, if $s \leq n$ then*

$$\left\|\left(|\langle X_{:,j}, \lambda^\star\rangle|\right)_{j\in S}\right\|_t = t_\star \left\|\left(|\langle X_{:,j}, Y\rangle|\right)_{j\in S}\right\|_t \pm C_3 t_\star \tau_s \sqrt{n} s^{\max\{1/t, 1/2\}}. \quad (97)$$

*In the last display we used $\|Y\|_2 = \tau_s \sqrt{n} (1 + o(1))$ from Lemma A.5.*

*Proof.* We work on the intersection of the high-probability events supplied by Lemma A.8 (both equation 41 and equation 42), Lemma A.6, and the singular-value bound equation 22; this intersection has probability at least $1 - Ce^{-cn}$.

*(One-sided value comparison equation 93).* By optimality of $\lambda^\star$ and the definition of $t_\star$,

$$D(\lambda^\star) \geq D(t_\star).$$

Using the Fenchel-Young identity at the optimum (see equation 13) and equation 92,

$$D(\lambda^\star) = \left(1 - \tfrac{1}{q}\right)\|X^\top \lambda^\star\|_q^q, \qquad D(t_\star) = \left(1 - \tfrac{1}{q}\right)\|X^\top (t_\star Y)\|_q^q,$$

hence equation 93.

*(Dual-norm scale equation 94).* **Lower bound.** From $D(\lambda^\star) \geq D(t_\star)$ and equation 92,

$$D(\lambda^\star) \geq \left(1 - \tfrac{1}{q}\right)t_\star\|Y\|_2^2.$$

Since $D(\lambda^\star) \leq \langle Y, \lambda^\star\rangle \leq \|Y\|_2\|\lambda^\star\|_2$, we get

$$\|\lambda^\star\|_2 \geq \left(1 - \tfrac{1}{q}\right) t_\star \|Y\|_2.$$

**Upper bound.** Let

$$S(\lambda) := \sum_{j\in S} |\langle X_{:,j}, \lambda\rangle|^q, \qquad B(\lambda) := \sum_{j\notin S} |\langle X_{:,j}, \lambda\rangle|^q.$$

From equation 13,

$$D(\lambda^\star) = \left(1 - \tfrac{1}{q}\right)\left(S(\lambda^\star) + B(\lambda^\star)\right).$$

By Lemma A.8 (left inequality in equation 41),

$$B(\lambda^\star) \geq c_q (d - s) \|\lambda^\star\|_2^q.$$

Combining with $D(\lambda^\star) \leq \|Y\|_2\|\lambda^\star\|_2$ gives

$$\left(1 - \tfrac{1}{q}\right)c_q(d - s) \|\lambda^\star\|_2^{q-1} \leq \|Y\|_2. \quad (98)$$

Next, Lemma A.6 yields

$$\sum_{j\notin S} |\langle X_{:,j}, Y\rangle|^q = (d - s) m_q \|Y\|_2^q (1 + o(1)),$$

so $\|X^\top Y\|_q^q \geq c (d - s) \|Y\|_2^q$. From equation 91,

$$\left(t_\star\|Y\|_2\right)^{q-1} = \frac{\|Y\|_2^{q+1}}{\|X^\top Y\|_q^q} \leq \frac{1}{c} \cdot \frac{\|Y\|_2}{(d - s)}.$$

Comparing with equation 98 gives $\|\lambda^\star\|_2^{q-1} \leq C (t_\star\|Y\|_2)^{q-1}$ and hence $\|\lambda^\star\|_2 \leq C_1 t_\star \|Y\|_2$.

*(Bulk block equation 95).* Apply Lemma A.8 at level $t$ (two-sided inequality equation 42) with $\lambda = \lambda^\star$:

$$c_t^{1/t}(d-s)^{1/t}m_t^{1/t}\,\|\lambda^\star\|_2 \;\le\; \left\|\big(|\langle X_{:,j}, \lambda^\star\rangle|\big)_{j \notin S}\right\|_t \;\le\; C_t^{1/t}(d-s)^{1/t}m_t^{1/t}\,\|\lambda^\star\|_2.$$

Substitute $\|\lambda^\star\|_2 \asymp t_\star\|Y\|_2$ from equation 94.

*(Spike block equation 96-equation 97).* Set $h := \lambda^\star - t_\star Y$. Then

$$X_{:,S}^\top \lambda^\star \;=\; t_\star\, X_{:,S}^\top Y \;+\; X_{:,S}^\top h.$$

For any $t \ge 1$, the triangle inequality gives

$$\left\|\big(|\langle X_{:,j}, \lambda^\star\rangle|\big)_{j \in S}\right\|_t \;\le\; t_\star \left\|\big(|\langle X_{:,j}, Y\rangle|\big)_{j \in S}\right\|_t \;+\; \|X_{:,S}^\top h\|_{\ell_t},$$

and the analogous lower bound with a minus sign. By norm monotonicity in $\mathbb{R}^s$ and operator norm submultiplicativity,

$$\|X_{:,S}^\top h\|_{\ell_t} \;\le\; s^{(1/t-1/2)_+}\|X_{:,S}^\top h\|_2 \;\le\; s^{(1/t-1/2)_+}\, s_{\max}(X_{:,S})\,\|h\|_2.$$

From equation 22 with $t = \sqrt{s}$, $s_{\max}(X_{:,S}) \le C(\sqrt{n}+\sqrt{s})$ w.h.p., and from equation 94,

$$\|h\|_2 \;=\; \|\lambda^\star - t_\star Y\|_2 \;\le\; \|\lambda^\star\|_2 + t_\star\|Y\|_2 \;\le\; (C_1+1)\,t_\star\,\|Y\|_2.$$

Putting these together yields equation 96. If $s \le n$, Lemma A.5 gives $\|Y\|_2 = \tau_s\sqrt{n}(1+o(1))$ and

$$s^{(1/t-1/2)_+}(\sqrt{n}+\sqrt{s}) \;\le\; 2\sqrt{n}\,s^{\max\{1/t,1/2\}},$$

which implies equation 97. $\qquad\qquad\square$

## A.3    PROOF OF THEOREM A.2

With these lemmas in place, we are ready to prove Theorem A.2.

*Proof of Theorem A.2.* We work on the intersection of the high-probability events provided by Lemmas A.5, A.6, A.7, A.8, A.11, and A.12; this event has probability at least $1-Ce^{-c(d-s)}-Ce^{-c\sqrt{ns}}-2d^{-\gamma}$, consistent with Remark A.1. All constants implicit in $\asymp$ depend only on $(q, \kappa_{\text{bulk}})$.

Along the ray $\lambda = tY$, the one-dimensional dual objective

$$D(t) \;=\; t\,\|Y\|_2^2 - \frac{t^q}{q}\,\|X^\top Y\|_q^q$$

is strictly concave with unique maximizer given by the first-order condition (see equation 91)

$$t_\star^{q-1} \;=\; \frac{\|Y\|_2^2}{\|X^\top Y\|_q^q}. \tag{99}$$

By Lemma A.5, $\|Y\|_2^2 = \tau_s^2 n(1+o(1))$, and by the decomposition equation 40,

$$\|X^\top Y\|_q^q \;=\; n^q W_q\,(1+o(1)) \;+\; (d-s)\,m_q\,\tau_s^q\,n^{q/2}\,(1+o(1)) \;+\; O\big(s\,\tau_s^q\,n^{q/2}\big).$$

Substituting into equation 99 yields

$$t_\star^{q-1} \;\asymp\; \frac{\tau_s^2 n}{n^q W_q + \big((d-s)\,m_q + O(s)\big)\,\tau_s^q n^{q/2}} \qquad \text{w.h.p.} \tag{100}$$

By strong duality and Fenchel-Young (see equation 13),

$$\sup_\lambda D(\lambda) \;=\; \Big(1-\frac{1}{q}\Big)\,\|X^\top\lambda^\star\|_q^q \;=\; \Big(1-\frac{1}{q}\Big)\,\|\widehat{w}_p\|_p^p. \tag{101}$$

Evaluating $D$ on the ray at $t_\star$ and using $D(\lambda^\star) \ge D(t_\star)$ gives

$$\|\widehat{w}_p\|_p^p \;=\; \|X^\top\lambda^\star\|_q^q \;\ge\; \|X^\top(t_\star Y)\|_q^q \;=\; t_\star^q\,\|X^\top Y\|_q^q \;=\; \frac{\|Y\|_2^{\frac{2q}{q-1}}}{\|X^\top Y\|_q^{\frac{q}{q-1}}}. \tag{102}$$

Moreover, by Cauchy–Schwarz and equation 94,

$$\|X^\top \lambda^\star\|_q^q = \langle Y, \lambda^\star \rangle \;\leq\; \|Y\|_2\,\|\lambda^\star\|_2 \;\lesssim\; t_\star \|Y\|_2^2 \;=\; t_\star^q\,\|X^\top Y\|_q^q.$$

Combining with equation 102 we obtain the two-sided scale

$$\|\widehat{w}_p\|_p^p \;=\; \|X^\top \lambda^\star\|_q^q \;\asymp\; t_\star^q\,\|X^\top Y\|_q^q.$$

Using the coordinatewise KKT map equation 12,

$$\widehat{w}_p \;=\; \nabla f^\star(X^\top \lambda^\star) \;=\; \mathrm{sgn}(X^\top \lambda^\star) \odot |X^\top \lambda^\star|^{\,q-1}.$$

Hence, for any $r \in [1, p]$,

$$\|\widehat{w}_p\|_r \;=\; \|X^\top \lambda^\star\|_{(q-1)r}^{q-1}. \tag{103}$$

Split the $(q-1)r$-norm over the spike block $S$ and the bulk block $S^c$ and note that $\|u\|_t^t = \|u_S\|_t^t + \|u_{S^c}\|_t^t$ implies $\|u\|_t \asymp \max\{\|u_S\|_t, \|u_{S^c}\|_t\}$:

$$\|\widehat{w}_p\|_r \;\asymp\; \max\Big\{\; \big\|(|\langle X_{:,j}, \lambda^\star\rangle|)_{j \in S}\big\|_{(q-1)r}^{q-1}, \;\; \big\|(|\langle X_{:,j}, \lambda^\star\rangle|)_{j \notin S}\big\|_{(q-1)r}^{q-1} \;\Big\}. \tag{104}$$

*(We used $\max\{a, b\} \leq (a^t + b^t)^{1/t} \leq 2^{1/t}\max\{a, b\}$ for $t \geq 1$.)*

Set $t := (q-1)r \leq q$. By the spike-ray perturbation from Lemma A.12 (see equation 97 when $s \leq n$),

$$\big\|\big(|\langle X_{:,j}, \lambda^\star\rangle|\big)_{j \in S}\big\|_{\ell_t} \;=\; t_\star\, \big\|\big(|\langle X_{:,j}, Y\rangle|\big)_{j \in S}\big\|_{\ell_t} \;\pm\; C\, t_\star\, \tau_s\, \sqrt{n}\, s^{\max\{1/t,\, 1/2\}}. \tag{105}$$

*(If $s > n$, use the general form equation 96; the conclusion below is unchanged up to constants since $(\sqrt{n} + \sqrt{s})\, s^{(1/t-1/2)_+} \leq 2\sqrt{n}\, s^{\max\{1/t, 1/2\}} + s^{1 + (1/t - 1/2)_+}$, which is captured by the final "spike remainder" term.)* By Lemma A.11 at level $t$,

$$\big\|\big(|\langle X_{:,j}, Y\rangle|\big)_{j \in S}\big\|_{\ell_t} \;=\; n\, \|w^\star\|_t\, (1 + o(1)) \;\pm\; C\, \tau_s\, \sqrt{n}\, s^{\max\{1/t,\, 1/2\}}. \tag{106}$$

Combining equation 105-equation 106 and using $(a + b)^{q-1} \leq 2^{q-2}(a^{q-1} + b^{q-1})$ for $a, b \geq 0$, we obtain the following uniform two-sided bounds (recall $t = (q-1)r \leq q$):

$$\big\|\big(|\langle X_{:,j}, \lambda^\star\rangle|\big)_{j \in S}\big\|_{\ell_t}^{q-1} \leq C\Big\{\, t_\star^{q-1}\, n^{q-1}\, \|w^\star\|_t^{q-1} \;+\; (t_\star \tau_s \sqrt{n})^{q-1}\, s^{(q-1)\max\{1/t,\, 1/2\}} \,\Big\}, \tag{107}$$

$$\big\|\big(|\langle X_{:,j}, \lambda^\star\rangle|\big)_{j \in S}\big\|_{\ell_t}^{q-1} \geq c\Big(\, t_\star\, n\, \|w^\star\|_t \;-\; C\, t_\star\, \tau_s\, \sqrt{n}\, s^{\max\{1/t,\, 1/2\}} \,\Big)_+^{q-1}. \tag{108}$$

Applying the mean-value inequality to the map $z \mapsto z^{q-1}$,

$$|(x \pm y)^{q-1} - x^{q-1}| \leq C\,(x^{q-2} y + y^{q-1}),$$

with $x = t_\star n \|w^\star\|_t$ and $y = C t_\star \tau_s \sqrt{n}\, s^{\max\{1/t, 1/2\}}$, we obtain

$$\big\|\big(|\langle X_{:,j}, \lambda^\star\rangle|\big)_{j \in S}\big\|_{\ell_t}^{q-1} \;=\; t_\star^{q-1}\, n^{q-1}\, \|w^\star\|_t^{q-1}\, (1 + o(1)) \;\pm\; C\, (t_\star \tau_s \sqrt{n})^{q-1}\, s^{\max\{(q-1)/2,\, (q-1)/t\}}. \tag{109}$$

Recalling $t = (q-1)r$ and $\|w^\star\|_t \asymp \|w^\star\|_{(q-1)r}$, we obtain the spike contribution stated in equation 6. *(For completeness: specializing equation 96 to $t = q$ together with Lemma A.11 at $t = q$ yields the same rate and remainder exponent as in equation 109.)*

By Lemma A.12 (bulk control equation 95) together with equation 94,

$$\big\|(|\langle X_{:,j}, \lambda^\star\rangle|)_{j \notin S}\big\|_{(q-1)r} \;\asymp\; (d - s)^{1/((q-1)r)}\, t_\star\, \|Y\|_2.$$

Raising to the $(q-1)$-th power and using $\|Y\|_2 \asymp \tau_s \sqrt{n}$ (Lemma A.5),

$$\big\|(|\langle X_{:,j}, \lambda^\star\rangle|)_{j \notin S}\big\|_{(q-1)r}^{q-1} \;\asymp\; (d - s)^{1/r}\, \big(t_\star\, \tau_s\, \sqrt{n}\big)^{q-1}. \tag{110}$$

Plug equation 109 and equation 110 into equation 104. This yields

$$\|\widehat{w}_p\|_r \;\asymp\; \max\Big\{\, t_\star^{q-1}\, n^{q-1}\, \|w^\star\|_{(q-1)r}^{q-1}, \;\; (d-s)^{1/r}\, \big(t_\star\, \tau_s\, \sqrt{n}\big)^{q-1}, \;\; s^{\max\{1/r,\, (q-1)/2\}}\, \big(t_\star\, \tau_s\, \sqrt{n}\big)^{q-1} \,\Big\},$$

which is exactly the three-term unified bound in equation 6. When $r < 2(p-1)$ and $(d-s) \gtrsim s$, the third term is absorbed by the bulk term, recovering the two-term maximum.

In the proportional regime $(d-s) \asymp \kappa_{\text{bulk}} n$, balance the two leading terms in $\|X^\top Y\|_q^q$ (cf. equation 40) to define

$$n^q W_q \asymp (d-s) \tau_s^q n^{q/2} \iff n^{q/2} \asymp \kappa_{\text{bulk}} \frac{\tau_s^q}{W_q} \iff n_\star \asymp \left( \kappa_{\text{bulk}} \frac{\tau_s^q}{W_q} \right)^{\frac{2}{q-2}},$$

which matches equation 8.

*(i) Dual spike-dominated regime $n \gg n_\star$.* Then $\|X^\top Y\|_q^q \asymp n^q W_q$ and equation 100 gives

$$t_\star^{q-1} \asymp \frac{\tau_s^2 n}{n^q W_q} = \frac{\tau_s^2}{W_q} n^{-(q-1)}. \tag{111}$$

Consequently

$$(d-s)^{1/r} \left( t_\star \tau_s \sqrt{n} \right)^{q-1} \asymp \frac{\tau_s^{q+1}}{W_q} n^{\frac{1}{r} - \frac{1}{2(p-1)}}, \tag{112a}$$

$$s^{\max\{1/r, (q-1)/2\}} \left( t_\star \tau_s \sqrt{n} \right)^{q-1} \asymp \frac{\tau_s^{q+1}}{W_q} s^{\max\{1/r, (q-1)/2\}} n^{-\frac{1}{2(p-1)}}. \tag{112b}$$

In particular, when $r \le 2(p-1)$ the two "bulk-type" terms are of the same order (and are dominated by the spike main when $r \ge 2(p-1)$); this recovers equation 9.

*(ii) Dual bulk-dominated regime $n \ll n_\star$.* Then $\|X^\top Y\|_q^q \asymp (d-s) \tau_s^q n^{q/2}$ and

$$t_\star^{q-1} \asymp \frac{\tau_s^2 n}{(d-s) \tau_s^q n^{q/2}} = \frac{\tau_s^{2-q}}{(d-s)} n^{1-\frac{q}{2}}. \tag{113}$$

Therefore

$$(d-s)^{1/r} \left( t_\star \tau_s \sqrt{n} \right)^{q-1} \asymp \kappa_{\text{bulk}}^{\frac{1}{r}-1} \tau_s \, n^{\frac{1}{r}-\frac{1}{2}}, \tag{114a}$$

$$s^{\max\{1/r, (q-1)/2\}} \left( t_\star \tau_s \sqrt{n} \right)^{q-1} \asymp \kappa_{\text{bulk}}^{-1} \tau_s \, s^{\max\{1/r, (q-1)/2\}} n^{-1/2}. \tag{114b}$$

Taking the maximum together with the spike main term gives equation 10 whenever the third term is absorbed; otherwise the third term with exponent $\max\{1/r, (q-1)/2\} - 1/2$ may dominate.

This completes the proof of equation 6 (three-term form), the energy scale equation 102, hence the proof of Theorem A.2. $\qquad\square$

### A.4 TWO CONCRETE COROLLARIES: SINGLE SPIKE AND FLAT SUPPORT

We keep $p \in (1, 2]$, $q = \frac{p}{p-1} \in [2, \infty)$, $r \in [1, p]$, and $\kappa_{\text{bulk}} = \liminf(d-s)/n > 0$. Recall the unified bound from Theorem A.2. We will repeatedly use the identity

$$\|\widehat{w}_p\|_r \asymp \max \left\{ t_\star^{q-1} n^{q-1} \|w^\star\|_{(q-1)r}^{q-1}, \ (d-s)^{1/r} \left( t_\star \tau_s \sqrt{n} \right)^{q-1}, \right. \tag{115}$$

$$\left. s^{\max\{1/r, (q-1)/2\}} \left( t_\star \tau_s \sqrt{n} \right)^{q-1} \right\}, \tag{116}$$

together with

$$t_\star^{q-1} = \frac{\|Y\|_2^2}{\|X^\top Y\|_q^q}, \qquad n_\star \asymp \left( \kappa_{\text{bulk}} \frac{\tau_s^q}{W_q} \right)^{\frac{2}{q-2}}, \qquad W_q = \sum_{j \in S} |w_j^\star|^q, \quad \tau_s^2 = \|w^\star\|_2^2 + \sigma^2. \tag{117}$$

**Case (i): single spike** ($s = 1$). Let the support be $\{j_0\}$ and write $a := |w_{j_0}^\star| > 0$. Then

$$W_q = a^q, \qquad \|w^\star\|_{(q-1)r} = a, \qquad \tau_s^2 = a^2 + \sigma^2. \tag{118}$$

The transition scale simplifies to

$$n_\star \asymp \left( \kappa_{\text{bulk}} \frac{(a^2 + \sigma^2)^{q/2}}{a^q} \right)^{\frac{2}{q-2}}. \tag{119}$$

In equation 115, the spike remainder is dominated by the bulk term since

$$\frac{\text{spike remainder}}{\text{bulk}} = (d-1)^{-1/r} \ll 1 \quad \text{for large } d. \tag{120}$$

*Dual spike-dominated* ($n \gg n_\star$). Using the phase form equation 9, we obtain

$$\|\widehat{w}_p\|_r \asymp \begin{cases} \dfrac{(a^2 + \sigma^2)^{\frac{q+1}{2}}}{a^q} \, n^{\frac{1}{r} - \frac{1}{2(p-1)}}, & r \leq 2(p-1), \\[2ex] \dfrac{a^2 + \sigma^2}{a}, & r > 2(p-1). \end{cases} \tag{121}$$

*Dual bulk-dominated* ($n \ll n_\star$). Using equation 10,

$$\|\widehat{w}_p\|_r \asymp \max\left\{ \kappa_{\text{bulk}}^{\frac{1}{r}-1} (a^2 + \sigma^2)^{1/2} n^{\frac{1}{r} - \frac{1}{2}}, \; \kappa_{\text{bulk}}^{-1} (a^2 + \sigma^2)^{\frac{2-q}{2}} a^{q-1} n^{\frac{q}{2}-1} \right\}. \tag{122}$$

(The third term in equation 10 equals $\kappa_{\text{bulk}}^{-1} \tau_s n^{-1/2}$ and is dominated by the first term for large $n$.)

**Case (ii): flat signal on its support.** Assume $w_j^\star = a \, s_j$ for all $j \in S$ with $|s_j| = 1$ and $|S| = s$. Then

$$\|w^\star\|_2 = \sqrt{s} \, |a|, \qquad W_q = s \, |a|^q, \qquad \|w^\star\|_{(q-1)r} = s^{\frac{1}{(q-1)r}} |a|, \qquad \tau_s^2 = s \, a^2 + \sigma^2. \tag{123}$$

The transition scale grows linearly in $s$:

$$n_\star \asymp \left( \kappa_{\text{bulk}} \frac{(sa^2 + \sigma^2)^{q/2}}{s \, |a|^q} \right)^{\frac{2}{q-2}} = \kappa_{\text{bulk}}^{\frac{2}{q-2}} \, s \left( 1 + \frac{\sigma^2}{sa^2} \right)^{\frac{q}{q-2}}. \tag{124}$$

*Dual spike-dominated* ($n \gg n_\star$). From equation 9,

$$\|\widehat{w}_p\|_r \asymp \begin{cases} \dfrac{(sa^2 + \sigma^2)^{\frac{q+1}{2}}}{s \, |a|^q} \, n^{\frac{1}{r} - \frac{1}{2(p-1)}}, & r \leq 2(p-1), \\[2ex] s^{\frac{1}{r}-1} \, \dfrac{sa^2 + \sigma^2}{|a|}, & r > 2(p-1). \end{cases} \tag{125}$$

In the noiseless case ($\sigma = 0$),

$$r > 2(p-1): \quad \|\widehat{w}_p\|_r \asymp s^{1/r} |a|, \qquad r \leq 2(p-1): \quad \|\widehat{w}_p\|_r \asymp s^{\frac{q-1}{2}} |a| \, n^{\frac{1}{r} - \frac{1}{2(p-1)}}. \tag{126}$$

*Dual bulk-dominated* ($n \ll n_\star$). From equation 10,

$$\|\widehat{w}_p\|_r \asymp \max\left\{ \kappa_{\text{bulk}}^{\frac{1}{r}-1} (sa^2 + \sigma^2)^{1/2} n^{\frac{1}{r} - \frac{1}{2}}, \; \kappa_{\text{bulk}}^{-1} (sa^2 + \sigma^2)^{\frac{2-q}{2}} s^{1/r} |a|^{q-1} n^{\frac{q}{2}-1}, \tag{127}$$

$$\kappa_{\text{bulk}}^{-1} (sa^2 + \sigma^2)^{1/2} s^{\max\{1/r, \, (q-1)/2\}} n^{-1/2} \right\}. \tag{128}$$

When $r \leq 2(p-1)$ and $s \lesssim (d-s)$, the third term is absorbed by the first (Remark A.3).

## B    FROM INITIALIZATION SCALE TO AN EFFECTIVE $\ell_p$: A SLOPE-MATCHING VIEW

Figure S1 visualizes the mapping $\alpha \mapsto p_{\text{eff}}(\alpha)$ we use throughout. The construction is data-free (independent of $n$ and $\sigma$) and relies only on the gradient-flow potential that characterizes the two-layer DLN implicit bias. Pseudocode can be found in Algorithm 1.

We start from the separable potential

$$Q_\alpha(\beta) = \alpha^2 \sum_{i=1}^{d} q\left(\frac{\beta_i}{\alpha^2}\right), \tag{129}$$

$$q(z) = \int_0^z \operatorname{arcsinh}\left(\frac{u}{2}\right) du \;=\; 2 - \sqrt{4 + z^2} \;+\; z \operatorname{arcsinh}\left(\frac{z}{2}\right). \tag{130}$$

At the coordinate level, letting $\psi_\alpha(t) \equiv \alpha^2 q(t/\alpha^2)$ gives

$$\psi'_\alpha(t) = \operatorname{arcsinh}\left(\frac{t}{2\alpha^2}\right), \tag{131}$$

$$\psi''_\alpha(t) = \frac{1}{\alpha^2 \sqrt{4 + (t/\alpha^2)^2}} \;=\; \frac{1}{\sqrt{4\alpha^4 + t^2}}. \tag{132}$$

Asymptotics for $q$ control the limiting geometry (all logs are natural):

$$q(z) = \frac{z^2}{4} - \frac{z^4}{192} + O(z^6), \qquad\qquad z \to 0, \tag{133}$$

$$q(z) = z(\log z - 1) + 2 - \frac{1}{z} + O\left(\frac{1}{z^3}\right), \qquad\qquad z \to \infty. \tag{134}$$

Hence $Q_\alpha$ behaves like $\ell_2^2$ as $\alpha \to \infty$ and like an $\ell_1$-type penalty (up to a log) as $\alpha \to 0$.

To turn this into a quantitative $\alpha \mapsto p$ mapping, we evaluate $Q_\alpha$ on the $k$-sparse, unit-$\ell_2$ probes

$$\beta^{(k)} \in \mathbb{R}^d, \qquad \beta_i^{(k)} \in \{0, k^{-1/2}\}, \qquad \|\beta^{(k)}\|_2 = 1, \qquad \#\{i : \beta_i^{(k)} \neq 0\} = k. \tag{135}$$

For this family,

$$Q_\alpha(\beta^{(k)}) = \alpha^2 k\, q\left(\frac{1}{\alpha^2 \sqrt{k}}\right), \tag{136}$$

while $\ell_p$ (calibrated via $\|\beta\|_p^p$) has the exact scaling

$$\|\beta^{(k)}\|_p^p = k \left(\frac{1}{\sqrt{k}}\right)^p \;=\; k^{1-\frac{p}{2}}. \tag{137}$$

We now fit a log-log slope to the $k$-dependence of $Q_\alpha$ and match exponents. Fix $\alpha > 0$, choose a logarithmic grid $\mathcal{K} \subset \{1, 2, \ldots, d\}$ (e.g., up to $10^4$), and solve

$$\log Q_\alpha(\beta^{(k)}) \approx c(\alpha) + s(\alpha) \log k, \qquad k \in \mathcal{K}. \tag{138}$$

Comparing with equation 137 (which grows as $k^{1-p/2}$) yields

$$s(\alpha) = 1 - \frac{p_{\text{eff}}(\alpha)}{2} \quad\implies\quad p_{\text{eff}}(\alpha) = 2(1 - s(\alpha)). \tag{139}$$

The limits in equation 133–equation 134 imply

$$\alpha \to \infty: \quad Q_\alpha(\beta^{(k)}) = \frac{1}{4\alpha^2} + O\left(\frac{1}{\alpha^6 k}\right), \quad s(\alpha) \to 0, \quad p_{\text{eff}}(\alpha) \to 2, \tag{140}$$

$$\alpha \to 0: \quad Q_\alpha(\beta^{(k)}) = \sqrt{k}\left(\log(\tfrac{1}{\alpha^2 \sqrt{k}}) - 1\right) + 2\alpha^2 k - \alpha^4 k \sqrt{k} + O(\alpha^8 k^2 \sqrt{k}), \tag{141}$$

$$s(\alpha) \to \tfrac{1}{2},$$

$$p_{\text{eff}}(\alpha) \to 1.$$

Thus $p_{\text{eff}}(\alpha)$ increases smoothly and monotonically from 1 to 2 as $\alpha$ grows, exactly as depicted in Figure S1. The inverse problem—choosing $\alpha$ for a target $p^\star \in [1, 2]$—is the scalar root

$$p_{\text{eff}}(\alpha) = p^\star, \tag{142}$$

which we solve by bisection using the monotonicity in $\alpha$ (Algorithm 2).

---

**Algorithm 1** Slope-matching map $\alpha \mapsto p_{\text{eff}}(\alpha)$

---

**Require:** Log-grid $\mathcal{A}$ of $\alpha$ values; log-grid $\mathcal{K} \subset \{1, \ldots, d\}$ of $k$ values
**Ensure:** $\{(\alpha, p_{\text{eff}}(\alpha)) : \alpha \in \mathcal{A}\}$
 1: **for all** $\alpha \in \mathcal{A}$ **do**
 2:     Initialize lists $X \leftarrow [\,], Y \leftarrow [\,]$                  $\triangleright X = \{\log k\}, Y = \{\log Q_\alpha(\beta^{(k)})\}$
 3:     **for all** $k \in \mathcal{K}$ **do**
 4:         $z_k \leftarrow 1/(\alpha^2 \sqrt{k})$
 5:         Compute $q(z_k)$ using the closed form in equation 130; if $|z_k|$ is small, use the series $q(z) = z^2/4 - z^4/192 + z^6/2560 + \cdots$ for stability
 6:         $Q_k \leftarrow \alpha^2 \, k \, q(z_k)$
 7:         Append $\log k$ to $X$; append $\log Q_k$ to $Y$
 8:     **end for**
 9:     Fit $Y \approx c(\alpha) + s(\alpha) \, X$ by least squares
10:     $p_{\text{eff}}(\alpha) \leftarrow 2 \, (1 - s(\alpha))$                         $\triangleright$ by equation 139
11: **end for**
12: **return** $\{(\alpha, p_{\text{eff}}(\alpha)) : \alpha \in \mathcal{A}\}$

---

**Algorithm 2** Inverse map $p^\star \mapsto \alpha^\star$ by bisection in $\log \alpha$

---

**Require:** Target $p^\star \in [1, 2]$; grid $\mathcal{K}$; bracket $0 < \alpha_{\min} < \alpha_{\max}$ with $p_{\text{eff}}(\alpha_{\min}) \le p^\star \le p_{\text{eff}}(\alpha_{\max})$; tolerance $\varepsilon > 0$
**Ensure:** $\alpha^\star$ with $\left| p_{\text{eff}}(\alpha^\star) - p^\star \right| \le \varepsilon$
 1: $u_{\min} \leftarrow \log \alpha_{\min}, \; u_{\max} \leftarrow \log \alpha_{\max}$
 2: **while** $u_{\max} - u_{\min} > \varepsilon$ **do**
 3:     $u_{\text{mid}} \leftarrow \frac{1}{2}(u_{\min} + u_{\max}), \quad \alpha_{\text{mid}} \leftarrow e^{u_{\text{mid}}}$
 4:     Compute $p_{\text{eff}}(\alpha_{\text{mid}})$ via Algorithm 1 restricted to this single $\alpha$
 5:     **if** $p_{\text{eff}}(\alpha_{\text{mid}}) < p^\star$ **then**
 6:         $u_{\min} \leftarrow u_{\text{mid}}$
 7:     **else**
 8:         $u_{\max} \leftarrow u_{\text{mid}}$
 9:     **end if**
10: **end while**
11: **return** $\alpha^\star \leftarrow e^{(u_{\min} + u_{\max})/2}$

---

## C    Additional noise sweeps: $\sigma \in \{0, 0.5\}$

**Experimental protocol.** We replicate the experiments of §4.3 and §4.4 at two additional noise levels, $\sigma = 0$ and $\sigma = 0.5$, keeping everything else fixed (same $p \in \{1.1, 1.5, 1.9\}$ for explicit minimum-$\ell_p$ runs; same $\alpha \in \{0.00102, 0.0664, 0.229\}$ for DLNs with the same $\alpha \mapsto p_{\text{eff}}$ calibration as in Appendix B; same seeds and learning rates as indicated in the panel captions). Each plot overlays test MSE (left axis) and representative $\ell_r$ curves (right axis).

*What the figures show and why.* In Fig. S2-Fig. S9, the slopes and regime rules from Theorem 3.1 and Corollaries 4.1-4.2 are unchanged across $\sigma$; noise only rescales $\tau_s$ and thereby shifts the transition size $n_\star \asymp (\kappa_{\text{bulk}} \tau_s^q / W_q)^{2/(q-2)}$ [equation 8] and the spike-side plateau levels [equation 9]. Thus, compared to $\sigma = 0.1$ in the main text: (i) at $\sigma = 0$ elbows appear earlier and plateaus (for $r > 2(p-1)$) occur sooner and at lower levels; (ii) at $\sigma = 0.5$ elbows are delayed and spike-side plateaus are higher. Bulk-dominated panels retain the $n^{1/2}$ growth and the $r$-ordering in equation 10.

## D    Finite learning rate effects

We consider the single-spike case $w^\star = e_1$ and a small shape parameter $\alpha = 0.00102$ (so the calibrated $p_{\text{eff}}(\alpha) \approx 1.10$). We vary the learning rate $\text{lr} \in \{10^{-1}, 10^{-2}, 10^{-3}\}$ and the label-noise level $\sigma \in \{0, 0.1, 0.5\}$. All panels plot generalization error (left axis) and $\ell_{1.1}$ norm (right axis) versus sample size $n$.

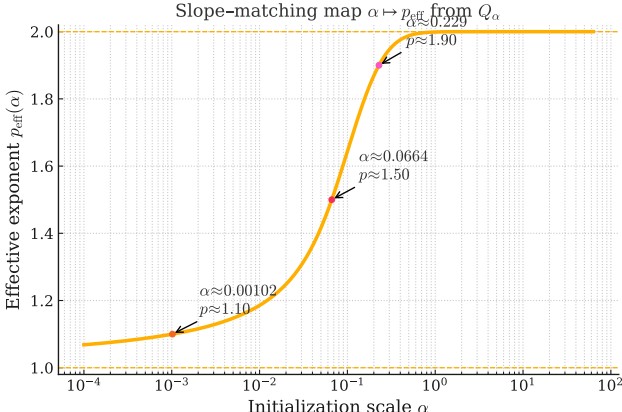

Figure S1: Slope-matching map $\alpha \mapsto p_{\text{eff}}(\alpha)$ (Algorithm 1), obtained by fitting the $k$-sparse scaling of $Q_\alpha(\beta^{(k)})$ against the exact $k^{1-p/2}$ scaling of $\|\beta^{(k)}\|_p^p$. Target points ($p \in \{1.1, 1.5, 1.9\}$) are annotated; their corresponding $\alpha$ are solved by Algorithm 2.

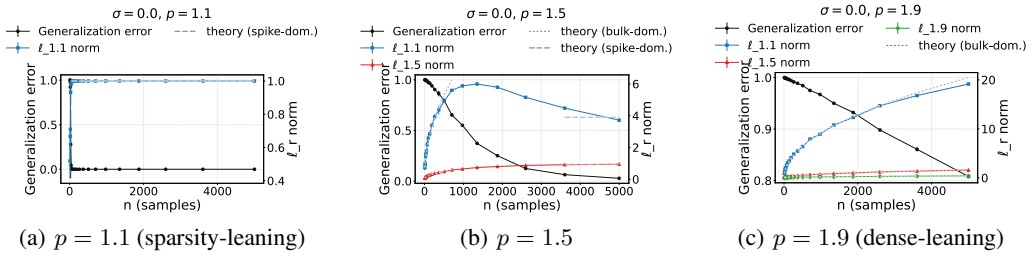

(a) $p = 1.1$ (sparsity-leaning)  (b) $p = 1.5$  (c) $p = 1.9$ (dense-leaning)

Figure S2: **Single spike $w^\star = e_1$; explicit minimum-$\ell_p$ interpolation ($\sigma = 0$).** Earlier elbows and lower spike-side plateaus than at $\sigma$=0.1; bulk-side traces keep the $n^{1/2}$ slope, consistent with equation 9–equation 10.

**Observed effect.** With **clean labels** ($\sigma = 0$), the $\ell_{1.1}$ norm is essentially flat across $n$ and insensitive to lr (Fig. S10), consistent with a low-$p_{\text{eff}}$ (sparse) implicit bias at small $\alpha$. When **label noise is present** ($\sigma \in \{0.1, 0.5\}$), increasing the learning rate makes $\ell_{1.1}$ *increase with* $n$ (Figs. S11, S12); the transition point (the "elbow") beyond which the norm would plateau shifts to larger $n$ as lr grows. Within the accessible sample sizes this rightward shift makes the curve look bulk-dominated and rising—*as if* the effective exponent $p_{\text{eff}}$ were larger.

**Why this happens.** Finite step size together with label/gradient noise injects additional stochasticity into the discrete dynamics. A useful approximation views (stochastic) gradient descent as a Langevin-type process with an *effective temperature* controlled by the learning rate and the noise level; this broadens the stationary distribution and leads to wider, less sparse solutions (Mandt et al., 2017; Smith et al., 2018; Yaida, 2018; Jastrzebski et al., 2017). For a single-spike target, that diffusion leaks mass into off-signal coordinates during early training, nudging the geometry away from "$\ell_1$-like" toward a higher-$p$ regime and delaying when the spike dominates—hence the elbow shifts right. With **clean labels**, the gradient remains aligned with the spike and the small-step implicit bias toward path/diagonal-norm solutions is recovered (Neyshabur et al., 2015a; Gunasekar et al., 2018). The same qualitative phenomenon also appears for the denser case $s$=50 with a smaller magnitude.

# E  LARGER SPARSITY $s$ FOR EXPLICIT $\min \|w\|_p$ LINEAR REGRESSION

We revisit the explicit $\min \|w\|_p$ experiments at larger sparsities $s \in \{500, 5000\}$ for $p \in \{1.1, 1.5, 1.9\}$ under the same Gaussian design and noise $\sigma = 0.1$ as in the main text. Each

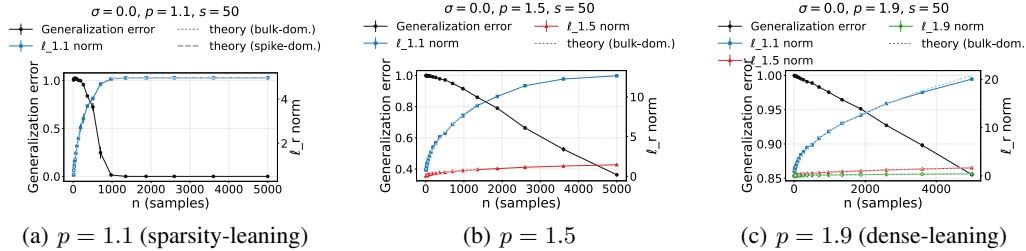

Figure S3: **Flat** $w^\star$ **($s = 50$); explicit minimum-$\ell_p$ interpolation ($\sigma = 0$).** Same slope/plateau rules as Corollary 4.2, with a reduced transition scale and lower absolute $\ell_r$ levels compared to $\sigma=0.1$.

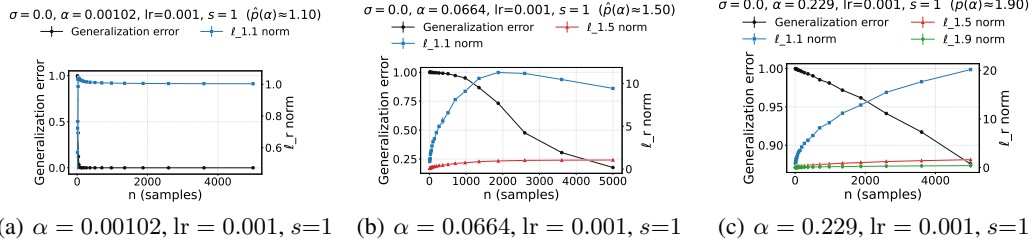

Figure S4: **Single spike** $w^\star = e_1$; **DLN ($\sigma = 0$).** With $\alpha$ calibrated to $p_{\text{eff}}(\alpha)$, the regime structure mirrors the explicit $p$ case: smaller $p_{\text{eff}}$ exhibits earlier spike dominance and plateaus for $r > 2(p-1)$; larger $p_{\text{eff}}$ stays bulk-dominated longer.

panel reports generalization error (left axis) and several $\ell_r$-norms of the *same* interpolating $w$ (right axis); gray dashed curves are the bulk/spike theory overlays used earlier.

**Comparison to $s=50$.** Across all three $p$ values, the larger-$s$ experiments reprise the main-text regime structure at larger sample sizes. For $p \approx 1$, lengthening the bulk-dominated segment makes the initial *increase* in generalization error clearly visible (especially at $s=5000$), after which the curve turns downward as alignment improves. For $p \in \{1.5, 1.9\}$, the same right-shift occurs yet the curves remain monotone; the rounder objectives keep the estimator from over-relying on noisy directions early on. In every panel, the blue $\ell_{1.1}$ curve remains a useful "regime meter": rapid growth signals bulk influence, and gradual approach toward the spike guide signals improving alignment—even though none of the $\ell_r$ curves truly flatten within our plotted range.

**Small $p$ (here $p=1.1$).** Relative to the $s=50$ panels in the main text, both larger-$s$ slices preserve the same two-phase story but the handoff happens later in $n$. At $s=500$ (Fig. S13a), generalization error is flat-to-slightly higher at small $n$ while $\|w\|_{1.1}$ rises rapidly; as $n$ grows, generalization error begins to fall and the blue curve bends toward (but, in our range, does not meet) the spike overlay. At $s=5000$ (Fig. S14a), the shape is unmistakable: generalization error *first increases* to a visible peak at intermediate $n$ and then drops. The $\ell_{1.1}$ curve keeps climbing throughout the displayed range, tracking the bulk-dominated guide before gradually approaching the spike prediction (without flattening). This "up-then-down" with more samples matches the double-descent picture for interpolating estimators—early fits lean on high-variance bulk directions; only later does the solution align with signal—well documented in linear and deep settings (Belkin et al., 2019; Nakkiran et al., 2020b; Hastie et al., 2022).

**Larger $p$ (here $p=1.5$ and $p=1.9$).** Compared to $s=50$, the curves again shift rightward in $n$, but the qualitative picture is unchanged: generalization error decreases *monotonically* over the whole range for both sparsities (Figs. S13b-c and S14b-c). The minimized $\ell_p$-norms (red for $p=1.5$, green for $p=1.9$) drift only slightly upward rather than plateauing, while the auxiliary $\ell_{1.1}$ diagnostic continues its steady growth along the bulk guide. The absence of an initial increase in generalization error is consistent with the rounder geometry of larger-$p$ balls: the interpolating solution spreads

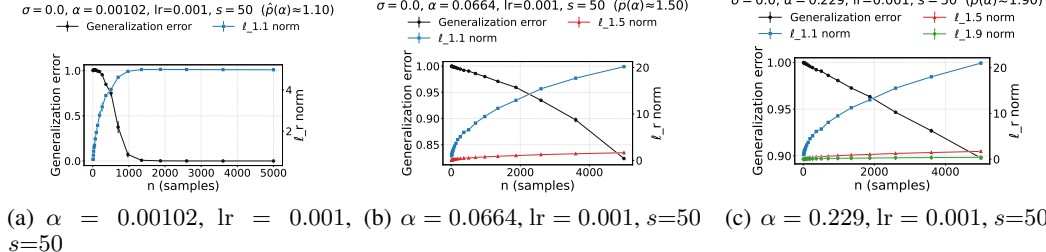

(a) $\alpha = 0.00102$, lr $= 0.001$, (b) $\alpha = 0.0664$, lr $= 0.001$, $s{=}50$ (c) $\alpha = 0.229$, lr $= 0.001$, $s{=}50$
$s{=}50$

Figure S5: **Flat $w^\star$ ($s = 50$); DLN ($\sigma = 0$).** The elbow shifts with support size as in the flat-support corollary; plateaus for $r > 2(p{-}1)$ occur earlier and at lower levels than at $\sigma{=}0.1$, while bulk-side $n^{1/2}$ growth persists where predicted.

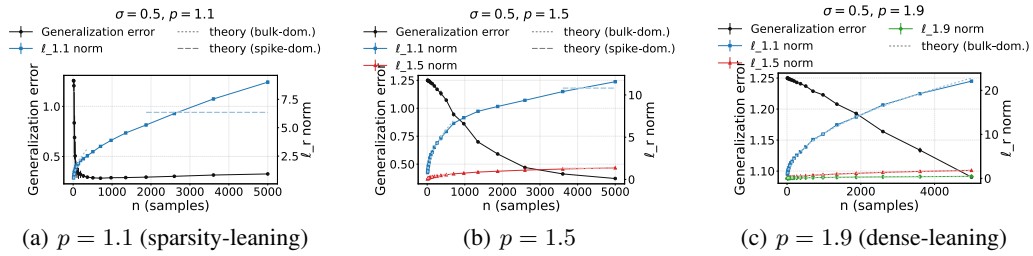

(a) $p = 1.1$ (sparsity-leaning)  (b) $p = 1.5$  (c) $p = 1.9$ (dense-leaning)

Figure S6: **Single spike $w^\star = e_1$; explicit minimum-$\ell_p$ interpolation ($\sigma = 0.5$).** Larger $\tau$ increases both $n_\star$ and plateau heights relative to $\sigma{=}0.1$. Bulk-dominated panels retain the $n^{1/2}$ trend; $r > 2(p{-}1)$ traces flatten only after the later transition, in line with equation 9–equation 10.

weight more evenly and avoids the brittle, variance-heavy fits that create the small-$p$ bump, echoing analyses of benign overfitting/ridgeless least squares and convex-geometric shrinkage of descent cones (Bartlett et al., 2020; Hastie et al., 2022; Chandrasekaran et al., 2012; Amelunxen et al., 2014).

## F  EXTENDING THE $\ell_r$-SCALING THEOREM TO DIAGONAL LINEAR NETWORKS

This section is a blueprint for porting our main $\ell_r$-scaling theorem from the minimum-$\ell_p$ interpolator to predictors selected by training *diagonal linear networks* (DLNs) with arbitrary depth. The goal is to reuse the entire spike+bulk argument with minimal surgery by swapping in the right implicit regularizer and the right one-dimensional balance. The guidance below covers both the two-layer case and the general depth-$D$ case, aligning with the characterization of implicit bias in DLNs proved by Woodworth et al. (2020).

In our $\min \ell_p$ analysis, the predictor among all interpolators is selected by a separable power potential, and the proof runs through a dual "link" that maps the ray variable back to primal coordinates. DLNs fit exactly the same template:

- For two layers, the implicit regularizer is the hypentropy-type separable potential, and the link is the corresponding odd, strictly increasing map (Woodworth et al., Thm. 1). Non-uniform initialization simply reweights coordinates multiplicatively throughout.

- For depth $D \geq 3$, the implicit regularizer is again separable but with a depth-dependent link; Woodworth et al. (Thm. 3) identify the unique depth-$D$ link and its inverse. Practically, you can treat it as "the $D$-link" playing the role occupied by the power map in $\min \ell_p$ and by the hypentropy link at $D = 2$.

No other structural change is needed: once the link is fixed, every step of our proof goes through with the same spike/bulk decomposition and the same ray reduction.

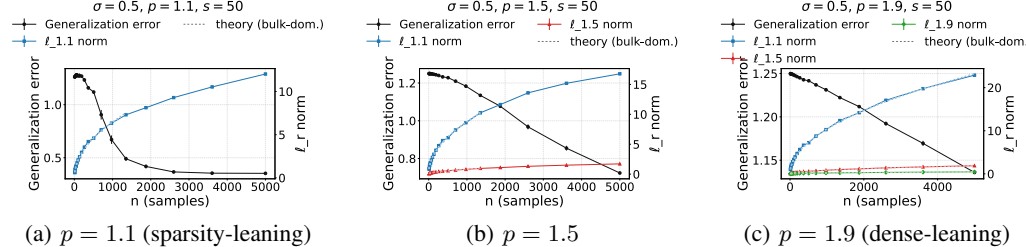

Figure S7: **Flat $w^\star$ ($s = 50$); explicit minimum-$\ell_p$ interpolation ($\sigma = 0.5$).** The same slope/plateau rules apply, but both the elbow and plateau heights shift upward with $\sigma$ via $\tau_s$ and equation 8.

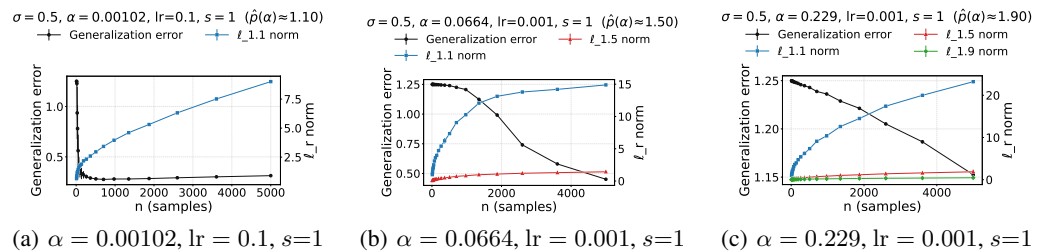

Figure S8: **Single spike $w^\star = e_1$; DLN ($\sigma = 0.5$).** After calibrating $\alpha \mapsto p_{\text{eff}}$, bulk growth persists to larger $n$ (larger $n_\star$), and spike-side plateaus for $r > 2(p-1)$ emerge later and at higher levels.

As in the $\min \ell_p$ proof, restrict the dual variable to the ray spanned by the labels and determine a single scale $t$ from a strictly monotone one-dimensional balance. Conceptually:

- In the *kernel-like window* (small arguments of the link on both spike and bulk), the link linearizes and the entire analysis collapses to the $p = 2$ case *verbatim*. This is the "lazy" regime.

- In the *rich-like window* (arguments large on the bulk and/or a dominant spike), the nonlinearity of the link controls the transition. For two layers, the balance yields a Lambert–$W$ controlled scale; for $D \geq 3$, the depth-$D$ link gives a faster, polynomial-in-initialization transition. You do not need a closed form—just the monotonicity and the small/large-argument asymptotics.

**Bulk block.** Replace the power moment used in the $\min \ell_p$ bulk bound by the depth-appropriate scalar functional that averages the link across a standard Gaussian coordinate. Operationally:

- Define a *bulk scalar* by applying the DLN link at the ray scale to a single Gaussian coordinate and taking its $\ell_r$ moment (to the $1/r$). This plays the exact role of $m_t^{1/t}$ in the $\min \ell_p$ proof.

- Use the same Gaussian embedding for the bulk design to lift this scalar to the full bulk contribution. In the kernel-like window you recover the $p = 2$ scaling exactly; in the rich-like window you get the accelerated depth-$D$ growth predicted by the link's large-argument behavior.

- Keep track of the global scaling coming from the link's overall prefactor (this carries the initialization scale); it multiplies both bulk and spike-remainder terms.

**Spike block.** On the spike coordinates, keep the original two-part structure:

- *Spike-main:* apply the link to the mean shift determined by the signal; if a single coordinate dominates the one-dimensional balance, the selected predictor saturates at the spike scale and becomes essentially independent of the initialization (up to lower-order logarithmic or depth-dependent corrections).

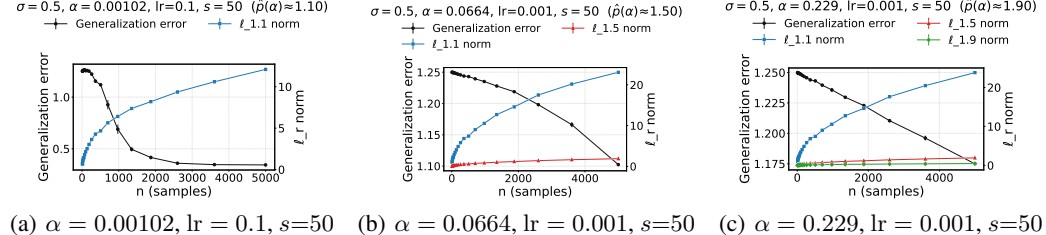

(a) $\alpha = 0.00102$, $\mathrm{lr} = 0.1$, $s{=}50$    (b) $\alpha = 0.0664$, $\mathrm{lr} = 0.001$, $s{=}50$    (c) $\alpha = 0.229$, $\mathrm{lr} = 0.001$, $s{=}50$

Figure S9: **Flat $w^\star$ ($s = 50$); DLN ($\sigma = 0.5$).** The $\sigma$-driven increase in $\tau_s$ shifts $n_\star$ to larger $n$; otherwise the bulk vs. spike regime behavior matches the theory and the explicit $p$ experiments.

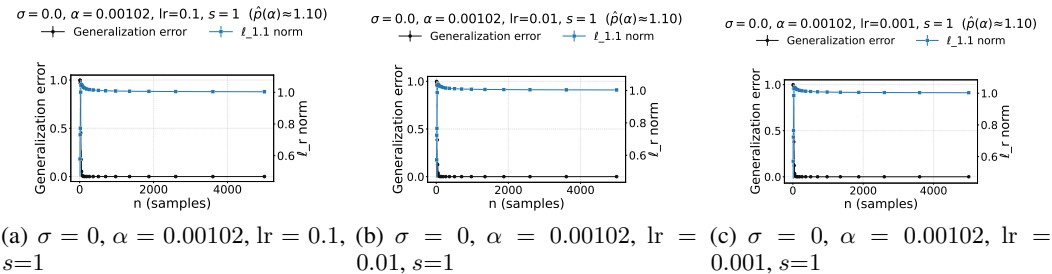

(a) $\sigma = 0$, $\alpha = 0.00102$, $\mathrm{lr} = 0.1$, $s{=}1$    (b) $\sigma = 0$, $\alpha = 0.00102$, $\mathrm{lr} = 0.01$, $s{=}1$    (c) $\sigma = 0$, $\alpha = 0.00102$, $\mathrm{lr} = 0.001$, $s{=}1$

Figure S10: **$w^\star = e_1$ (sparsity $s{=}1$), clean labels.** $\ell_{1.1}$ rapidly plateaus and is insensitive to learning rate, consistent with a low-$p_{\mathrm{eff}}$ implicit bias at small $\alpha$.

- *Spike-remainder:* control the residual Gaussian fluctuation by the same operator-norm and concentration events as in the $\min \ell_p$ proof; its $\ell_r$ size is the bulk scalar (at the ray scale) times $s^{\max\{1/r,\, 1/2\}}$, again multiplied by the link's global prefactor.

When spikes are *meek* relative to the bulk (no dominant coordinate), the spike block linearizes and you are back in the $p = 2$ laws.

**Unified bound.** After these replacements, the final display has the identical three-term structure:

$$\textit{DLN predictor's } \ell_r \textit{ size} = \textit{maximum of (spike-main, bulk, spike-remainder)},$$

with each term obtained from the $\min \ell_p$ counterpart by: (i) replacing the power link with the DLN link; (ii) inserting the link's global prefactor; and (iii) using the DLN bulk scalar in place of the power moment. In the kernel-like window this reproduces the $p = 2$ version *exactly*; in the rich-like window you get either bulk-controlled growth (Lambert–$W$ for two layers; depth-accelerated for $D \geq 3$) or spike saturation.

**Depth and initialization intricacy.**

- **Depth $D \geq 3$.** The depth-$D$ link is odd, strictly increasing, and has a simple linearization at the origin and an explicit rational form away from it (Woodworth et al., Thm. 3). This yields the same kernel-like reduction and a sharper rich-like transition than at $D = 2$. You never need its closed form—only its monotonicity and asymptotics.

- **Non-uniform initialization.** The per-coordinate *shape* of the initialization simply reweights the separable potential and carries multiplicatively through the link. Every bound inherits these weights in a purely multiplicative way (Woodworth et al., Thm. 1).

- **Limits.** Large initialization recovers the minimum-$\ell_2$ norm predictor; vanishing initialization recovers the minimum-$\ell_1$ predictor (with the usual caveats on how small "small" must be). These are the DLN analogues of the kernel and rich limits and hold for all depths covered above.

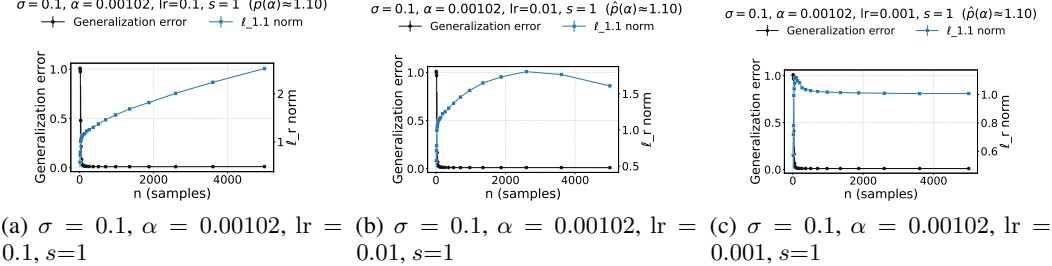

(a) $\sigma = 0.1$, $\alpha = 0.00102$, lr = 0.1, $s{=}1$
(b) $\sigma = 0.1$, $\alpha = 0.00102$, lr = 0.01, $s{=}1$
(c) $\sigma = 0.1$, $\alpha = 0.00102$, lr = 0.001, $s{=}1$

Figure S11: $w^\star = e_1$ (**sparsity** $s{=}1$**), moderate noise.** Larger learning rates produce a steadily rising $\ell_{1.1}$ and shift the elbow to larger $n$; decreasing lr suppresses the rise and restores a near-plateau.

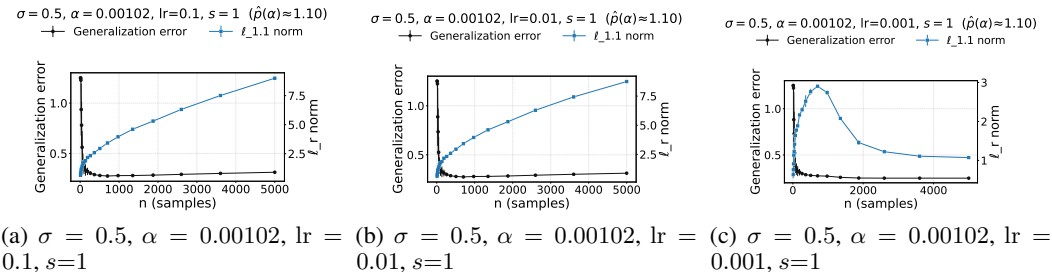

(a) $\sigma = 0.5$, $\alpha = 0.00102$, lr = 0.1, $s{=}1$
(b) $\sigma = 0.5$, $\alpha = 0.00102$, lr = 0.01, $s{=}1$
(c) $\sigma = 0.5$, $\alpha = 0.00102$, lr = 0.001, $s{=}1$

Figure S12: $w^\star = e_1$ (**sparsity** $s{=}1$**), heavy noise.** The learning-rate-induced increase in $\ell_{1.1}$ is strongest at high noise: lr=0.1 (and to a lesser extent 0.01) yields monotone growth with $n$, whereas lr=0.001 shows a transient bump and then relaxes toward a plateau—evidence that the elbow shifts right under larger lr.

**A handy dictionary for porting the proof.** To translate any display or lemma from the $\min \ell_p$ analysis to DLNs, we can make the following substitutions:

1. **Power link $\to$ DLN link**: replace the power map by the depth-appropriate link (hypentropy at two layers; the depth-$D$ link from Woodworth et al. otherwise), including its global prefactor.

2. **Ray scale $\to$ DLN balance**: keep the same one-dimensional, strictly monotone balance along the label ray; solve it numerically or via asymptotics (linear in the kernel-like window; Lambert–$W$ at two layers and power-law at depth $D \geq 3$ in the rich-like window).

3. **Bulk scalar**: replace the power moment by the $\ell_r$ moment of the DLN link applied to a single Gaussian coordinate at the ray scale; lift via the Gaussian embedding exactly as before.

4. **Spike block**: reuse the deterministic-plus-Gaussian decomposition, the operator-norm and concentration events, and the same $\ell_r$ geometry; only the link and its global prefactor change.

With the substitutions above, the $\ell_r$-scaling analysis for the minimum-$\ell_p$ interpolator transfers directly to DLNs of any depth. The proof structure, the spike/bulk decomposition, and the final three-term form remain identical; only the link and its scalar balance change. Two layers inherit a Lambert–$W$ bulk scale; deeper networks transition faster with initialization due to their depth-$D$ link. In the kernel-like window, everything collapses to the $p = 2$ bounds almost word-for-word.

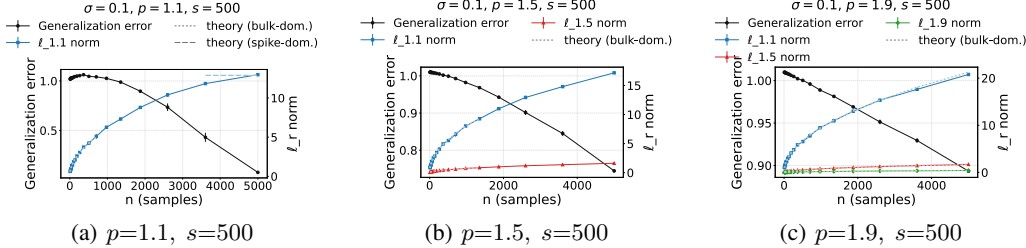

Figure S13: **Large sparsity,** $s$=500. Black—generalization error; colored—$\ell_r$-norms of the same interpolator (blue: $\ell_{1.1}$, red: $\ell_{1.5}$, green: $\ell_{1.9}$); gray dashed—bulk/spike overlays.

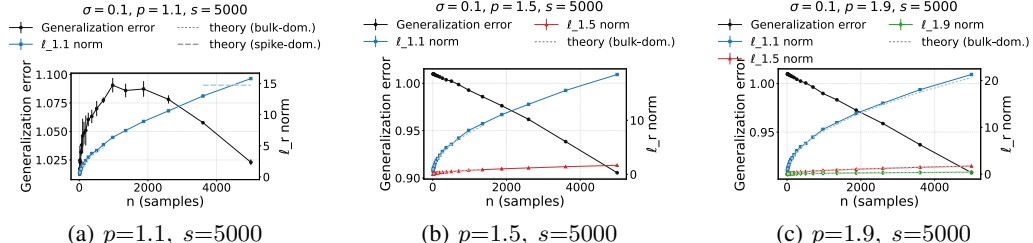

Figure S14: **Even larger sparsity,** $s$=5000. Same conventions as Fig. S13. Increasing $s$ shifts the bulk→spike crossover to larger $n$.

