# OpenReview forum: "Closed-form $\ell_r$ norm scaling with data for overparameterized linear regression and diagonal linear networks under $\ell_p$ bias"
_ICLR.cc/2026/Conference — ICLR 2026 Poster_

### Official Review · Reviewer_ksga · 2025-10-29

**Soundness:** 3
**Presentation:** 2
**Contribution:** 1
**Rating:** 4
**Confidence:** 3

**Summary:**

This paper aims to characterize one interesting term in overparameterized linear (models) regression: the behavior of $p$-norm minimum interpolator $||\hat{w}_{p}||_r$ against the number of data for different $r$ given $p\in(1, 2]$. To achieve this, the authors propose a novel "dual-ray" analysis to derive the corresponding scaling laws. Then they identify a data-dependent transition point and a universal threshold. They conduct some numerical experiments to verify their theoretical claims.

**Strengths:**

I think the most significant strength of this paper is the new "dual-ray" proof technique. I find it direct and intuitive, providing a better approach for the problem studied in this paper than approaches like GMT that might be too complex to be applied.

Furthermore, the derived $r$-norm of $p$-norm solution $ || \hat{w}_p ||_r $ as a function of several meaningful constants is also interesting. This form can somewhat support the authors' motivation "using norm-based bounds or proxies should be cautious".

**Weaknesses:**

However, I still have several concerns, and I will list them below.

1. I find the primary objective of this paper a bit unconvincing: studying the $r$-norm of the $p$-norm interpolator solution. There lacks a compelling motivation for this focus. Because it seems that it is the $p$-norm of the $p$-norm interpolator solution that matters, beyond the cautioning that generalization bounds may depend on the choice of $r$. This makes the scope of this paper a bit narrow.

2. The theoretical analysis heavily depends on the isotropic Gaussian features and $p \in (1, 2]$. This does not explore the validity of the findings that violates this assumption, e.g., perhaps the most direct case $p = 1$.

3. Theorem 3.1 is based on a general $w_{\star}$ (neither single spike nor flat), but the corresponding scaling laws are only studied in two very extreme case (single spike and flat) to discuss the corresponding laws. I understand that this might be due to the clarity reason, but a thorough discussion for how the scaling laws behave for arbitrary $w_{\star}$ should also be crucial. This also leads to the next weakness.

4. The actionable guidance (line 463 - 467), together with the implications for practice (line 81-85), is a significant overclaim. The theoretical results are built upon linear regression, and there even lacks discussion (both theoretical and empirical) for general $w_{\star}$ in this task, then I think claiming the corresponding actionable guidance for general settings is very misleading.

Minor: I think this paper should be reviewed in a venue for statistical learning theory or statistics, rather than ICLR (the connection with diagonal linear networks seems unnecessary because the authors directly convert the diagonal linear network to a minimizer of certain norm, which almost has nothing to do with the model itself).

**Questions:**

1. Why the authors do not include $p = 1$ (and other values of $p$) in the analysis?

2. How can the authors make their actionable guidance more reliable?

---

> ### Author Response · Authors · 2025-11-22
> **Thank you for your review (Author's comments 1 of 3)**
>
> We thank the referee for the careful review and helpful comments and appreciate the opportunity to clarify some points, which we have done below.  We sincerely hope that if our clarifications ease some of their concerns, they can feel that they are able to raise their score.
>
> **Weaknesses**:
>
> **Weakness 1**:
> Firstly, while we agree that the $\ell_p$ norm of min-$\ell_p$ norm interpolation is important, we still argue that the case for general  $r$ remains interesting for linear regression because it is such a fundamental problem.
>
> Secondly, some of the lessons learned may be relevant for other settings where $r=p$ is not the only relevant choice.  The first is that in some settings, e.g. the DLNs we study in the paper, the value of $p$ is not known *a-priori*, but changes with hyperparameters like the initialization. Furthermore, in the more complicated case of generalization bounds, where $p$ is also not known and even the question of whether a DNN is minimizing a norm is not that clear, a range of $r$ values are nevertheless used.   (See the general comment above for a longer discussion of these more complex systems, including indirect evidence that the qualitative behaviour we uncover for some $r,p$ values may be reflected in generalization bounds for DNNs).
>
> Thirdly, our theory tells you how $\ell_p$ norm of min-$\ell_p$ interpolator scales if you take $r=p$, in closed form, not just an upper bound. Indeed the mainstream technique to provide generalization guarantees for min-$\ell_p$ interpolators usually has two steps, localization and uniform convergence, where in localization they try to provide an upper bound for the $\ell_p$  norm of the solution so a level set can be formed, then they apply standard uniform convergence techniques to this level set. The upper bound given by the localization is not necessarily tight.
> Theorem 3.1 went beyond an upper bound, to give a full characterization of the $\ell_r$ norm, including the case of $r=p$. In fact, when $r<p$, $\ell_r$ norm itself is an upper bound for $\ell_p$ norm, so you can plug the $\ell_r$ norm right in as the localization control.
> In practice, when you use some gradient descent optimizer with some regularization, most of the time you don’t even know $p$. How can you control the $\ell_p$ norm then? Our theory, by giving a range of $r$, suggests that you must be weary because what effective $r$ you end up choosing as part of generalization guarantee can have qualitatively different scaling behavior.
>
> **Weakness 2**
>
> We chose the Gaussian design for its simplicity to present our results, which is standard in many other previous works (e.g. (Koehler et al., 2021; Donhauser et al., 2022)). The good news is, for a broad class of isotropic sub‑Gaussian designs with independent columns, we expect the same exponents and the same threshold $r^\star=2(p-1)$; the elbow $n^\star$ and prefactors rescale by constants that depend on distributional moments.
>
> For anisotropic designs with a covariance that’s not an identity matrix, whitening reduces the problem to the isotropic case, so the exponents and $r^\star$ persist while $n^\star$ shifts with the spectrum of the covariance. This is the case considered in (Koehler et al., 2021).
>
> If features have heavier tails, the bulk term can be controlled with sub‑Weibull tools provided the relevant $q$‑th absolute moments exist; constants worsen but exponents persist. When those moments do not exist, we can invoke some sort of truncation or robustification arguments(e.g., coordinate‑wise clipping before interpolation) which restores finite‑moment control. We probably need to lose some generality in terms of the assumptions we made, e.g. it now needs to be $d\gg n$ or $p$ can’t be too close to $1$ (so $q$ isn’t too big). There is a lot of wiggle room here.
>
> For the case of $p=1$, we didn’t include it in this work simply because 1) the conjugate $q$ will diverge and 2) $p=1$ corresponds to the LASSO regression which is extensively studied.

---

> ### Author Response · Authors · 2025-11-22
> **Author's comments 2 of 3**
>
> **Weakness 3**
>
> The reviewer correctly pointed out that our Theorem 3.1 applies to arbitrary $w^\star$, which we consider as one of the strengths of this work, extending the previous result (Donhauser et al., 2022) which only works with $w^\star = e_1$. We specialized to “single‑spike” and “flat” $w^\star$ only to make the scales concrete in figures. Due to the page limit we didn’t include other shapes of $w^*$, but it is expected that qualitatively the figures should look the same. The “shape” only moves the elbow and rescales the curves; it does not change the two‑regime structure or which $r$’s plateau vs grow.
>
> To see this, note that in Theorem 3.1 the only ways the target enters are through a few scalar summaries,
>  $$
>  W_q = |w^\star|q^q,\qquad
>  \tau_s^2 = |w^\star|2^2+\sigma^2,\qquad
>  s = |\operatorname{supp}(w^\star)|.
>  $$
>  The exponents in the scaling laws (including the universal threshold $r^\star = 2(p-1)$ and the slopes in the spike/bulk regimes) do not depend on the detailed pattern of entries in $w^\star$. The target shape only changes (i) the elbow location
>  $$
>  n^\star \asymp\Big(\kappa_{\mathrm{bulk}}\tau_s^q/W_q\Big)^{2/(q-2)},
>  $$
>  and (ii) the plateau height via norms such as $\Vert w^\star\vert_{(q-1)r}$. For a fixed support size $s$ and $\Vert w^\star\Vert_2$, the quantity $W_q=\Vert w^\star\Vert_q^q$ is extremized by our two examples: it is maximized by the single‑spike $w^\star$ and minimized by the flat $w^\star$ on its support. Any other “intermediate” shape (e.g. power‑law, block‑sparse) yields $W_q$ between these two extremes, hence an elbow $n^\star$ and plateau level lying between the single‑spike and flat cases. In other words, for arbitrary $w^\star$ we still see the same two‑regime behavior (pre‑ and post‑elbow) and the same plateau‑vs‑growth dichotomy across $r$; only the horizontal position of the elbow and vertical scaling of the curves shift within the range illustrated by the two extremes.
>
> **Weakness 4**:
>
> The basic motivation for our Actionable guidance is this:  For linear regression, we find a strong *qualitative difference*  between bulk and spike modes for the behaviour of the $\ell_r$ norm with data $n$.  That was a surprise to us, and suggests that in more complicated settings, similar strong qualitative differences in norm behaviour with $n$ may obtain when the effective $r,p$ combination changes.   As discussed at more length in General comment1 above, such qualitative behaviour may also be observed in these more complicated settings. Yes, there remains a ton of work to do to analyse these other systems, which is well beyond the scope of this paper, but we believe that the very recent empirical results in (*Position: Many generalization measures for deep learning are fragile* [arXiv:2510.18934]) suggests that our  “Actionable guidance (line 463 - 467)”, together with the “implications for practice (line 81-85)” may be useful for those who produce new generalization bounds.  Having said that, we are happy to soften the strength of our Actionable guidance if the reviewer would like us to.
>
> **Minor**: We thank the reviewer for their suggestion. While many other venues accommodate learning theory papers, ICLR also has a long tradition of hosting good learning theory research. For instance Donhauser et al., 2022, which inspired this work, is published here.
>
> Another very interesting point was raised by the reviewer about DLN. It is true that a multi-linear network (which is another common name for DLNs, used by Saxe 2014 [1], Nam 2024 [2] etc.) is still linear,  but because of this multi-linear structure, the optimization landscape for the parameters now becomes non-convex, unlike the landscape for the effective linear parameters. This tiny change has inspired many brilliant works, including the famous (Woodworth 2020 [3]), which identified the initialization-induced kernel and rich regimes by purely characterizing the “norm” (or potential function as they call it)  that the models are minimizing. You can almost say, at least with the case of Woodworth 2020, that what norms these DLNs are minimizing are exactly what’s interesting about these unconventional linear models.

---

> ### Author Response · Authors · 2025-11-22
> **Author's comments 3 of 3**
>
> **Question1**
> We didn't include $p=1$ because:
> 1) the conjugate $q$ will diverge if $p = 1$
> 2) $p=1$ corresponds to the LASSO regression which is extensively studied.
>
> We didn't include $p \notin [1,2]$ because in one of the seminal papers on DLNs (Woodworth 2020) they identified the kernel-rich regime transition which happens between $p \to 1$ (rich regime) and $p \to 2$ (kernel regime). By tuning the scale of initialization this is the only range of $p$ that the implicit bias can be achieved in DLNs. In linear regression, other values of $p$, while being less interesting in this kernel-rich regime context, might be analyzable, but that's beyond the capacity of the current paper.
>
> **Question 2**
>
> To soften the actionable guidance, we can perhaps
> 1) Point out that we have more confidence for limit (i) within linear systems where we have theoretically or empirically proved the regime changes depending on the relation between $r$ and $p$, and that our guidance for generalization measures relevant to non-linear DNNs is more speculative and needs further testing.
>
> 2) We can refer to *Position: Many generalization measures for deep learning are fragile* [arXiv:2510.18934]), which came out after the current paper was submitted,  and present our work as a potential direction towards theoretically understanding what has been observed in that work.
>
>
>
> [1] Saxe, Andrew M., James L. McClelland, and Surya Ganguli. "Exact solutions to the nonlinear dynamics of learning in deep linear neural networks." arXiv preprint arXiv:1312.6120 (2013).
>
> [2] Nam, Yoonsoo, et al. "An exactly solvable model for emergence and scaling laws." High-dimensional Learning Dynamics 2024: The Emergence of Structure and Reasoning. 2024.
>
> [3] Woodworth, Blake, et al. "Kernel and rich regimes in overparametrized models." Conference on Learning Theory. PMLR, 2020.

---

> > ### Comment · Reviewer_ksga · 2025-11-27
> >
> > I thank the authors for their clarification. I've updated my score.
> >
> > Minor (this does not affect my score): I think it is better to further emphasize the novelty of the "dual-ray" proof technique and its advantages, such that readers can appreciate the strengths more directly.

---

### Official Review · Reviewer_4hrx · 2025-10-31

**Soundness:** 3
**Presentation:** 3
**Contribution:** 3
**Rating:** 8
**Confidence:** 3

**Summary:**

The paper proves a closed form for the $\ell_r$ norm of the min $\ell_p$-norm interpolator under certain standard assumptions. Through experiments, the paper demonstrates that the trends predicted by the theory for linear regression, carry over to a slightly broader class of predictors.

**Strengths:**

- Very well-motivated, and clearly written.
- The mathematical derivations are sound and the application of the proof technique is, to my knowledge, novel in the context of analyzing min-norm interpolators.

**Weaknesses:**

- Theorem 3.1 is quite dense and could perhaps be split into the closed form result in eq 2, and the result characterizing the different sample size regimes
- It is not particularly easy to grasp what the main terms that determine $||\hat{w}_p||_r$ are. Would it be possible to give an interpretation of the quantities in equation 2?
- To further demonstrate the significance of the result, it would be nice to have a concrete example (perhaps supported by experiments) where one can take advantage of the relationship between $r$, $p$, $n$ and $d$ proved in Theorem 3.1.

**Minor remarks**
- It would help carry the message across more easily if one could visualize the motivation in lines 55-66 with the help of a teaser plot (perhaps a condensed version of figures 1-2?).
- Some of the markers are not visible in the figures e.g. dotted line in fig 1a.

**Questions:**

- Is there an explanation for the non-monotonical trend of the $\ell_{1.1}$ norm in figure 1b?
- What is $d$ for figures 1-2? does varying $\kappa=d/n$ change the trends in the figures? also would it be possible to indicate for each figure the predicted transition scale $\n^\star$?
- In [1] the authors try to extend the observations from Donhauser et al, 2022 to inductive biases present in non-linear models. Would a similar extension be applicable to the results of this paper?

[1] - https://arxiv.org/pdf/2301.07605

---

> ### Author Response · Authors · 2025-11-22
> **Thank you for your review (comment 1 of 3)**
>
> Thank you for your review! We appreciate the positive rating, acknowledgment of motivation and presentation, as well as constructive feedbacks! Below we try to address your concerns and questions:
>
> 1) Theorem 3.1 is too dense.
>
> We do apologize for having a main theorem which taks ¾ of a page. We had the same thought as the reviewer rightfully proposed, but then what we couldn’t decide is what to call the regime-wise statements. They are neither independent theorems nor corollaries. Maybe calling them remarks is fine but then we feel it might understate their importance, because we feel that the $2(p-1)$ threshold is non-obvious and quite interesting. What does the reviewer think is the best option? We are happy to work with the reviewer to improve the presentation of our main result.
>
> 2)  Would it be possible to give an interpretation of the quantities in equation 2?
>
> With pleasure! Intuitively, Eq. (2) decomposes the $\ell_r$-norm of the interpolator into three contributions that correspond to different parts of the parameter vector and to the spike–vs–bulk structure of $X^\top Y$.
> Recall that $t_\star$ in (1) is the “ray scale”: it is the amount by which we can push the dual variable $\lambda$ along the label direction $Y$ (i.e. $\lambda = tY$) before the $\ell_q$ norm constraint $\Vert X^\top \lambda\Vert_q^q$ becomes tight. The denominator in (1) splits $\Vert X^\top Y\Vert_q^q$ into:
>
> i) a spike term $n^q W_q$, coming from the true support $S$ (coordinates where $w^\star\neq 0$);
>
> ii) a bulk term $(d-s)m_q,\tau_s^q n^{q/2}$, coming from the many null coordinates $S^c$ (noise-driven correlations);
>
> iii) a smaller remainder, capturing finite-$s$ corrections.
>
> Which corresponds to the three terms in equation 2.
>
>
> 3) A  concrete example where one can take advantage of the relationship between $r$, $p$, $n$ and $d$ proved in Theorem 3.1.
>
> We didn’t prove a generalization bound in this work because the main theorem is already nearly a page long. But we think the reviewer would be glad to know that, as a working example, our Theorem 3.1 is consistent with, and helps explain, the fast‑rate result of [Donhauser et al. 2022] (Thm. 2.1). In the 1‑sparse Gaussian setting, our spike/bulk decomposition and the $r=p$ specialization give (i) an $n$-independent $\ell_p$ radius in the spike‑dominated regime and (ii) a bulk coefficient structure whose $\ell_2$ energy scales like $n/d$. These are precisely the inputs that drive the two branches $d^{2p-2}/n^p$ and $n/d$ in their localized uniform‑convergence analysis.
>
> To see this, just define $t := \mathbb{E}[(x^\top(\hat w_p - w^\star))^2]$, which equals $\Vert \hat w_p - w^\star\Vert_2^2$ under isotropic Gaussian design, and consider the localized class $\mathcal{F}_t = {x \mapsto x^\top(w - w^\star) : \Vert w \Vert_p^p \le M,\ \mathbb{E}[(x^\top(w - w^\star))^2] \le t}$ with radius $M \simeq \Vert \hat w_p\Vert_p^p$. In the spike‑dominated regime, our Theorem 3.1 with $r=p$ gives $M = \Theta(1)$, and the Gaussian width of $\mathcal{F}_t$ scales as $w(\mathcal{F}_t) \asymp M^{1/p} d^{1/q} t^{1/2 - 1/(2p)}$ with $q = p/(p-1)$. Plugging this into the standard localized fixed‑point inequality for squared loss, $t \lesssim (w(\mathcal{F}_t)/\sqrt{n})^2$, and solving for $t$ yields $t \lesssim M^2 d^{2p-2}/n^p$, i.e. the $d^{2p-2}/n^p$ branch once $M$ is constant.
>
> In the bulk‑dominated regime, our theorem shows that each null coordinate satisfies $ |\hat w_{p,j}| \approx \tau \sqrt{n}/d$, so the bulk energy is $\sum_{j\notin S} \hat w_{p,j}^2 \asymp \tau^2 n/d$; since the prediction risk under isotropic design is exactly $\mathbb{E}[(x^\top(\hat w_p - w^\star))^2]$, this gives the $n/d$ branch for $t$.
>
> 4) A teaser plot for motivation
>
> While a condensed version of Figure 1-2 is a good way to motivate the theory, we kindly refer the reviewer to a recent work (arXiv:2510.18934) in which the authors empirically show that many generalization measures used in practice can have drastically different scalings with data, whereas the true generalization error shows only a weak dependence on the hyperparameter settings. Maybe just citing this work can help motivate the theory? Again we are happy to work with the reviewer to write a better introduction.

---

> ### Author Response · Authors · 2025-11-22
> **comment 2 of 3**
>
> 5) Is there an explanation for the non-monotonic trend of the norm in Figure 1b?
>
> Very good question and we were puzzled by it too when the experimental results first came in. Turns out the answer lies in the experimental parameters. The non‑monotonic $\ell_{1.1}$ behavior in Fig. 1(b) is exactly what our spike–bulk theory predicts for the single‑spike setting, once we plug in the experimental parameters (here $d=50{,}000$, $p=1.5$, $r=1.1$, $\sigma=0.1$). For this configuration, Corollary  4.1 implies that in the bulk‑dominated regime the $\ell_{1.1}$ norm grows as $\Vert \hat w_p\Vert_{1.1}\asymp\tau(d-1)^{1/r-1}\sqrt{n}$, while in the spike‑dominated regime (since $r>2(p-1)=1$) it plateaus at $\Vert \hat w_p\Vert_{1.1}\asymp\tau^2$ (with $\tau^2=1+\sigma^2$). Balancing the spike and bulk contributions in $\Vert X^\top Y\Vert_q$ (Eq. (1)) gives a finite‑$d$ elbow at $n_\star\asymp\tau^2(d-1)^{2/3}\approx 1.4\times 10^3$. Evaluating the bulk formula at this elbow yields $\Vert \hat w_p\Vert_{1.1}^{\text{bulk}}(n_\star)\approx 14\tau^2$, i.e., the bulk branch around $n_\star$ is several times larger than the spike‑side plateau. As $n$ increases past $n_\star$, the spike takes over, the solution becomes much sparser, and the $\ell_{1.1}$ norm relaxes down toward the smaller plateau, producing the apparent “bump” in Fig. 1(b). In contrast, for the flat‑support experiment of Fig. 2(b) with $s=50$, Corollary 4.2 predicts an elbow at $n_\star^{\text{flat}}\asymp\tau^2(d-s)^{2/3}s^{1/3}\approx 5\times 10^3$, so the plotted range $n\le 5000$ remains bulk‑dominated and the $\ell_{1.1}$ curve continues its monotone $n^{1/2}$‑type growth without yet turning over toward the plateau.
>
> 6) What is $d$ for figures 1-2? does varying $\kappa$ change the trends in the figures? also would it be possible to indicate for each figure the predicted transition scale?
>
> We answered $d$ and the estimating $n_{*}$ above. Changing the aspect ratio $\kappa$ only moves the elbow; it does not change the qualitative trends. This is because in our theory the aspect ratio only enters through the bulk density $\kappa_{\text{bulk}} = (d-s)/n$, and this shows up just in the transition scale $n_\star \asymp (\kappa_{\text{bulk}}\tau_s^q/W_q)^{2/(q-2)}$ and in a few multiplicative constants in the bulk terms, not in the exponents of $n$ or in the threshold $r_\star = 2(p-1)$ itself. This means that changing $\kappa$ simply rescales when bulk and spike balance—moving the elbow $n_\star$ left or right and slightly rescaling the vertical level of bulk‑dominated curves—while the regime structure (which $\ell_r$ norms plateau vs. grow, and with what $n$-slopes) and the spike‑side plateau formulas remain unchanged. Qualitatively, the shapes of the curves are the same; only the $n$-location of the elbow and some constant factors move as $\kappa$ varies.

---

> ### Author Response · Authors · 2025-11-22
> **comment 3 of 3**
>
> 7) In [https://arxiv.org/pdf/2301.07605] (paper [1]) the authors try to extend the observations from [Donhauser et al, 2022] to inductive biases present in non-linear models. Would a similar extension be applicable to the results of this paper?
>
> This is an very interesting work that we followed a while ago. We thank the reviewer for pointing it out. Paper [1] transfers the insights from [Donhauser et al, 2022], which is on linear models and inspired our work, to non‑linear models by moving to convolutional kernels and specific neural network architectures.
> Because [1] spiritually also proceeds by exposing a simple low‑dimensional “signal’’ structure plus a high‑dimensional “noise’’ bulk (now in a convolutional kernel feature space rather than in standard coordinates), we believe that a conceptually similar extension of our work to genuinely non‑linear models is feasible. Concretely, we see a direct three-step path to extending our results beyond linear models, in close analogy to [1]: (i) first, transfer our spike‑vs‑bulk and $\ell_r$‑norm scaling arguments to nonlinear but fixed feature maps (e.g., convolutional kernels, random features, NTK‑type models) by working with their Gram matrices, in the same high‑dimensional hypercube setting; (ii) second, use existing implicit‑bias results for shallow nonlinear networks (ReLU/convolutional architectures with path‑ or layer‑wise $\ell_p$‑type norms) to identify the effective $\ell_p$‑regularizer selected by gradient flow, and then apply our linear machinery in that induced geometry to derive the corresponding $n_*$ and $r_*$ thresholds; and (iii) finally, combine these nonlinear norm‑scaling laws with the bias–variance analysis in [1] to characterize when interpolation in such networks is harmless or harmful as a function of an explicit “bias‑strength” knob (filter size, invariance, etc.), yielding a nonlinear analogue of our phase transition picture.
>
> Technically, this program would require new random‑feature and spectral analyses in the non‑Gaussian, hypercontractive settings used in [1], but the structural similarities---spike/bulk decomposition, explicit control of inductive‑bias strength via architecture, and interpolation in the high‑dimensional regime---make us optimistic that our methods can indeed be generalized along the same lines. We view this as a promising direction for future work and will clarify this potential extension in the discussion section.
>
> Final words:
> Thank you for recognizing the clarity and utility of our contribution. We appreciate your reading the work in the broader context of generalization theory and seeing value beyond its seemingly narrow focus. If this perspective reflects your overall assessment, we would be grateful if you would consider raising the score or, alternatively, add a brief note to help convey the paper’s scope to the other reviewers.

---

### Official Review · Reviewer_MDQs · 2025-11-01

**Soundness:** 3
**Presentation:** 3
**Contribution:** 3
**Rating:** 4
**Confidence:** 3

**Summary:**

This work studies the ell_r norm of the minimum ell_p norm interpolator for regression over gaussian data in the overparametrized setting where number of data n >> number of dimensions d. the setting assumes a sparse ground truth w^star.

In particular, they focus on how the ell_r norm grow when n grows when d/n has a limit kappa > 1.


The main finding (theorem 3.1) is that for r <= 2 ( p - 1 ), the ell_r norm of the min ell_p norm interpolator grows, while r > 2 (p -1), the same quantity plateaus.

Another finding is a "phase transition" from spiky-to-bulk dependency in terms of the number of samples (denoted n_star).

**Strengths:**

The theory is well-supported as seen in Figure 1. The provable behavior from the linear model seems to also carry over empirically to diagonal linear networks.

**Weaknesses:**

Can the authors give a more explicit reference to previous work that uses ell_r for bounding generalization error where r is not in {1,2}? I'm not convinced whether ell_r for r not in {1,2}. A more specific reference to a paper + lemma number would be helpful.

The legend of figure 1 looks to suggest that there should be error bars/confidence bands, but it's not visible in the plot. Is the result replicated?

**Questions:**

While i understand the spiky-vs-bulk part of X^TY, it's not clear to me how why the RHS of eqn (4) is said to be "spike-dominated". By contrast, in eqn (5), kappa_bulk clearly shows up on the RHS, so that makes sense.

Line 303 says middle panel exhibits a clear elbow near the predicted n_star. but it's not clear to me where is n_star on the middle panel plot.

The definition of t_star in eqn (1) is confusing. since it is a "definition", why is there a "w.h.p" at the end?

Minor:

Line 229 equation (8) links to the appendix when it could link to eqn (3) instead. Same issue in the caption of figure 1.

---

> ### Author Response · Authors · 2025-11-22
> **Thank you for your review**
>
> Thank you for the careful read and constructive suggestions! We can see that it’s the question of “why are we doing this” that hasn’t had you fully convinced. In general comment 2 we give a few precise literature pointers in which generalization bounds appear in a form with $\ell_r$ norm where $r \notin \\{1,2\\} $.
> We also address each of the technical questions raised below.
>
>
> **Weaknesses**:
> The questions about the broader relevance of our bounds for other $r,p$ combinations was also asked by other referees, and so we address it in general comment 2 above.  See also general comment 1 for some broader background.
>
> We thank the reviewer for their careful examination of the figures. In Fig. 1, we do include error bars calculated across multiple seeds. However, because the errors are very small, the error bars are not clearly visible at the provided scale. Zooming in on Fig. 1, particularly around n = 400 in panel (b), makes these tiny error bars easier to see.
>
> **Questions**:
> 1) The reviewer asks about  “spike-vs-bulk” – Briefly, in our notation, Eq. (4) is the spike-dominated simplification and Eq. (5) is the bulk-dominated simplification of Theorem 3.1. In terms of $n$, the only difference between these two regimes is whether $n\gg n^{\ast}$ or $n \ll n^{\ast}$. Intuitively when $n\gg n^{*}$ , the spike part of $X^\top Y$ dominates the bulk part.
> In the spike-dominated branch, the right-hand side involves only $\tau_s$, $W_q$, and $n$ but not $\kappa_{\mathrm{bulk}}$, because the scale $t_\star$ is controlled by the spike term in $\Vert X^\top Y\Vert_q^q$, namely $n^q W_q$, while the bulk contribution is lower order; see the decomposition of $\Vert X^\top Y\Vert_q^q$ in Eq. (40) and the ray-scale identity in Eq. (99), which yield $t_\star^{q-1}\asymp \tau_s^2/W_q\cdot n^{-(q-1)}$ in the spike regime (Eq. (111)), leading to Eq. (4). By contrast, in the bulk-dominated branch, $\Vert X^\top Y\Vert_q^q$ is controlled by the bulk term $(d-s)m_q\tau_s^q n^{q/2}$, so the simplification inherits an explicit $\kappa_{\mathrm{bulk}}$, as shown in Eq. (5) and its derivation (Eq. (113)–(114)).
>
> The spike vs bulk parts of $X^\top Y$ is indeed more intuitive, but we think the threshold $r=2(p-1)$ inside the spike-dom branch is also very intriguing and much less obvious, especially the fact that it comes purely from the algebra of norm equivalence and the spike-bulk decomposition, and was able to be validated by our numerical experiments.
>
> 2) About the exact location of $n^{*}$
>
> Again, we thank the reviewer for this good catch. For clarity, we fix the $d=50,000$ for all experiments. In Figure 1b, (with $w^\star=e_1$, $\sigma=0.1$, and $p=1.5$). Corollary  4.1 together with Eq. (8) gives the single-spike elbow as $n_\star \asymp (\kappa_{\mathrm{bulk}}\tau^q/W_q)^{2/(q-2)}$ with $q=p/(p-1)=3$, $\tau^2 = 1+\sigma^2$, and $W_q = 1$. In our finite-(d) setting, we approximate $\kappa_{\mathrm{bulk}}\approx (d-1)/n_\star$ at the elbow, which yields the self-consistent formula $n_\star^3 \approx \tau^6 (d-1)^2$, i.e. $n_\star \approx \big(\tau^6 (d-1)^2\big)^{1/3}$. Plugging in the actual parameters for the middle panel, $\sigma=0.1\Rightarrow \tau^2=1.01\Rightarrow \tau^6=(1.01)^3\approx 1.03$ and $d-1=49{,}999$, gives
>  $$
>  n_\star \approx \big(1.03\times 49{,}999^2\big)^{1/3} \approx 1.37\times 10^3,
>  $$
>  so the predicted elbow is at $n_\star \approx 1.4\times 10^3$. which is almost exactly where the $r>2(p-1)$ curves (e.g., the $\ell_{1.5}$ norm) bend from the $n^{1/2}$-like bulk growth into a near-plateau.
>
> 3) About the definition of $t_*$ in eq. 1:
> We thank the reviewer for pointing out this slip. The first $\asymp$ should be $=$ because that’s the KKT condition for the optimal ray scale $t_*$, and section  $\asymp$ is correct because that’s after we control the bulk and spike parts of $X^\top Y$. We use the word “define” because it’s the first time  $t_*$ appears in the theorem and we do need it in the later expression of $\Vert w_p \Vert_r$. Maybe there are better ways of introducing  $t_*$, and we are happy to revise the wording if the reviewer could kindly give some editing suggestions.
>
> **Minor**:
> Again we thank the reviewer for reading the manuscript carefully! We have fixed all of the points raised (and some other cross-reference slips) in the revision.
>
> **Overall**:
> We hope that the reviewer is happy with our answers and is willing to raise their score.

---

### Official Review · Reviewer_cAzm · 2025-11-04

**Soundness:** 3
**Presentation:** 4
**Contribution:** 3
**Rating:** 8
**Confidence:** 4

**Summary:**

This paper presents a unified theoretical and empirical study of how the parameter norms of overparameterized linear regression models scale with data under an ℓp-biased interpolation. Specifically, for minimum-ℓp interpolators with $p \in (1,2]$, the authors derive closed-form, high-probability laws describing how the family of norms $\{\lVert \widehat{w}_p \rVert_r\}$ for $r\in[1,p]$ changes with sample size $n$. A central insight is the identification of a data-dependent transition point n^\star (the “elbow”) separating bulk- and spike-dominated regimes, and a universal threshold $r^* = 2(p-1)$ that determines which norms plateau and which continue to grow. Using “dual-ray” analysis, the paper provides explicit expressions for these quantities and verifies the predictions empirically across both explicit $\ell_p$ regression and diagonal linear networks (DLNs) trained by gradient descent. By calibrating the DLN initialization scale $\alpha$ to an effective $p{\text{eff}}(\alpha)$, the authors demonstrate that DLNs inherit the same scaling laws, revealing a consistent connection between explicit and implicit bias. Overall, the work offers a clear, closed-form picture of how different norm measures behave with data growth and highlights that the predictive power of norm-based generalization proxies crucially depends on which ℓr norm is chosen.

**Strengths:**

See summary. The paper has a very clear and concrete contribution. The paper’s main strength lies in its clear and unified theoretical characterization of how all $\ell_r$ norms scale under $\ell_p$-biased interpolation. It successfully bridges explicit and implicit bias through an effective p_{\text{eff}}(\alpha) mapping in diagonal linear networks, and its empirical results convincingly validate the theoretical predictions across multiple regimes, offering practically valuable insights into the sensitivity of norm-based generalization measures.

**Weaknesses:**

I do not seen any major weaknesses of the study. The paper addresses a clear question. One minor weakness is that the connection to generalization is still less clear to me. It would be good to expand on this, and point out other work that also share the same weaknesses.

**Questions:**

Can authors comment on the Gaussian design choice? Do they expect Gaussian universality, or can the results significantly change under other distributions, and how under other natural distributions?

---

> ### Author Response · Authors · 2025-11-22
> **Thank you for your review**
>
> Thank you for your review! We appreciate the clear summary, positive feedback and most importantly, the constructive questions. Below we (1) tighten the connection to generalization and note closely related work with the same limitation, and (2) answer the Gaussian-universality question with precise claims about what changes under non‑Gaussian designs.
>
> 1) Clarifying the connection to generalization
>
> The referee asks a good question about the link between our analytic results to theories of generalization for DNNs.  Since all referees asked versions of this question, we have summarised our main points in general comment 1 above, and have added some extra references on norms with different $r$ in general comment 2.
>
> 2) On Gaussian design
>
> We chose the Gaussian design for its simplicity to present our results, like many other previous works (e.g. (Koehler et al., 2021; Donhauser et al., 2022)). The good news is, for a broad class of isotropic sub‑Gaussian designs with independent columns, we expect the same exponents and the same threshold $r^\star=2(p-1)$; the elbow $n^\star$ and prefactors rescale by constants that depend on distributional moments.
>
> For anisotropic designs with a covariance that’s not an identity matrix, whitening reduces the problem to the isotropic case, so the exponents and $r^\star$ persist while $n^\star$ shifts with the spectrum of the covariance. This is the case considered in (Koehler et al., 2021).
>
> If features have heavier tails, the bulk term can be controlled with sub‑Weibull tools provided the relevant $q$‑th absolute moments exist; constants worsen but exponents persist. When those moments do not exist, we can invoke some sort of truncation or robustification arguments (e.g., coordinate‑wise clipping before interpolation) which restores finite‑moment control. We probably need to lose some generality in terms of the assumptions we made, e.g. it now needs to be $d\gg n$ or $p$ can’t be too close to $1$ (so $q$ isn’t too big). There is a lot of wiggle room here.
>
> Final words:
> Thank you for recognizing the clarity and utility of our contribution. We appreciate your reading the work in the broader context of generalization theory and seeing value beyond its seemingly narrow focus. If this perspective reflects your overall assessment, we would be grateful if you would consider raising the score or, alternatively, add a brief note to help convey the paper’s scope to the other reviewers.

---

### Author Response · Authors · 2025-11-22
**General Comment 1 (of 2) --- On the connection to generalization measures.**

The **main goal of our paper** is to provide a closed‑form, high‑probability description of how the whole family  $ \\{ \lVert \widehat{w_p}  \rVert_r \\}_{r \in [1,p]} $ scales with data size $n$ under an $\ell_p$ bias for linear regression settings.   We believe that this is  *an interesting and fundamental problem in its own right*.  A surprising outcome of our analysis is the identification of significant qualitative differences in how these norms vary with $n$. This observation points towards an intriguing potential application in generalization bounds, which commonly rely on parameter norms. Could similar qualitative differences be found there as well?  In one way or another, all of the referees ask questions about this connection to bounds, and so rather than answer each partially, we thought it would be helpful to collect the questions in one place to sketch out this connection further.

1) Do other learning models also exhibit similar qualitative changes in their weight norms?

This question is much harder to answer analytically of course, but we do provide clear evidence that the weight norms of diagonal linear networks (DLNs) can exhibit similar qualitative behaviour to what we predicted analytically for the simpler setting of linear regression. By varying the initialization scale, the DLN system effectively changes its bias $p$.  Thus, for a single value of $p$, the behaviour of the weight norm $\ell_r$ can change from exponentially growing to non-monotonic as a function of $n$, even though the generalization behaviour does not show this non-monotonic behaviour. In this case, by extracting the value of $p$, we are able to make a close link to our bulk and spike-dominated modes for the DLNs.

2) Do generalization bounds for DNNs exhibit qualitative changes as a function of hyperparameter changes?

For a deep neural network (DNN) on complex data, an analytic analysis of the type we perform for linear regression is well beyond the scope of the current paper.  However, we want to point to a recent paper (*Position: Many generalization measures for deep learning are fragile* [arXiv:2510.18934]) that gives many empirical examples that mimic the sensitivity of the norm that we observed for DLNs in our current paper.  Namely, a small change in hyperparameters that does not strongly affect generalization performance can lead to large qualitative changes in popular generalization measures such as those based on the path norm. For example, see Fig. 1 in [arXiv:2510.18934]  which shows that changing the learning rate for the ADAM optimiser on a ResNET50 trained on FashionMNIST qualitatively changes the path-norm generalization bound from monotonic to non-monotonic behaviour (note that this bound is divided by $\sqrt{n}$) .  Clearly much more work needs to be done to establish the mathematical equivalence between our spike and bulk dominated modes and the behaviour of the path-norm, but we believe that these results are intriguing.  At the least , they point out that a small change in the inductive bias of a network can strongly affect measures that are based on a norm.  This is the main message we are trying to convey to the generalization theory community, besides the technicality of the theory itself.

---

### Author Response · Authors · 2025-11-22
**General comment 2 (of 2) --- References using $\ell_r$ with $r \notin \{1, 2\}$ for generalization bounds**

We agree that to a broader community, bounds with $r = 1$ or $2$ (lasso and ridge) are the most well known ones. However there is a well-established line of results where the generalization guarantee depends on an arbitrary norm not restricted to $1$ or $2$.  We list a few below:
1) Kakade et al  (NIPS 2008).
 Their Theorem 1 (Complexity Bounds) together with §3.1, Example (1) “$L_p/L_q$ norms” yields the Rademacher complexity bound
 $$
 \mathfrak{R}n(F_W)\ \le\ XW\ast\sqrt{(p-1)/n},
 $$
 for linear predictors when inputs are bounded in $\ell_p$ and the parameter is controlled in the dual $\ell_q$, with $1/p+1/q=1$. This gives generalization bounds that depend on $|w|_q$ for any $q\in[2,\infty)$, i.e., $r=q\notin\\{1,2\\}$ is permitted. See Theorem 1 and Example (1) with equation (3) in the paper.

2) Neyshabur et al (JMLR 2015).
 Appendix A.1, Lemma 15 (Rademacher complexity of linear separators with bounded $\ell_p$ norm) gives explicit upper bounds for linear predictors with any $1\le p\le\infty$, covering both $p\le 2$ and $p>2$ cases. The bounds scale with $\max_i \Vert x_i\Vert_{p^\ast}$ and thus translate directly to margin-based generalization with $\Vert w \Vert_{p}$ (hence $\Vert w \Vert_{q}$ with $q=p^\ast$). This treats $r=q$ over the full range, not only $1$ or $2$.

3) Gonon et al (ICLR2024).
Another work on path norm. While the Theorem 3.1 in the main text only gives a generalization bound in the $\ell_1$ path norm, they did prove a Lipschitz control for any output norm $r$ in lemma A.2, which can be plugged into generic Lipschitz-based generalization bounds. They also empirically calculated $\ell_1$, $\ell_2$ and $\ell_4$ path norm in Table 4. Note that in linear regression, $\ell_r$ path norm just reduces to $\ell_r$ norm of the solution.

4) Balcan et al (AISTATS 2014).
 Theorem 4 gives a data-dependent generalization bound for large $L_qL_p$ margins for any finite $p\ge 1$ (again $1/p+1/q=1$), where the guarantee depends explicitly on the $L_q$-norm of $w$ and the $L_p$-geometry of inputs via the $L_qL_p$ margin. The theorem statement and discussion appear explicitly in their paper (see Theorem 2 $\to$ Theorem 3 $\to$ Theorem 4 chain leading to the bound).

Together, these references show that $\ell_r$ (with $r\notin\\{1,2\\}$) is standard in norm-based capacity control for linear classes, and concrete bounds (with theorem/lemma identifiers) are available.

---

### Author Response · Authors · 2025-12-01
**Notes to the AC**

We thank the Area Chairs for the additional work imposed by the recent data‑leakage incident and for taking the time to reconsider our submission. The most common concern is about the motivation of this work and the connection to generalization. We answered those questions in our responses to each reviewer. Very briefly:

1. We argue that because linear regression is such a fundamental problem, a significant new result is important in and of itself.

2. Our results are also surprising: qualitatively different behavior of the norms, e.g., growing or plateauing, depending on the values of $r$ and $p$.

3. Many norms in the generalization bound literature depend on  $r \notin \\{1,2\\}$, and we list a few in our response.

4. A very recent paper that came out after ICLR deadlines (so the reviewers may not have seen it), entitled  "Position: Many generalization measures for deep learning are fragile, [Zhang and Louis 2025, arXiv:2510.18934]", demonstrates empirically that many generalization bounds for deep neural networks that are based on parameter norms can exhibit qualitatively similar trends to what we find in the much simpler case of linear regression here.

5. Finally, we answered each one of the other smaller questions in detail in our responses.  We note that the reviewer who initially gave the lowest score (ksga) updated their score from 4 to 6.

---

### Meta-Review · Area_Chair_Gh4n · 2026-01-06

**Summary:**

This paper started off as borderline paper with scores of 8 (cAzm), 8 (4hrx), 4 (MDQs), 4 (ksga).

The main strengths of the paper are its simple formulation and clear answer for the question asked. It presents a surprising result on the norm of the minimum norm interpolating solution. Reviewer ksga also found the proof to be a strong contribution of the paper.

On the side of concerns, the reviewers seemed to have a few primary concerns

1. First, is the setting of the problem (reviewers MDQs and ksga). That is why do we care about the minimum norm interpolating solution for $p \not\in \{1,2\}$. (Note the result does not apply for $p=1$).

2. Second, reviewer cAzm had concerns about the connection between generalization for linear regression and the connection to the given results

3. Third, reviewer cAzm, and ksga, question the usage of isotropic Gaussian design. This is in particular, to me, is a very valid concern, as now there is a mismatch between the covariance and the norm.

4. Fourth, reviewer MDQs and 4hrx found some part so the main theorem to be quite dense and difficult to parse.

**Reviewer Concerns:**

In terms of the addressing the reviewer concerns. I think the following concerns are addressed:

## Addressed

1. Setting of the problem

I think this issue is the one that is most sufficiently addressed. The authors provide sufficient context for their problem and I believe any reviewer reading these concerns would find these concerns addressed.

## Not sufficiently addressed

I found the response to this concern to be not very convincing.

2. Connection to generalization. I think the question is not about DNNs but for linear regression itself. It would be great if the authors could provide a concrete manner. In particular, if the authors could take the complexity bounds in General comment 2 and connect it to their norm scaling results. This would make the paper quite strong.

3. Isotropic Gaussian. I urge the authors to think deeply about this. In particular the comment on whitening data means the results directly translate to anistropic case, is not true.

4. Difficulty to parse. The authors have made some updates. But I urge the authors to further clarify bulk and spike terminology. In particular, to make them consistent with other literature in this field. Here spikes usually refer to isolated eigenvalues/singular values of covariance/data matrix. The bulk refers to continuous spectrum. However, this paper seems to talk about bulk and spike for a vector. $X^TY$.

**Reviewer Scores:**

Reviewer cAzm and 4hrx. Already had high scores. I do not see them reducing their score to lower than a 6. Hence I believe that they would be in favor of acceptance. Reviewer ksga, comments that their concerns were addressed and hence raised their score. So I believe that score change would be from a 4 to a 6.

Finally, this leaves reviewer MDQs. Here I believe their primary concern is the applicability of the setting which was addressed. I think they would be amenable to increasing their score.

---

### Decision · Program_Chairs · 2026-01-26

Accept (Poster)